# Dynamic Symmetric Point Tracking:
# Tackling Non-ideal Reference in Analog In-memory Training

Quan Xiao [* 1]   Jindan Li [* 1]   Zhaoxian Wu [1]   Tayfun Gokmen [2]   Tianyi Chen [1 3]

## Abstract

Analog in-memory computing (AIMC) performs computation directly within resistive crossbar arrays, offering an energy-efficient platform to scale large vision and language models. However, non-ideal analog device properties make the training on AIMC devices challenging. In particular, its update asymmetry can induce a systematic drift of weight updates towards a device-specific symmetric point (SP), which typically does not align with the optimum of the training objective. To mitigate this bias, most existing works assume the SP is known and pre-calibrate it to zero before training by setting the reference point as the SP. Nevertheless, calibrating AIMC devices requires costly pulse updates, and residual calibration error can directly degrade training performance. In this work, we present the first theoretical characterization of the pulse complexity of SP calibration and the resulting estimation error. We further propose a dynamic SP estimation method that tracks the SP during model training, and establishes its convergence guarantees. In addition, we develop an enhanced variant based on chopping and filtering techniques from digital signal processing. Numerical experiments demonstrate both the efficiency and effectiveness of the proposed method. Our code will be released at https://github.com/Jindanli898/E-RIDER.

## 1. Introduction

Recent breakthroughs in large vision and language models have been driven by the rapid maturation of modern hardware accelerators, including GPU, TPU (Jouppi et al., 2023),

*Equal contribution [1]Department of Electrical and Computer Engineering, Cornell University, New York, NY [2]IBM T. J. Watson Research Center, Yorktown Heights, NY [3]Rensselaer Polytechnic Institute, Troy, NY. Correspondence to: Tianyi Chen <chentianyi19@gmail.com>.

*Proceedings of the 43$^{rd}$ International Conference on Machine Learning*, Seoul, South Korea. PMLR 306, 2026. Copyright 2026 by the author(s).

NPU (Esmaeilzadeh et al., 2012), and emerging AI-specific chips such as NorthPole (Modha et al., 2023). Despite these advances, the energy costs of training and deploying such models still remain prohibitively high (Touvron et al., 2023; Brown et al., 2020).

To mitigate the energy bottleneck, analog in-memory computing (AIMC) has emerged as a promising platform that leverages resistive crossbar arrays to enable in-situ analog matrix–vector multiplications (MVMs). In AIMC, the weight matrix is stored as the conductance of resistive devices on a crossbar array, while MVM inputs and outputs are encoded as analog voltages and currents. By leveraging Kirchhoff's and Ohm's laws, AIMC devices perform MVMs directly without data movement, achieving $10\times$-$10,000\times$ lower inference energy than GPUs (Jain et al., 2019; Cosemans et al., 2019; Papistas et al., 2021).

Consider a model training problem with the objective $f(\,\cdot\,) : \mathbb{R}^D \to \mathbb{R}$ and its model parameters $W \in \mathbb{R}^D$ by

$$W^* := \underset{W \in \mathbb{R}^D}{\arg\min} \ \big\{ f(W) := \mathbb{E}_\xi[f(W;\xi)] \big\} \qquad (1)$$

where $\xi$ is a random data sample. Different from digital training, on AIMC hardware, the weights are updated by the so-called *pulse update*. When receiving electrical pulses, a resistive element updates its conductance based on its pulse polarity (Gokmen & Vlasov, 2016). At each pulse cycle, the weight changes by either $\Delta w_{\min} \cdot q_+(w)$ or $-\Delta w_{\min} \cdot q_-(w)$, where $\Delta w_{\min} > 0$ is the known *response granularity*, and $q_+(w)$ and $q_-(w)$ are fixed but unknown *response functions*, that describe how current weight $w$ responds to a small weight change $\Delta w_{\min}$. The response granularity $\Delta w_{\min}$ is one source of training errors, since it quantizes the desired gradients into finite conductance changes. Another key challenge in analog training is *update asymmetry*: at a fixed weight $w$, the conductance change induced by a positive pulse, $q_+(w)$, generally differs from that induced by a negative pulse, $q_-(w)$. This asymmetry stems from the underlying asymmetric physical mechanisms of analog devices, such as unequal conductive filament formation and rupture in HfO$_2$-based ReRAM devices (Jang et al., 2014), or nonreciprocal crystallization and amorphization in PCM devices (Burr et al., 2016). Although asymmetric devices offer practical advantages, such as faster response

and lower pulse voltage (Stecconi et al., 2024), making them attractive for deployment, their asymmetric update behavior can induce systematic weight drift during analog training.

By decomposing $q_+(w)$ and $q_-(w)$ into a *symmetric component* and an *asymmetric component* that capture the device-specific drift (will explain in (6)), we model the analog update as a scaled desired update plus an asymmetric drift term. Mathematically, each resistive element admits a device-specific *symmetric component* $F(\cdot)$, an *asymmetric component* $G(\cdot)$. To apply an increment $\Delta W_k = \alpha \nabla f(W_k; \xi_k)$ with stepsize $\alpha$, the analog device performs the following `Analog Update` (Wu et al., 2025; Li et al., 2025)

$$W_{k+1} = W_k + \Delta W_k \odot F(W_k) - |\Delta W_k| \odot G(W_k) + b_k \tag{2}$$

where $|\cdot|$ and $\odot$ denote the coordinate-wise absolute value and multiplication, and $b_k$ quantifies the stochastic discretization error of sending a finite number of pulses of length $\Delta w_{\min}$ to change each $d$-th element of $W_k$. Unlike on digital devices, where $G(\cdot) \equiv 0$, $G(\cdot) \not\equiv 0$ on analog devices induces the *update asymmetry*. Specifically, even without the discretization error $b_k$, the training optimum $W^*$ is generally not a stationary point of the `Analog Update` because

$$\mathbb{E}[W_{k+1}|W_k = W^*] = W^* - \alpha \mathbb{E}_\xi[|\nabla f(W^*; \xi)|] \odot G(W^*)$$

does not equal $W^*$ whenever the device-asymmetry term is nonzero (i.e., $G(W^*) \neq 0$) as $\mathbb{E}_\xi[|\nabla f(W^*; \xi)|] > 0$. This motivates us to study the symmetric points where we do not have device-asymmetry as defined below.

> **Definition 1.1 (Symmetric point).** A point is termed *symmetric point* (SP) if the following equation holds
>
> $$G(W^\diamond) = 0. \tag{3}$$

With this, SGD with `Analog Update` can be interpreted as implicitly minimizing a regularized objective of the form

$$\min_W \quad f(W) + \Theta(\sigma^2 \|W - W^\diamond\|^2) \tag{4}$$

where $\sigma^2$ denotes the variance of the stochastic gradient noise (Wu et al., 2025). Consequently, obtaining an accurate estimate of $W^\diamond$ is a key prerequisite for analog training algorithm design, as it enables effective compensation of the drift induced by *update asymmetry*.

In existing algorithm design and convergence theory of analog training (Wu et al., 2024; 2025), $W^\diamond$ is assumed to be 0 for simplicity. However, in practice, the SP is device-specific, so it requires per-device estimation and calibration to zero before training (Gokmen, 2021). To estimate the SP, a typical zero-shifting (ZS) algorithm has been proposed by

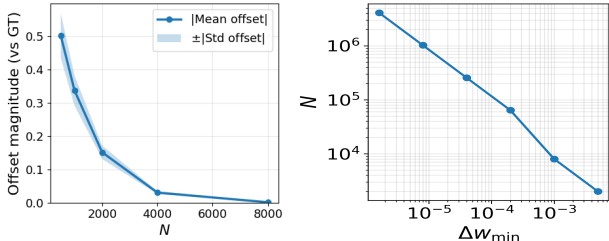

*(a)* Offset vs. pulse budget.  *(b)* Pulse cost v.s. $\Delta w_{\min}$

*Figure 1.* Trade-off between SP estimation accuracy and pulse cost for ZS algorithm. (a) For each $N$, we obtain per-cell SP estimates on a $512 \times 512$ array, and compute the mean and standard deviation across all cells. We plot the offsets of these statistics relative to the ground truth. (b) As $\Delta w_{\min}$ decreases, achieving a target accuracy (e.g., $\leq 1\%$ relative mean error) needs substantially more pulses.

Kim et al. (2019), which sends alternative up-down pulses to push the device to its SP. However, as shown in Figure 1, its pulse count significantly impacts the precision of SP estimation, and higher-precision devices (smaller $\Delta w_{\min}$) require more pulses to achieve a target accuracy. These observations lead to a critical question regarding the trade-off between pulse overhead and accuracy for SP estimation:

> **Q1)** *How to quantify the pulse efficiency of the SP estimation algorithm in terms of the number of pulses?*

After obtaining the estimate of SP and calibrating it to zero (through a reference device), we can leverage existing zero-SP-based analog training algorithms for the model training (Wu et al., 2025; 2024; Gokmen, 2021). However, this renders the subsequent training phase vulnerable to estimation error propagated from the SP estimation stage when we do not have enough pulse budget; see Figure 2. Intuitively, the training process can also provide valuable feedback for SP tracking: if $W^\diamond$ is inaccurate, the resulting compensation deviates from the true drift term in (4), which in turn degrades training. Considering the benefits of integrating SP tracking with training, another natural question is

> **Q2)** *Can we develop a dynamic SP tracking algorithm during training to reduce the overall pulse complexity?*

To address the above two questions, we develop a dynamic SP-tracking algorithm based on multi-sequence update and the filtering theory from digital signal processing.

### 1.1. Our contributions

We summarize the contributions of this paper as follows:

**C1)** We provide the first theoretical complexity analysis of the standard ZS algorithm for SP estimation (Kim et al., 2019). Crucially, we prove that the pulse count required for accurate estimation scales with the inverse device update granularity $\mathcal{O}(\Delta w_{\min}^{-1})$. This reveals a

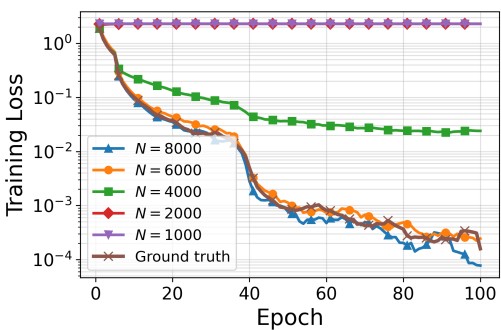

*Figure 2.* Training loss on MNIST (LeNet-5, TT-v1 (Gokmen & Haensch, 2020)) using ground-truth SP and SPs estimated with different numbers of pulses $N$ via zero-shifting Algorithm 1.

fundamental **device dilemma**: as hardware becomes more precise, static SP estimation becomes more expensive, motivating the need for dynamic tracking.

**C2)** We propose a novel ResIdual learning with Dynamic symmEtric point tRacking (RIDER) algorithm that jointly performs SP tracking and model training on the analog devices. We proved that the proposed RIDER algorithm can estimate the SP during the model training with reduced pulse complexity, and achieves the same rate $\mathcal{O}(1/\sqrt{K})$ as standard SGD.

**C3)** Building on C2), we propose an Enhanced variant of RIDER, termed E-RIDER, that leverages the frequency theory to accelerate the SP estimation and a periodic synchronization mechanism to reduce weight programming costs, making the method feasible for energy-constrained edge implementations.

**C4)** Numerical experiments validate the effectiveness and efficiency of the E-RIDER for simultaneous SP tracking and model training. With a more accurate SP estimation, the proposed method is able to improve the analog training accuracy in neural network training, and outperforms other analog training algorithms.

### 1.2. Related works

AIMC accelerates deep learning by executing compute-intensive MVM operations directly within resistive crossbar arrays (Yao et al., 2017; Wang et al., 2019). However, hardware imperfection introduces errors into the training. A series of works proposes methods to offload parts of error-sensitive operations on digital circuits to mitigate the issues (Nandakumar et al., 2020; Wan et al., 2022; Yao et al., 2020). While effective, this digital overhead compromises the intrinsic energy and latency benefits of the architecture. Consequently, this paper focuses on the more challenging regime of fully on-chip training, which maximizes operations in the analog domain to preserve efficiency.

The primary challenges of on-chip training arise from inherent hardware imperfections, such as asymmetric up-

---

1: **Inputs:** initialization $W_0$; response granularity $\Delta w_{\min}$
2: **for** $n = 0, 1, \ldots, N-1$ **do**
3:     draw $\epsilon_n \in \mathbb{R}^D$ with $\epsilon_n^{[d]} \sim \mathcal{U}(\{-\Delta w_{\min}, \Delta w_{\min}\})$
4:     update $W_{n+1}$ via analog pulse update, i.e. (7)
5: **end for**
6: **outputs:** $W_N$

---

date (Burr et al., 2015; Chen et al., 2015), reading/writing noise (Agarwal et al., 2016; Zhang et al., 2022; Deaville et al., 2021), device/cycle variations (Lee et al., 2019). Researchers have developed various architectural and algorithmic solutions to mitigate these imperfections. For example, (Gokmen & Haensch, 2020; Gokmen, 2021; Wang et al., 2020; Huang et al., 2020) introduce an auxiliary analog array to suppress update noise; Li et al. (2025) addresses the constraints of limited update granularity through a multi-tile orchestration method. Complementing these empirical methods, recent works have established rigorous theoretical frameworks to model these imperfections, proposing *residual learning* as a principled mechanism to recover training performance (Wu et al., 2024; 2025). However, their theory requires the SP to be exactly zero, and empirically they are not robust to nonzero SP. Most closely related to our work, Rasch et al. (2024) proposed a dynamic SP tracking method AGAD with strong empirical performance. However, their methodology lacks theoretical guarantees even without chopping. As will be demonstrated in this paper, omitting a residual learning mechanism results in inferior performance compared to the E-RIDER in this work.

## 2. Pulse Complexity of SP Estimation

In this section, we model the ZS algorithm as a discrete-time dynamic using pulse updates, and analyze its convergence as the number of pulses increases.

### 2.1. Mathematical Model of Zero-shifting Algorithm

We will first introduce the mathematical formulation of `Analog Update` in (2), and then abstract the update rule for the ZS algorithm (Kim et al., 2019).

For each AIMC device, we stack the response functions $q_-(w)$ and $q_+(w)$ into $Q_+(W)$ and $Q_-(W) : \mathbb{R}^D \to \mathbb{R}^D$, and express the analog weight updates as

$$W_{k+1}^{[d]} = \begin{cases} W_k^{[d]} + \Delta W_k^{[d]} \cdot Q_+(W_k)^{[d]}, & \Delta W_k^{[d]} \geq 0, \\ W_k^{[d]} + \Delta W_k^{[d]} \cdot Q_-(W_k)^{[d]}, & \Delta W_k^{[d]} < 0 \end{cases} \quad (5)$$

where $\Delta W_k \in \mathbb{R}^D$ is the target update increment and $A^{[d]}$ denotes the $d$-th element of $A \in \mathbb{R}^D$. Following (Wu et al., 2025), the symmetric component $F(\cdot)$ and the asymmetric component $G(\cdot)$ of this device are given by

$$F(W) := (Q_-(W) + Q_+(W))/2, \quad (6a)$$

$$G(W) := (Q_-(W) - Q_+(W))/2, \quad (6b)$$

pluging which into (5) leads to the `Analog Update` (2).

ZS algorithm (Kim et al., 2019) estimates the SP of an analog device by applying positive (up) and negative (down) update pulses alternatively until the weight no longer drifts, which we model as the following stochastic dynamic

$$W_{n+1} = W_n + \epsilon_n \odot F(W_n) - |\epsilon_n| \odot G(W_n) \quad (7)$$

where $\epsilon_n \in \mathbb{R}^D$ and $\epsilon_n^{[d]} \sim \mathcal{U}(\{-\Delta w_{\min}, \Delta w_{\min}\})$ is uniformly randomly generated as either up or down pulse, which is summarized in Algorithm 1. Note that the original ZS algorithm proposed in (Kim et al., 2019) is a cyclic sampling version of Algorithm 1; i.e. $\epsilon_{2n'}^{[d]} = \Delta w_{\min}$ and $\epsilon_{2n'+1}^{[d]} = -\Delta w_{\min}$. As the convergence analysis for the cyclic version is built upon the stochastic version (7) and the convergence rate order of them are the same, we only present the stochastic case in the main paper and defer the cyclic-case result to the Appendix C.3–C.4.

### 2.2. Convergence Analysis for ZS Algorithm

Following (Wu et al., 2025), we focus on the analog devices with the following training-friendly response functions.

> **Definition 2.1** (Training-friendly response functions).
> Response functions $q_+(\cdot)$ and $q_-(\cdot)$ are said to be training-friendly if they satisfy
> - **(Positive-definiteness)** there exist $q_{\min} > 0$ and $q_{\max} > 0$ such that $q_{\min} \le q_+(w) \le q_{\max}$ and $q_{\min} \le q_-(w) \le q_{\max}, \forall w$; and,
> - **(Differentiable)** Both $q_+(\cdot)$ and $q_-(\cdot)$ are differentiable.

Training-friendly response functions capture a wide range of AIMC devices: the positive lower bound $q_{\min}$ excludes dead-update regions, the upper bound $q_{\max}$ rules out unbounded gains, and the differentiability is motivated by continuous conductance mechanisms (ion migration, phase boundary movement, electrochemical reactions) in PCM (Burr et al., 2016; Le Gallo & Sebastian, 2020), ReRAM (Jang et al., 2014; 2015; Stecconi et al., 2024), ECRAM (Tang et al., 2018; Onen et al., 2022).

For this family of response functions, the convergence of ZS algorithm in Algorithm 1 can be characterized below.

**Theorem 2.2** (Convergence rate of Algorithm 1). *Considering the response functions in Definition 2.1, then the iterates given by Algorithm 1 with $N$ pulses satisfy*

$$\frac{1}{N} \sum_{n=0}^{N-1} \mathbb{E}\left[\|G(W_n)\|^2\right] \le \mathcal{O}\left(\frac{1}{N\Delta w_{\min}}\right) + \Theta(\Delta w_{\min}).$$

The proof of Theorem 2.2 is deferred to Appendix C.1. Theorem 2.2 shows that, for a device with response granularity $\Delta w_{\min}$, the minimal achievable SP estimation error is $\delta = \Theta(\Delta w_{\min})$, meaning that higher-precision devices (smaller $\Delta w_{\min}$) can attain more accurate SP estimates.

However, it also reveals a **device dilemma**: to achieve the same fixed estimation error $\delta \ge \Theta(\Delta w_{\min})$, the required number of pulses scales inversely with the response granularity $\Delta w_{\min}$, which is $N = \mathcal{O}(\delta^{-1}\Delta w_{\min}^{-1})$, suggesting that high-precision devices with smaller $\Delta w_{\min}$ require more pulses for SP calibration in Algorithm 1. In practice, the response granularity $\Delta w_{\min}$ of AIMC device is made sufficiently small to ensure the gradient conversion precision (Rao et al., 2023; Sharma et al., 2024), so that the computational complexity of Algorithm 1 is relatively high.

*Remark* 2.3. With additional assumptions on device behavior, we can derive tighter bounds for $N$ w.r.t. target error $\delta$ for specific classes of response functions; we defer these results (Theorem C.2) to Appendix C.2. Nonetheless, the number of pulses required to achieve a target SP estimation error still remains inversely proportional to $\Delta w_{\min}$.

**Empirical evidence.** To illustrate the negative impact of $\Delta w_{\min}$ on the pulse complexity of the ZS algorithm, we test its pulse cost on a ReRAM array device presets from IBM AIHWKit simulator (Rasch et al., 2021), which is essentially analog device with linear response functions; see detailed setup in Appendix F.2. Figure 1a shows the empirical estimated offset of the SP mean and standard deviation when $\Delta w_{\min} = 0.001$, defined as the ground-truth statistic minus the estimated statistic. It can be seen that achieving relative $1\%$ error requires more than 2000 number of pulses in ZS algorithm, which is computationally heavy. In Figure 1b, we report the smallest $N$ such that the relative error of the estimated SP mean is within $1\%$ under different device response granularity $\Delta w_{\min}$. The result shows that higher-precision devices (smaller $\Delta w_{\min}$) require substantially more pulses to reach the same target error $\delta$. Moreover, the relationship of $\Delta w_{\min}$ and $N$ is nearly inversely linear, as predicted by our Theorem 2.2.

To study the effect of SP estimation error on model training, we train a LeNet-5 on MNIST with TT-v1 using SPs estimated from Algorithm 1 with different numbers of pulses and report the results in Figure 2. It shows that a smaller $N$ in Algorithm 1 leads to significant training loss degradation and even failure to converge, which motivates the design of dynamic SP tracking algorithm.

## 3. Dynamic SP Tracking Algorithm

In this section, we aim to develop a dynamic SP tracking algorithm that leverages the inherent SP-attraction property of AIMC devices, i.e., repeated gradient pulse updates drive

the device state towards its SP. The challenge of algorithm design is deferred to Appendix B.3.

## 3.1. Basic Design

We first design a basic dynamic SP-tracking algorithm based on Residual Learning in (Wu et al., 2025) that rectifies the update asymmetry in AIMC devices. Residual Learning introduces another AIMC device to learn the asymmetric residual and compensate for it through gradient updates. However, it *assumes an exactly zero SP*, which in turn requires precise zero-shifting calibration prior to training. To enable dynamic SP tracking, we introduce an SP-tracking variable $Q_k \in \mathbb{R}^D$ that approaches the SP over iterations. For each iteration $k$, we aim to solve the following bilevel problem to correct the asymmetry bias

$$\min_{W \in \mathbb{R}^D} \|P^*(W, Q_k) - Q_k\|^2,$$
$$\text{s.t. } P^*(W, Q_k) \in \arg\min_{P \in \mathbb{R}^D} f(W + \gamma(P - Q_k)) \quad (8)$$

where $P - Q_k$ serves as a zero-shifting (residual) vector. The lower-level problem seeks the optimal compensation vector to correct device-induced errors, while the upper-level problem drives the system toward a zero-shifted state. In short, if we can design a SP-tracking sequence $Q_k$ such that $Q_k \to W^\diamond$, then the optimal solution for (8) is $P^* = Q^* = W^\diamond$ and $W^* = \arg\min_W f(W)$. We make the following assumptions.

**Assumption 3.1** (*L*-smoothness). The objective $f(W)$ is *L*-smooth, that is for any $W, W' \in \mathbb{R}^D$, it follows

$$\|\nabla f(W) - \nabla f(W')\| \le L\|W - W'\|. \quad (9)$$

**Assumption 3.2** (Unbiasness and bounded variance). The samples $\{\xi_k\}$ are i.i.d. sampled from a distribution. Moreover, the stochastic gradient is unbiased and has bounded variance, i.e., $\mathbb{E}_{\xi_k}[\nabla f(W_k; \xi_k)] = \nabla f(W_k)$ and $\mathbb{E}_{\xi_k}[\|\nabla f(W_k; \xi_k) - \nabla f(W_k)\|^2] \le \sigma^2$.

**Assumption 3.3** ($\mu$-SC condition). The objective $f(W)$ is strongly convex (SC) with modulus $\mu$.

**Assumption 3.4.** The stochastic discretization error $b_k$ in (2) satisfies $\mathbb{E}[b_k] = 0$ and $\text{Var}[b_k] = \Theta(\alpha \Delta w_{\min})$.

Assumptions 3.1–3.3 are standard in the stochastic optimization (Bottou et al., 2018), and are all used in the pilot theoretical analysis for analog training (Wu et al., 2025). Assumption 3.4 is used and verified in (Li et al., 2025).

With these assumptions, the lower-level solution of problem (8) is unique, given by

$$P^*(W, Q_k) = Q_k + (W^* - W)/\gamma. \quad (10)$$

Basically, the optimal $P^*(W, Q_k) - Q_k$ tracks the difference of current $W_k$ towards the optimal solution $W^*$. Besides, the gradient of the bilevel problem (8) with respect to $W$ can be computed via chain rule

$$\nabla_W \|P^*(W, Q_k) - Q_k\|^2 = -2(P^*(W, Q_k) - Q_k)/\gamma.$$

Let us define $\bar{W}_k = W_k + \gamma(P_k - Q_k)$. Inspired by alternating bilevel algorithms (Chen et al., 2021; Ji et al., 2021; Hong et al., 2020; Ghadimi & Wang, 2018; Maclaurin et al., 2015; Franceschi et al., 2017; 2018; Pedregosa, 2016) and putting $P_k$ and $Q_k$ on different analog devices, the update rules using `Analog Update` in (2) can be written as

$$P_{k+1} = P_k - \alpha \nabla f(\bar{W}_k; \xi_k) \odot F_p(P_k)$$
$$\qquad - \alpha|\nabla f(\bar{W}_k; \xi_k)| \odot G_p(P_k) + b_k \quad (11a)$$
$$W_{k+1} = W_k + \beta(P_{k+1} - Q_k) \odot F_w(W_k)$$
$$\qquad - \beta|P_{k+1} - Q_k| \odot G_w(W_k) + b'_k \quad (11b)$$

where both $b_k, b'_k$ denote the discretization errors, and we denote the corresponding response functions by $F_p, G_p$ for the device of $P_k$, and by $F_w, G_w$ for the device of $W_k$.

**Design for $Q_k$ sequence.** We first note that the SP-drifting issue arises only in the presence of stochastic noise (see (4)), i.e., the device used for $P_k$, as the $W_k$ sequence is conditionally deterministic given $P_{k+1}$ and $Q_k$, so we only need to estimate the SP of the device for $P_k$. Our key observation is that the update of $P_k$ in (11a) combines both descent on $f(\bar{W})$ via a stochastic-gradient increment, and an additional increment that drives $G_p(P_k)$ towards zero, scaled by a positive stepsize $|\nabla f(\bar{W}_k; \xi_k)|$. This suggests that the update of $P_k$ contains an inherent component that pulls it towards the SP. Therefore, we propose a moving averaging sequence $Q_k$ to magnify the SP-attracting property of $P_k$ sequence

$$Q_{k+1} = (1 - \eta)Q_k + \eta P_{k+1} \quad (12)$$

where the $Q_k$ sequence is updated on the digital device so that there is no analog update bias. The complete ResIdual learning with Dynamic symmEtric point tRacking (RIDER) algorithm is summarized in Algorithm 2. The following lemma formalizes the key insight that moving averaging amplifies the SP-attraction property.

**Lemma 3.5.** *Let $W^\diamond$ be the SP used for $P_k$, i.e. $G_p(W^\diamond) = 0$. For any iteration $k$, if $\cos(P_{k+1} - W^\diamond, P_{k+1} - Q_k) > 0$, then there exists $\eta \in (0, 1)$ in (12) such that*

$$\|Q_{k+1} - W^\diamond\|^2 < \|P_{k+1} - W^\diamond\|^2. \quad (13)$$

The proof of Lemma 3.5 is provided in Appendix D.1. Because the update of $P$ includes a gradient-descent term for the objective in addition to SP drift, so that both $W^\diamond$ and

---

**Algorithm 2** RIDER algorithm

1: **Inputs:** initialization $P_0, Q_0, W_0$; residual parameter $\gamma$; response granularity $F_p(\cdot), F_w(\cdot), G_p(\cdot), G_w(\cdot)$
2: **for** $k = 0, 1, \ldots, K - 1$ **do**
3:     evaluate $\bar{W}_k = W_k + \gamma(P_k - Q_k)$
4:     sample stochastic gradient $\nabla f(\bar{W}_k; \xi_k)$
5:     update $P_{k+1}$ on analog device via (11a)
6:     update $Q_{k+1}$ on digital device via (12)
7:     update $W_{k+1}$ on analog device via (11b)
8: **end for**
9: **outputs:** $\{P_K, Q_K, W_K\}$

---

$Q_k$ tend to sit on the same side from $P_{k+1}$. This suggests that the angle condition is likely to hold so that $Q_k$ sequence remains closer to $W^\diamond$ than $P_k$ sequence.

To analyze the convergence of Algorithm 2 with respect to three sequences, we define the convergence metric as

$$
E_K = \frac{1}{K} \sum_{k=0}^{K-1} \mathbb{E}\big[\|W_k - W^*\|^2 + \mathcal{O}\left(\|P_k - Q_k\|^2\right)
$$
$$
+ \mathcal{O}\left(\|G_p(P_k)\|^2\right)\big] \tag{14}
$$

where the three terms quantifies the convergence of $W_k, Q_k$ and $P_k$, respectively. For simplicity, the constants in front of some terms in $E_K$ are hidden. When $E_k \to 0$, it can be seen that $W_k \to W^*$ and $P_k, Q_k \to W^\diamond$. To prove the convergence, we need the following assumption.

**Assumption 3.6** (Rayleigh-type lower bound). *There exists $C_\star > 0$ such that for all $k$, $\mathbb{E}_{\xi_k}\big[|\nabla f(\bar{W}_k; \xi_k)|_d\big] \geq C_*$.*

Assumption 3.6 is mild since the operations in the analog domain intrinsically involve thermal and electrical noise.

**Theorem 3.7** (Convergence of Algorithm 2). *Suppose Assumptions 3.1-3.4, 3.6 hold and the response functions $F_p, G_p, F_w, G_w$ satisfy Definition 2.1. If $C_\star \geq \frac{4\sqrt{2}\sigma}{\mu}\left(\frac{q_{\max}}{q_{\min}}\right)^{\frac{3}{2}}$, let $\alpha = \Theta\left(\frac{1}{\sqrt{K}}\right)$, $\beta = \Theta(\alpha\gamma\mu)$, $\eta = \Theta(\alpha\mu)$, $\gamma = \Theta(1)$, it holds that*

$$
E_K \leq \mathcal{O}\left(\frac{\kappa_1 \kappa_2^5}{\sqrt{K}}\right) + \Theta(\Delta w_{\min}). \tag{15}
$$

*where $\kappa_1 := L/\mu$ and $\kappa_2 := q_{\max}/q_{\min}$ are the condition number of the function and device.*

The proof of Theorem 3.7 is provided in Appendix E. Theorem 3.7 shows that, Algorithm 2 can simultaneously track the SP and perform model training, and the minimal achievable training error $\|W_k - W^*\|^2 \leq \delta$ is $\delta = \Theta(\Delta w_{\min})$. Moreover, the convergence rate of Algorithm 2 matches that of Residual Learning (Wu et al., 2025), although we address a more challenging setting with nonzero and unknown SP.

---

**Algorithm 3** Enhanced version of RIDER (E-RIDER)

1: **Inputs:** initialization $P_0, \tilde{Q}_0, W_0$ on analog device and $Q_0 = \tilde{Q}_0$ on digital device; residual parameter $\gamma$; response granularity $F_p(\cdot), F_w(\cdot), G_p(\cdot), G_w(\cdot)$; chopper initialization $c_0 = 1$
2: **for** $k = 0, 1, \ldots, K - 1$ **do**
3:     draw the chopper variable $c_k$ via (17)
4:     **if** $\text{sign}(c_k) \neq \text{sign}(c_{k-1})$ **then**
5:         correct $\tilde{Q}_k = Q_k$ via weight programming
6:     **end if**
7:     sample stochastic gradient $\nabla f(\bar{W}_k; \xi_k)$
8:     update $P_{k+1}$ on analog device via (18a)
9:     update $Q_{k+1}$ on digital device via (12)
10:    update $W_{k+1}$ on analog device via (18b)
11: **end for**
12: **outputs:** $\{P_K, Q_K, W_K\}$

---

*Remark* 3.8. As a theoretical baseline, we consider Residual Learning (Wu et al., 2025) paired with SP estimation via the ZS algorithm, which we refer to as two-stage Residual Learning with ZS algorithm. This two-stage method first estimates a static SP $\hat{W}^\diamond$ with ZS and then fixes $Q_k \equiv \hat{W}^\diamond$ during training. We provide its pseudocode in Appendix B.1.

**Corollary 3.9** (Overall pulse complexity). *To achieve training accuracy $\delta \geq \Theta(\Delta w_{\min})$, RIDER in Algorithm 2 requires $\mathcal{O}(K) = \mathcal{O}(\delta^{-2})$ number of pulses, but the two-stage Residual Learning with ZS algorithm requires $\mathcal{O}(K + N) = \mathcal{O}(\delta^{-2} + \delta^{-1}\Delta w_{\min}^{-1})$ number of pulses.*

Corollary 3.9 directly follows from Theorem 3.7 and Theorem 2.2. This corollary suggests that when the target training error $\delta > \Theta(\Delta w_{\min})$, RIDER offers a clear benefit in pulse complexity. In practical high-precision AIMC devices, $\Delta w_{\min}$ is typically engineered to be as small as possible (e.g. $\Delta w_{\min} = 10^{-4}$) to preserve gradient-conversion fidelity (Rao et al., 2023; Sharma et al., 2024), whereas the acceptable training error $\delta$ can be substantially larger to enhance generalization. Consequently, RIDER is expected to require fewer pulses than two-stage approaches.

### 3.2. Enhancement via Chopping and Filtering

To enhance the empirical performance, we will design a variant of Algorithm 2 through chopping and filtering, inspired by (Rasch et al., 2023).

From the digital signal processing perspective, we can treat $P_k$ and $Q_k$ as two time-domain signals, and view the mapping from $P_k$ to $Q_k$ as a difference equation system (Proakis, 2007). Then the following lemma shows that this system can filter out the high-frequency signal components in $P_k$.

**Lemma 3.10.** *The moving average update (12) defines a stable low-pass filter from $P_k$ to $Q_k$ with the following*

*Table 1.* Test accuracy on LeNet-5 (MNIST) of different methods under different reference mean/std. Best results are highlighted in bold.

| Method | Mean⟍Std | 0.05 | 0.2 | 0.3 | 0.4 | 0.7 | 1.0 |
|---|---|---|---|---|---|---|---|
| TT-v2 | | $75.19_{\pm 1.0}$ | $72.62_{\pm 0.9}$ | $72.39_{\pm 1.1}$ | $71.50_{\pm 0.6}$ | $68.55_{\pm 0.9}$ | $66.96_{\pm 1.3}$ |
| AGAD | 0 | $90.81_{\pm 0.2}$ | $90.69_{\pm 1.1}$ | $91.02_{\pm 0.7}$ | $90.42_{\pm 0.7}$ | $90.15_{\pm 0.3}$ | $88.11_{\pm 0.3}$ |
| E-RIDER | | $\mathbf{93.75}_{\pm 0.1}$ | $\mathbf{93.71}_{\pm 0.2}$ | $\mathbf{94.15}_{\pm 0.6}$ | $\mathbf{93.26}_{\pm 1.3}$ | $\mathbf{91.67}_{\pm 0.3}$ | $\mathbf{89.02}_{\pm 0.3}$ |
| TT-v2 | | $75.01_{\pm 1.4}$ | $72.60_{\pm 2.1}$ | $71.68_{\pm 2.2}$ | $70.70_{\pm 1.1}$ | $66.54_{\pm 2.9}$ | $66.43_{\pm 1.3}$ |
| AGAD | 0.2 | $91.14_{\pm 0.8}$ | $90.66_{\pm 0.9}$ | $91.61_{\pm 0.1}$ | $90.61_{\pm 0.9}$ | $88.59_{\pm 0.6}$ | $87.73_{\pm 1.1}$ |
| E-RIDER | | $\mathbf{93.90}_{\pm 0.6}$ | $\mathbf{93.06}_{\pm 1.0}$ | $\mathbf{93.33}_{\pm 0.8}$ | $\mathbf{93.15}_{\pm 0.5}$ | $\mathbf{91.99}_{\pm 0.1}$ | $\mathbf{89.41}_{\pm 0.8}$ |
| TT-v2 | | $73.64_{\pm 0.5}$ | $72.50_{\pm 1.1}$ | $72.66_{\pm 0.3}$ | $70.70_{\pm 1.7}$ | $65.43_{\pm 2.5}$ | $66.78_{\pm 1.6}$ |
| AGAD | 0.3 | $90.86_{\pm 0.4}$ | $89.87_{\pm 0.4}$ | $90.37_{\pm 1.2}$ | $89.42_{\pm 0.8}$ | $89.43_{\pm 1.2}$ | $86.76_{\pm 0.4}$ |
| E-RIDER | | $\mathbf{92.45}_{\pm 1.9}$ | $\mathbf{92.15}_{\pm 1.5}$ | $\mathbf{92.27}_{\pm 0.6}$ | $\mathbf{91.45}_{\pm 1.1}$ | $\mathbf{89.74}_{\pm 0.7}$ | $\mathbf{90.26}_{\pm 1.2}$ |
| TT-v2 | | $71.71_{\pm 1.8}$ | $71.89_{\pm 3.3}$ | $70.83_{\pm 3.1}$ | $71.01_{\pm 2.8}$ | $66.63_{\pm 1.2}$ | $67.08_{\pm 1.6}$ |
| AGAD | 0.4 | $89.63_{\pm 1.3}$ | $89.79_{\pm 1.1}$ | $89.99_{\pm 1.9}$ | $90.16_{\pm 0.8}$ | $87.32_{\pm 1.2}$ | $86.54_{\pm 1.1}$ |
| E-RIDER | | $\mathbf{91.72}_{\pm 0.4}$ | $\mathbf{91.76}_{\pm 0.4}$ | $\mathbf{91.29}_{\pm 1.1}$ | $\mathbf{91.54}_{\pm 1.5}$ | $\mathbf{91.17}_{\pm 0.5}$ | $\mathbf{88.02}_{\pm 0.3}$ |

*magnitude of the frequency response*

$$|H(e^{j\omega})|^2 = \frac{\eta^2}{1 + (1-\eta)^2 - 2(1-\eta)\cos\omega}. \quad (16)$$

The proof of this lemma is provided in Appendix D.2. Observing that the magnitude response is maximized at zero frequency ($\omega = 0$) and minimized at the high frequency ($\omega = \pm\pi$), the moving average operator functions as a low-pass filter. Consequently, it effectively attenuates the high-frequency (sign-flipping) components of $P_k$.

This motivates us to use a chopper variable to further diversify the frequency for the two components in the $P_k$ update and keep the SP drifting part in the low-frequency band. Specifically, let us define the chopper variable $c_k$ that flips the sign with probability (w. p.) $p \in (0,1)$, i.e.

$$c_{k+1} = \begin{cases} c_k, & \text{w. p.} \quad 1-p, \\ -c_k, & \text{w. p.} \quad p. \end{cases} \quad (17)$$

With $\bar{W}_k = W_k + \gamma c_k(P_k - Q_k)$, the update (11) becomes

$$P_{k+1} = P_k - \alpha c_k \nabla f(\bar{W}_k; \xi_k) \odot F_p(P_k)$$
$$\quad - \alpha |c_k \nabla f(\bar{W}_k; \xi_k)| \odot G_p(P_k) + b_k \quad (18a)$$
$$W_{k+1} = W_k + \beta c_k(P_{k+1} - Q_k) \odot F_w(W_k)$$
$$\quad - \beta |c_k(P_{k+1} - Q_k)| \odot G_w(W_k) + b_k'. \quad (18b)$$

**Frequency domain interpretation.** In the frequency domain, $P_k$ comprises two distinct components: a high-frequency, sign-flipping term (corresponding to $c_k \nabla f(\bar{W}_k; \xi_k) \odot F_p(P_k)$) and a low-frequency, slowly varying term (corresponding to $|c_k \nabla f(\bar{W}_k; \xi_k)| \odot G_p(P_k)$). The crucial distinction is that the absolute value in the second term eliminates sign changes, causing it to accumulate rather than oscillating. Following Lemma 3.10, applying a moving average to $P_k$ acts as a low-pass filter, which results in the $Q_k$ sequence suppressing the sign-flipping component and retaining the low-frequency signals only; see Figure 3. As

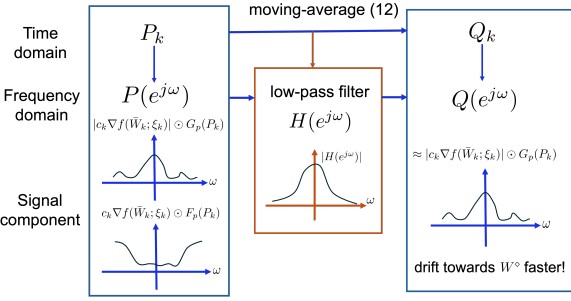

*Figure 3.* Chopping and filtering via moving average.

the low-frequency component of the $P_k$ update is a scaled increment proportional to $G_p(P_k)$, it forces the $Q_k$ drifting towards the SP of the $P$ device faster.

Furthermore, to stabilize the updates of $\bar{W}_k$, we incorporate the chopper $c_k$ directly into the $\bar{W}_k$ update rule, which is a novel modification compared to prior works (Wu et al., 2025; Rasch et al., 2023). This is because the update term $c_k(P_k - Q_k)$ is stable and (approximately) aligned in sign of $\nabla f(\bar{W}_k)$, which is independent of the flipping. Together, chopping and filtering help the $Q_k$ sequence converge to $W^\diamond$ faster while ensuring the objective descent for $\bar{W}_k$, yielding enhanced empirical performance.

**Practical implementation.** As $Q_k$ is stored on the digital device, but $P_k$ and $W_k$ are stored on the analog device, computing $P_k - Q_k$ and $P_{k+1} - Q_k$ requires frequent weight programming. To reduce this cost, we store a fake $\tilde{Q}_k$ on an additional analog device and only periodically correct it using the digitally stored $Q_k$ when the sign of $c_k$ flips. Overall, the weight programming cost of the E-RIDER is the same order as the existing dynamic SP tracking method AGAD (Rasch et al., 2023), and the difference of them is summarized in Appendix B.2. We summarize the Enhanced RIDER (E-RIDER) algorithm in Algorithm 3.

# 4. Experiments

We evaluate our method on MNIST dataset (LeCun et al., 1998b) using a fully analog LeNet-5 (LeCun et al., 1998a)

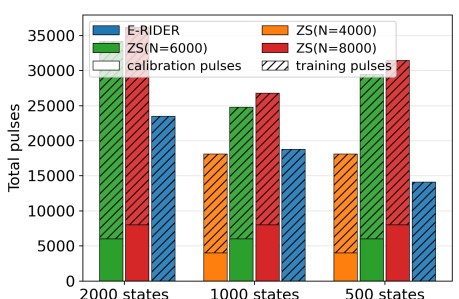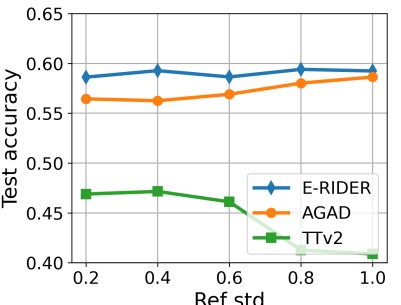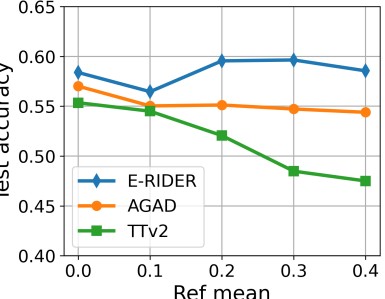

*Figure 4.* (Left) total pulse cost to reach the target training loss 0.2 on LeNet-5 (MNIST) across different number of states settings. Solid bars indicate the number of pulses using ZS algorithm, while hatched bars indicate the training cost computed as epochs $\times \lceil$ data size$/B \rceil \times$ BL, with batch size $B = 64$ and an average update pulse length BL $= 5$. For 2000 states, ZS ($N = 4000$) fails to reach the target loss. (Middle & right) training loss of E-RIDER and baselines under different reference std/mean on ResNet-18 (CIFAR-100) after 80 epochs.

and a fully analog fully connected network, and on CIFAR-100 dataset (Krizhevsky et al., 2009) by training a ResNet-18 model (He et al., 2016) with the fully connected layer and the last residual block implemented in analog. All experiments are implemented in the AIHWKit simulator (Rasch et al., 2021). Note that RIDER is a special case of E-RIDER with $p = 0$. As shown in the Figure 5, using a small $p > 0$ yields a clear improvement in test accuracy, so we use E-RIDER with the best-tuned $p$ in all subsequent experiments. For fairness, we also tune the chopper probability $p > 0$ for the dynamic SP tracking baseline AGAD (Rasch et al., 2023). Larger-scale experiments of analog ImageNet-1K fine-tuning (Deng et al., 2009), hyperparameters choices for different methods and ablation studies for key hyperparameters for E-RIDER are provided in Appendix F.3–F.6.

**Device model.** We use two ReRAM array device presets from IBM AIHWKit simulator (Rasch et al., 2021): the HfO$_2$-based RRAM preset and the ReRamArrayOM preset in (Gong et al., 2022b). Both presets inherit from the `SoftBoundsReferenceDevice` model, where the ReRAM update response is modeled by state-dependent linear soft-bound functions. Let $-\tau_{\min}$ and $\tau_{\max}$ denote the lower and upper weight bounds, respectively, with $\tau_{\min}, \tau_{\max} > 0$. For a scalar weight $w \in [-\tau_{\min}, \tau_{\max}]$, we write the normalized potentiation and depression response functions as

$$q_+(w) = \alpha_+ \left( 1 - \frac{w}{\tau_{\max}} \right), \quad q_-(w) = \alpha_- \left( 1 + \frac{w}{\tau_{\min}} \right),$$

where $\alpha_+, \alpha_- > 0$ determine the effective update magnitudes for potentiation and depression, respectively. The specific device parameters are provided in Appendix F.1.

**E-RIDER achieves better overall pulse complexity.** We conduct an ablation study on LeNet-5 (MNIST) with a convergence criterion of training loss $\leq 0.2$. Figure 4 (left) reports the total pulse cost required to reach this target for E-RIDER and the two-stage TT-v2 (Gokmen, 2021) with ZS algorithm under different numbers of device states. For

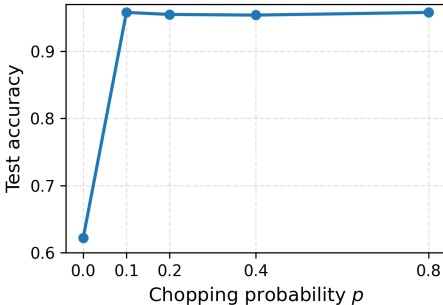

*Figure 5.* Test accuracy of E-RIDER on MNIST-FCN after 50 epochs under different chopper probabilities $p$.

the two-stage ZS approach, the total pulse cost is the sum of $N$ pulses for SP estimation and the subsequent training pulses, determined by the number of training epochs $K$ when training with TT-v2 after calibration. When the number of states is low ($\Delta w_{\min}$ is large), SP estimation with fewer pulses (e.g., $N = 4000$) leads to better convergence than using larger $N$, which consists with Theorem 2.2. However, as the number of states increases (smaller $\Delta w_{\min}$), the two-stage ZS approach becomes more expensive because accurate SP estimation requires a larger pulse budget; for 2000 states, ZS with $N = 4000$ pulses fails to meet the training target loss. Meanwhile, E-RIDER consistently achieves a lower total pulse cost across all settings.

**E-RIDER is robust to nonzero SP.** We conduct an ablation study on the robustness of E-RIDER to a nonzero SP reference. The device model follows the RRAM-RfO$_2$ preset (Gong et al., 2022a) to emulate practical update non-idealities in filamentary RRAM. We focus on a limited-state scenario where the number of states is $\sim$4–5. It also mimics the strong device-to-device mismatch and cycle-to-cycle writing noise specified in Appendix F.1. We initialize $W^{\diamond}$ by sampling each entry $W_{ij}$ i.i.d. from a Gaussian distribution with varying reference mean (Ref Mean) and standard deviation (Ref Std) to model different nonzero SP scenarios. We compare E-RIDER with both TT-v2 and AGAD, summarizing the test accuracy after 40 epochs in Tables 1

*Table 2.* Test accuracy on FCN (MNIST) for different algorithms under different reference mean/std. Best results are highlighted in bold.

| Method | Mean / Std | 0.05 | 0.2 | 0.3 | 0.4 | 0.7 | 1.0 |
|---|---|---|---|---|---|---|---|
| TT-v2 | | $90.26_{\pm0.4}$ | $88.75_{\pm0.6}$ | $87.40_{\pm0.7}$ | $85.98_{\pm0.2}$ | $82.85_{\pm2.4}$ | $79.20_{\pm3.2}$ |
| AGAD | 0 | $92.59_{\pm0.2}$ | $91.96_{\pm0.8}$ | $92.11_{\pm0.9}$ | $92.48_{\pm0.1}$ | $92.19_{\pm0.4}$ | $91.37_{\pm0.4}$ |
| E-RIDER | | $\mathbf{95.45}_{\pm0.2}$ | $\mathbf{95.48}_{\pm0.1}$ | $\mathbf{95.39}_{\pm0.1}$ | $\mathbf{95.34}_{\pm0.4}$ | $\mathbf{95.42}_{\pm0.3}$ | $\mathbf{93.86}_{\pm0.2}$ |
| TT-v2 | | $73.37_{\pm0.3}$ | $73.99_{\pm0.7}$ | $71.88_{\pm0.4}$ | $70.40_{\pm0.8}$ | $68.63_{\pm1.1}$ | $62.35_{\pm0.6}$ |
| AGAD | 0.2 | $92.86_{\pm1.3}$ | $91.95_{\pm0.4}$ | $92.63_{\pm0.7}$ | $92.80_{\pm0.2}$ | $92.22_{\pm0.3}$ | $90.77_{\pm0.3}$ |
| E-RIDER | | $\mathbf{95.78}_{\pm0.1}$ | $\mathbf{95.63}_{\pm0.1}$ | $\mathbf{95.84}_{\pm0.3}$ | $\mathbf{95.82}_{\pm0.3}$ | $\mathbf{95.82}_{\pm0.3}$ | $\mathbf{95.18}_{\pm0.3}$ |
| TT-v2 | | $72.86_{\pm1.3}$ | $72.65_{\pm0.5}$ | $69.70_{\pm1.6}$ | $69.96_{\pm1.5}$ | $66.44_{\pm2.0}$ | $61.45_{\pm1.0}$ |
| AGAD | 0.3 | $92.31_{\pm0.4}$ | $92.52_{\pm0.4}$ | $92.47_{\pm0.5}$ | $92.88_{\pm0.3}$ | $91.68_{\pm0.5}$ | $91.35_{\pm0.4}$ |
| E-RIDER | | $\mathbf{95.76}_{\pm0.1}$ | $\mathbf{95.81}_{\pm0.5}$ | $\mathbf{95.91}_{\pm0.6}$ | $\mathbf{95.88}_{\pm0.1}$ | $\mathbf{95.79}_{\pm0.2}$ | $\mathbf{94.84}_{\pm0.2}$ |
| TT-v2 | | $72.00_{\pm2.8}$ | $68.08_{\pm2.1}$ | $68.80_{\pm1.9}$ | $67.85_{\pm1.4}$ | $66.15_{\pm0.9}$ | $57.79_{\pm0.8}$ |
| AGAD | 0.4 | $92.55_{\pm0.2}$ | $92.42_{\pm0.6}$ | $92.42_{\pm0.8}$ | $92.06_{\pm0.4}$ | $90.62_{\pm0.3}$ | $90.43_{\pm0.3}$ |
| E-RIDER | | $\mathbf{96.07}_{\pm0.2}$ | $\mathbf{96.21}_{\pm0.1}$ | $\mathbf{96.20}_{\pm0.1}$ | $\mathbf{96.23}_{\pm0.3}$ | $\mathbf{95.87}_{\pm0.3}$ | $\mathbf{95.14}_{\pm0.3}$ |

and 2. Each setting is repeated three times, and we report the sample mean and standard deviation results; the hyperparameters are both tuned to optimum (See Appendix F.3 for details). The results demonstrate that: **1)** TT-v2 exhibits a substantial accuracy gap to both AGAD and E-RIDER, and its performance degrades markedly as the reference mean/std offsets increase. This is expected because TT-v2 cannot compensate for nonzero reference offsets. **2)** E-RIDER consistently outperforms TT-v2 and AGAD in all reference mean/std offset settings, especially with larger SP offset. This is because E-RIDER dynamically tracks SP, and further improves training under low-state devices, which increases the effective weight resolution and enables more accurate gradient updates.

We further evaluate the robustness of our method on CIFAR-100 using a ResNet-18 trained for 80 epochs. We first fix the reference mean to 0.4 and sweep the reference standard deviation, and then fix the reference standard deviation to 0.4 and sweep the reference mean. The results are summarized in Figure 4 (middle & right). We observe similar trends: **1)** TT-v2 suffers from a significant degrade in testing accuracy as the reference standard deviation and mean grows, whereas AGAD and E-RIDER remain more stable. **2)** E-RIDER consistently achieves higher testing accuracy than both TT-v2 and AGAD, especially for large SP offset.

We additionally provide large-scale ImageNet-1K (Deng et al., 2009) fine-tuning experiments with VGG-11-BN (Simonyan & Zisserman, 2014), a 132M-parameter model with 8 convolutional and 3 fully connected layers, for two dynamic SP-tracking methods, AGAD and E-RIDER, in Appendix F.5. The results further demonstrate the superiority of E-RIDER over AGAD in the large-scale setting.

## 5. Conclusions

In this paper, we characterize the pulse complexity of ZS algorithm and quantify how the required number of pulse updates for achieving a certain target SP estimation error

scales with device response granularity $\Delta w_{\min}$. Building on these insights, we propose a dynamic SP tracking algorithm, RIDER, which estimates the SP during training, achieving target training accuracy with substantially fewer pulse updates than the two-stage analog training approaches theoretically. We further develop an variant of RIDER, E-RIDER, that accelerates SP tracking through chopper-and-filtering, while reducing weight programming overhead by periodical synchronization. Numerical experiments validate the efficiency and effectiveness of the proposed E-RIDER.

## Acknowledgment

The work was supported by the National Science Foundation Projects 2401297 and 2532349, by NVIDIA Academic Grant, by IBM through the IBM-Rensselaer Future of Computing Research Collaboration, and by Cisco Research.

## Impact Statement

This paper aims to advance AIMC training via dynamic SP tracking, providing a principled algorithmic framework to mitigate update asymmetry and device drift during learning. By addressing the key challenges in dynamic SP estimation and calibration, our work contributes to the broader development of more robust and hardware-aware training methods for energy-efficient AIMC accelerators. Potential societal impacts include enabling lower-power AI training and adaptation at the edge, reducing the energy and carbon footprint of AI workloads. While we acknowledge the possibility of unintended uses, we do not identify any specific societal risks that need to be highlighted in this context.

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

# Appendix for "Dynamic Symmetric Point Tracking: Tackling Non-ideal Reference in Analog In-memory Training"

## Table of Contents

## A. Notations and Preliminaries

In this section, we define a series of notations that will be used in the analysis.

**Pseudo-inverse of diagonal matrix or vector.** For a given diagonal matrix $U \in \mathbb{R}^{D \times D}$ with its $d$-th diagonal element $[U]_d$, we define the pseudo-inverse of a diagonal matrix $U$ as $U^\dagger$, which is also a diagonal matrix with its $d$-th diagonal element

$$[U^\dagger]_d := \begin{cases} 1/[U]_d, & [U]_d \neq 0, \\ 0, & [U]_d = 0. \end{cases} \tag{19}$$

By definition, the pseudo-inverse satisfies $UU^\dagger V = U^\dagger U V$ for any diagonal matrix $U \in \mathbb{R}^D$ and any matrix $V \in \mathbb{R}^D$. With a slight abuse of notation, we also define the pseudo-inverse of a vector $W \in \mathbb{R}^D$ as $W^\dagger := \text{diag}(W)^\dagger$.

---

**Algorithm 4** Two-stage analog training algorithm

---

1: **Inputs:** initialization $P_0, Q_0, W_0$; residual parameter $\gamma$; response granularity $F_p(\cdot), F_w(\cdot), G_p(\cdot), G_w(\cdot)$
2: estimate $\hat{W}^\diamond$ by Algorithm 1 with $N$ number of pulses
3: **for** $k = 0, 1, \ldots, K - 1$ **do**
4:     evaluate $\bar{W}_k = W_k + \gamma(P_k - Q_k)$
5:     sample stochastic gradient $\nabla f(\bar{W}_k; \xi_k)$
6:     update $P_{k+1}$ on analog device via (11a)
7:     set $Q_{k+1} = \hat{W}^\diamond$
8:     update $W_{k+1}$ on analog device via (11b)
9: **end for**
10: **outputs:** $\{P_K, Q_K, W_K\}$

---

**Weighted norm.** For a weight $M \in \mathbb{R}_+^D$, the weighted norm $\| \cdot \|_M$ of $W \in \mathbb{R}^D$ is defined by

$$\|W\|_M := \sqrt{\sum_{d=1}^{D} [M]_d [W]_d^2} = \sqrt{W^\top \text{Diag}(M) W} \tag{20}$$

where $\text{Diag}(M) \in \mathbb{R}^{D \times D}$ rearranges the vector $M \in \mathbb{R}^D$ into a diagonal matrix.

**Lemma A.1** ((Wu et al., 2025, Lemma 1)). $\|W\|_M$ *has the following properties: (a)* $\|W\|_M = \|W \odot \sqrt{M}\|$; *(b)* $\|W\|_M \leq \|W\| \sqrt{\|M\|_\infty}$; *(c)* $\|W\|_M \geq \|W\| \sqrt{\min\{[M]_d : k \in [K], d \in [D]\}}$.

**Lemma A.2** ((Wu et al., 2025, Lemma 2)). *Consider response functions in Definition 2.1, the increment defined in* (2) *is Lipschitz continuous with respect to* $\Delta W$ *under any weighted norm* $\| \cdot \|_M$, *i.e., for any* $W, \Delta W, \Delta W' \in \mathbb{R}^D$ *and* $M \in \mathbb{R}_+^D$, *it holds*

$$\|\Delta W \odot F(W) - |\Delta W| \odot G(W) - (\Delta W' \odot F(W) - |\Delta W'| \odot G(W))\|_M \leq q_{\max} \|\Delta W - \Delta W'\|_M.$$

## B. Discussion of Algorithms

In this section, we discuss some implementation details of algorithms.

### B.1. Two-stage analog training approach: Residual Learning with ZS algorithm

The two-stage analog training algorithm consists of an independent SP estimation stage that uses Algorithm 1 to obtain a static SP estimate $\hat{W}^\diamond$, and an independent training stage that applies residual learning with $\hat{W}^\diamond$ fixed; i.e. we remove the $Q_k$-sequence update in Algorithm 2 and set $Q_k \equiv \hat{W}^\diamond$. We give a pseudo-code for this algorithm in Algorithm 4.

### B.2. Comparison of E-RIDER, Residual Learning/TT-v2 and AGAD

The proposed E-RIDER has a similar form of AGAD (Rasch et al., 2023). However, AGAD uses the gradient $\nabla f(W_k; \xi_k)$ that are solely computed on the main array $W_k$. Instead, E-RIDER computes gradient on a mixed weight $\bar{W}_k = W_k + \gamma c_k(P_k - Q_k)$ so that achieves better performance in simulation. The key reason is that introducing the residual term $\gamma c_k(P_k - Q_k)$ with a nonzero $\gamma$ effectively rescales the update, yielding a finer effective dynamic range and granularity than the raw device response. Moreover, this zero-shifting vector mitigates update asymmetry through the bilevel optimization. In simulation, we show that E-RIDER which uses the gradient evaluated at the mixed weight improves test accuracy, especially when the SP is substantially nonzero.

Compared with Residual Learning (Wu et al., 2025), E-RIDER uses a moving average sequence to track the SP and adds a chopper mechanism to amplify the SP tracking performance. From a theoretical perspective, Residual Learning fails to converge when the SP is nonzero, whereas E-RIDER with $p = 0$ is guaranteed to converge in our paper. Empirically, Residual Learning performs similarly to TT-v2, and both suffer substantial performance degradation under nonzero SP.

## B.3. Failure of dynamic zero-shifting algorithm design.

From an optimization perspective, to balance the pulse overhead of SP estimation and model training, a natural approach is to integrate Algorithm 1 into analog training algorithms via multi-sequence updates (Yang et al., 2019; Shen & Chen, 2022), enabling dynamic SP estimation during the model training process. However, this approach is infeasible in analog training because each update sequence is executed on a different device, and the SP is device-specific. As a result, the SP tracked by the zero-shifting Algorithm 1 cannot be directly transferred to the sequence used for model training.

# C. Proof of Convergence Rate of Algorithm 1 and Cyclic Version

## C.1. Proof of Theorem 2.2: Convergence rate of Algorithm 1

**Theorem 2.2** (Convergence rate of Algorithm 1). *Considering the response functions in Definition 2.1, then the iterates given by Algorithm 1 with N pulses satisfy*

$$\frac{1}{N}\sum_{n=0}^{N-1}\mathbb{E}\left[\|G(W_n)\|^2\right] \leq \mathcal{O}\left(\frac{1}{N\Delta w_{\min}}\right) + \Theta(\Delta w_{\min}).$$

*Theorem 2.2.* Define the following notations:

$$\varphi(W) := \int_{W^\diamond}^{W} G(W')dW', \qquad \nabla\varphi(W) := G(W), \qquad L_q := \max_{W}\Delta G(W). \tag{21}$$

Under gradient boundedness implied by Definition 2.1, it holds that:

$$\varphi(W_{n+1}) \leq \varphi(W_n) + \langle \nabla\varphi(W_n),\, W_{n+1} - W_n \rangle + \frac{L_q}{2}\|W_{n+1} - W_n\|^2$$

$$= \varphi(W_n) + \langle G(W_n),\, \varepsilon_n F(W_n) - |\varepsilon_n|G(W_n)\rangle + \frac{L_q}{2}\|\varepsilon_n F(W_n) - |\varepsilon_n|G(W_n)\|^2. \tag{22}$$

where $L_q$ is the gradient boundedness constant implied by Definition 2.1. The equality holds by substituting the stochastic updating equation (7). Taking expectation over $\varepsilon_n$ on both sides, we get:

$$\mathbb{E}[\varphi(W_{n+1}) \mid W_n] \leq \varphi(W_n) - \mathbb{E}[|\varepsilon_n|]\|G(W_n)\|^2 + \frac{L_q}{2}\Delta w_{\min}^2 q_{\max}^2$$

$$= \varphi(W_n) - \Delta w_{\min}\|G(W_n)\|^2 + \frac{L_q}{2}\Delta w_{\min}^2 q_{\max}^2$$

$$\leq \varphi(W_n) - \Delta w_{\min}\|G(W_n)\|^2 + \frac{L_q}{2}\Delta w_{\min}^2 q_{\max}^2. \tag{23}$$

Taking total expectation $\mathcal{F}_n$ to both sides of (23) and using $\mathbb{E}\left[\mathbb{E}\left[\varphi(W_{n+1}) \mid \mathcal{F}_n\right]\right] = \mathbb{E}\left[\varphi(W_{n+1})\right]$, we get:

$$\mathbb{E}[\varphi(W_{n+1})] \leq \mathbb{E}\left[\varphi(W_n)\right] - \Delta w_{\min}\mathbb{E}\left[\|G(W_n)\|^2\right] + \frac{L_q}{2}\Delta w_{\min}^2 q_{\max}^2. \tag{24}$$

Taking average over all $n$ form 0 to $N-1$, we get:

$$\frac{1}{N}\sum_{n=0}^{N-1}\mathbb{E}\left[\|G(W_n)\|^2\right] \leq \frac{\varphi(W_0) - \varphi(W^*)}{N\Delta w_{\min}} + \frac{L_q}{2}\Delta w_{\min}q_{\max}^2 \tag{25}$$

which completes the proof. $\qquad\square$

## C.2. Proof of Theorem C.2: Last-iterate convergence of Algorithm 1

To get the enhanced last-iterate pulse complexity for Algorithm 1, we focus on the family of devices with the following monotone response functions.

**Definition C.1** (Monotone response functions). Response functions $q_+(\cdot)$ and $q_-(\cdot)$ are said to be monotone with modulus $\mu_q > 0$ if

$$\nabla q_-(w) \geq \mu_q, \quad \text{and} \quad \nabla q_+(w) \leq -\mu_q. \tag{26}$$

Definition C.1 suggests that $q_-(\cdot)$ is monotonically increasing and $q_+(\cdot)$ is monotonically decreasing. The response functions of a wide range of AIMC devices, such as linear, exponential, and power device, satisfy Definition C.1 (Wu et al., 2025). With Definition C.1, $G(W)$ is strongly monotone with $\mu_q$ so that we have the following last-iterates convergence.

**Theorem C.2** (Last-iterate convergence of Algorithm 1). *Considering the response functions satisfying Definition 2.1 and C.1 and assuming $\mu_g < \frac{1}{2\Delta w_{\min}}$, then to achieve $\mathbb{E}[\|W^N - W^\diamond\|^2] \leq \delta$, we need*

$$N \leq \frac{1}{2\mu_q \Delta w_{\min}} \log\left(\frac{2\|W^0 - W^\diamond\|^2}{\delta}\right).$$

*and the minimal achievable error $\delta$ is $\delta = \frac{2q_{\max}^2 \Delta w_{\min}}{\mu_q}$.*

From Theorem C.2, we know that the zero-shifting Algorithm 1 achieves a sublinear dependence on the target error $\delta$, and that the required number of pulses scales linearly with $1/\Delta w_{\min}$. In practice, the response granularity $\Delta w_{\min}$ of AIMC device is made sufficiently small to ensure the gradient conversion precision (Rao et al., 2023; Sharma et al., 2024), so that the computational complexity of Algorithm 1 is relatively high.

*Theorem C.2.* We begin the proof by one step descent form stochastic updating equation (7) and taking expectation over $\varepsilon_n$ on both sides:

$$\mathbb{E}\left[\|W_{n+1} - W^\diamond\|^2 \mid W_n\right]$$
$$= \mathbb{E}\left[\|W_n - \Delta w_{\min}\varepsilon_n F(W_n) - \Delta w_{\min}G(W_n) - W^\diamond\|^2 \mid W_n\right]$$
$$= \|W_n - W^\diamond\|^2 - 2\Delta w_{\min}\langle W_n - W^\diamond, G(W_n) - G_w(W^\diamond)\rangle + \Delta w_{\min}^2 \mathbb{E}\left[\|\varepsilon_n F(W_n) + G(W_n)\|^2 \mid W_n\right]$$
$$\leq (1 - 2\Delta w_{\min}\mu_q)\|W_n - W^\diamond\|^2 + 2\Delta w_{\min}^2 q_{\max}^2. \tag{27}$$

The second equality holds for $\mathbb{E}[\varepsilon_n F(W_n)] = 0$ and $G_w(W^\diamond) = 0$. The inequality holds for the strongly convex assumptions in definition C.1 and bounded response functions in definition 2.1:

$$\langle W_n - W^\diamond, G(W_n) - G_w(W^\diamond)\rangle \geq \mu_q\|W_n - W^\diamond\|^2, \qquad \mathbb{E}\left[\|\varepsilon_n F(W_n) + G(W_n)\|^2 \mid W_n\right] \leq 2q_{\max}^2.$$

Taking total expectation $\mathcal{F}_n$ to both sides of (27) and iterating this recursion for $n = 0, \ldots, N-1$ gives:

$$\mathbb{E}\left[\|W_N - W^\diamond\|^2\right] \leq (1 - 2\Delta w_{\min}\mu_q)^N \|W_0 - W^\diamond\|^2 + 2\Delta w_{\min}^2 q_{\max}^2 \sum_{n=0}^{N-1}(1 - 2\Delta w_{\min}\mu_q)^n$$
$$\leq (1 - 2\Delta w_{\min}\mu_q)^N \|W_0 - W^\diamond\|^2 + \frac{\Delta w_{\min}q_{\max}^2}{\mu_q}. \tag{28}$$

To ensure that the error bound is below a given tolerance $\delta$, we split the upper bound into two terms and require each of them to be at most $\frac{\delta}{2}$, which means $\frac{\Delta w_{\min}q_{\max}^2}{\mu_q} \leq \frac{\delta}{2}$ and $(1 - 2\mu_q\Delta w_{\min})^N \|W_0 - W^\diamond\|^2 \leq \frac{\delta}{2}$.

The first inequality holds when $\Delta w_{\min} \leq \frac{\delta\mu_q}{2q_{\max}^2}$. Taking logarithms on both sides of the second inequality gives:

$$\log\left(\frac{2\|W_0 - W^\diamond\|^2}{\delta}\right) \leq N \log\left(\frac{1}{1 - 2\mu_q\Delta w_{\min}}\right). \tag{29}$$

Using the inequality $\log(\frac{1}{p}) \geq 1 - p$, when $\mu_q < \frac{1}{2\Delta w_{\min}}$, it suffices to require

$$N \geq \frac{1}{2\mu_q\Delta w_{\min}}\log\left(\frac{2\|W_0 - W^\diamond\|^2}{\delta}\right) \tag{30}$$

which completes the proof. $\square$

## C.3. Proof of Theorem C.3: Convergence rate of Algorithm 1, cyclic version

In this section, we model the empirical implementation of the zero-shifting technique, where cyclic alternating pulses are applied. We also establish a convergence guarantee analogous to Theorem 2.2. We first show the update dynamics below:

$$
\begin{aligned}
W_{2n} &= W_{2n-1} - \Delta w_{\min} F(W_{2n-1}) - \Delta w_{\min} G(W_{2n-1}), \\
W_{2n+1} &= W_{2n} + \Delta w_{\min} F(W_{2n}) - \Delta w_{\min} G(W_{2n}).
\end{aligned}
\tag{31}
$$

**Theorem C.3** (Convergence rate of Algorithm 1, cyclic version). *Considering the response functions satisfying Definition 2.1, then when $N = \mathcal{O}\left(\Delta w_{\min}^{-2}\kappa_2^{-2}\right)$, it holds that*

$$
\frac{1}{2N}\sum_{n=0}^{2N-1}\|G(W_n)\|^2 \leq \frac{\varphi(W_0)-\varphi^*}{2N\,\Delta w_{\min}} + \frac{3\,\Delta w_{\min}\,q_{\max}^2\,L_q}{2}
\tag{32}
$$

*where $L_q$ is the gradient boundedness constant implied by Definition 2.1. Therefore, the minimal achievable SP estimation error $\frac{1}{2N}\sum_{n=0}^{2N-1}\mathbb{E}\left[\|G(W_n)\|^2\right] \leq \delta$ is $\delta = \mathcal{O}(\Delta w_{\min})$. To achieve error $\delta \geq \mathcal{O}(\Delta w_{\min})$, it requires $N = \mathcal{O}(\frac{1}{\delta\Delta w_{\min}})$ number of pulses.*

*Theorem C.3.* We begin the proof by using the same notations defined in equation (21). Under gradient boundedness implied by Definition 2.1, it holds that:

$$
\begin{aligned}
\varphi(W_{n+1}) &\leq \varphi(W_n) + \langle \nabla\varphi(W_n),\, W_{n+1}-W_n\rangle + \frac{L_q}{2}\|W_{n+1}-W_n\|^2 \\
&= \varphi(W_n) + \Delta w_{\min}\langle G(W_n),\, F(W_n)-G(W_n)\rangle + \frac{L_q}{2}\Delta w_{\min}^2\|F(W_n)-G(W_n)\|^2 \\
&\leq \varphi(W_n) - \Delta w_{\min}\|G(W_n)\|^2 + \Delta w_{\min}\langle G(W_n),\, F(W_n)\rangle + \frac{L_q}{2}\Delta w_{\min}^2 q_{\max}^2.
\end{aligned}
\tag{33}
$$

The second equality holds by substituting the stochastic updating equation (31). The inequality holds for $F(W_n)-G(W_n) = \frac{q_+(W_n)+q_-(W_n)}{2} - \frac{q_+(W_n)-q_-(W_n)}{2} = q_+(W_n) \leq q_{\max}$. Similarly we have:

$$
\begin{aligned}
\varphi(W_{n+2}) &\leq \varphi(W_{n+1}) + \Delta w_{\min}\langle G(W_{n+1}),\, -F(W_{n+1})-G(W_{n+1})\rangle + \frac{L_q}{2}\Delta w_{\min}^2 q_{\max}^2 \\
&= \varphi(W_{n+1}) - \Delta w_{\min}\|G(W_{n+1})\|^2 - \Delta w_{\min}\langle G(W_{n+1}),\, F(W_{n+1})\rangle + \frac{L_q}{2}\Delta w_{\min}^2 q_{\max}^2.
\end{aligned}
\tag{34}
$$

Combining (33) and (34), we get:

$$
\begin{aligned}
\varphi(W_{n+2}) &\leq \varphi(W_n) - \Delta w_{\min}\|G(W_n)\|^2 - \Delta w_{\min}\|G(W_{n+1})\|^2 \\
&\quad + \Delta w_{\min}\Big(\langle G(W_n),\, F(W_n)\rangle - \langle G(W_{n+1}),\, F(W_{n+1})\rangle\Big) + L_q\,\Delta w_{\min}^2 q_{\max}^2 \\
&\leq \varphi(W_n) - \Delta w_{\min}\|G(W_n)\|^2 - \Delta w_{\min}\|G(W_{n+1})\|^2 + 3\,\Delta w_{\min}^2\,q_{\max}^2\,L_q.
\end{aligned}
\tag{35}
$$

The second inequality holds for

$$
\begin{aligned}
&\langle G(W_n),\, F(W_n)\rangle - \langle G(W_{n+1}),\, F(W_{n+1})\rangle \\
&= \langle G(W_n),\, F(W_n)-F(W_{n+1})\rangle - \langle G(W_{n+1})-G(W_n),\, F(W_{n+1})\rangle \\
&\leq \|G(W_n)\|\,\|F(W_n)-F(W_{n+1})\| + \|G(W_{n+1})-G(W_n)\|\,\|F(W_{n+1})\| \\
&\leq 2q_{\max}L_q\,\|W_{n+1}-W_n\| \\
&\leq 2q_{\max}L_q\,\Delta w_{\min}\,q_{\max}.
\end{aligned}
\tag{36}
$$

By summing the recursion in Eq. (35) over the iterations, we obtain:

$$
\varphi(W_{2N}) \leq \varphi(W_0) - \sum_{n=0}^{2N-1}\Delta w_{\min}\|G(W_n)\|^2 + 3\,\Delta w_{\min}^2\,q_{\max}^2\,L_q N.
\tag{37}
$$

Taking average, we get:

$$\frac{1}{2N} \sum_{n=0}^{2N-1} \|G(W_n)\|^2 \leq \frac{\varphi(W_0) - \varphi^*}{2N \Delta w_{\min}} + \frac{3 \Delta w_{\min} q_{\max}^2 L_q}{2}. \tag{38}$$

$\square$

## C.4. Proof of Theorem C.4: Last-iterate convergence of Algorithm 1, cyclic version

**Theorem C.4** (Last-iterate convergence of Algorithm 1, cyclic version). *Considering the response functions satisfying Definition 2.1 and C.1 and assuming $\mu_q < \frac{1}{3\mu_q \Delta w_{\min}}$, then it holds that*

$$(1 - 2\Delta w_{\min}\mu_q)^{2N} \|W_0 - W^\diamond\|^2 + \frac{\Delta w_{\min} q_{\max}^2 \left(4 + \frac{L_q^2}{\mu_q} + 2\mu_q\right)}{\mu_q}.$$

*Moreover, the minimal achievable error $\|W^{2N} - W^\diamond\|^2 \leq \delta$ is $\delta = \mathcal{O}(\Delta w_{\min})$, and to achieve $\|W^{2N} - W^\diamond\|^2 \leq \delta$ with $\delta \geq \mathcal{O}(\Delta w_{\min})$, one need the number of pulses satisfying*

$$N \geq \frac{1}{4\mu_q \Delta w_{\min}} \log \left( \frac{2\|W^0 - W^\diamond\|^2}{\delta} \right).$$

*Theorem C.4.* We begin the proof by one step descent form cyclic updating equation (31)

$$\begin{aligned}
\|W_{n+1} - W^\diamond\|^2 &= \|W_n - \Delta w_{\min}F(W_n) - \Delta w_{\min}G(W_n) - W^\diamond\|^2 \\
&= \|W_n - W^\diamond\|^2 - 2\Delta w_{\min}\langle W_n - W^\diamond, G(W_n) - G(W^\diamond)\rangle + \Delta w_{\min}^2 \|F(W_n) + G(W_n)\|^2 \\
&\quad - 2\Delta w_{\min}\langle W_n - W^\diamond, F(W_n)\rangle \\
&\leq (1 - 2\Delta w_{\min}\mu_q)\|W_n - W^\diamond\|^2 + 2\Delta w_{\min}^2 q_{\max}^2 - 2\Delta w_{\min}\langle W_n - W^\diamond, F(W_n)\rangle. \tag{39}
\end{aligned}$$

The second equality holds for $G_w(W^\diamond) = 0$. The inequality holds for the strongly convex assumptions in definition C.1 and bounded response functions in definition 2.1:

$$\langle W_n - W^\diamond, G(W_n) - G_w(W^\diamond)\rangle \geq \mu_q \|W_n - W^\diamond\|^2, \qquad \|F(W_n) + G(W_n)\|^2 \leq 2q_{\max}^2.$$

Similarly we have:

$$\begin{aligned}
\|W_{n+2} - W^\diamond\|^2 &= \|W_{n+1} + \Delta w_{\min}F(W_{n+1}) - \Delta w_{\min}G(W_{n+1}) - W^\diamond\|^2 \\
&\leq (1 - 2\Delta w_{\min}\mu_q)\|W_{n+1} - W^\diamond\|^2 + 2\Delta w_{\min}^2 q_{\max}^2 + 2\Delta w_{\min}\langle W_{n+1} - W^\diamond, F(W_{n+1})\rangle. \tag{40}
\end{aligned}$$

The last term in (39) and (40) can be bounded by:

$$\begin{aligned}
&2\Delta w_{\min}\langle W_n - W^\diamond, F(W_{n+1})\rangle - 2\Delta w_{\min}\langle W_n - W^\diamond, F(W_n)\rangle (1 - 2\Delta w_{\min}\mu_q) \\
&\leq 2\Delta w_{\min}\langle W_n - W^\diamond, F(W_{n+1}) - F(W_n)\rangle + 4\Delta w_{\min}^2 \mu_q \|W_n - W^\diamond\| \|F(W_n)\| \\
&\leq \Delta w_{\min}^2 \mu_q \|W_n - W^\diamond\|^2 + \frac{\|F(W_{n+1}) - F(W_n)\|^2}{\mu_q} + 4\Delta w_{\min}^2 \mu_q \|W_n - W^\diamond\| q_{\max} \\
&\leq \Delta w_{\min}^2 \mu_q \|W_n - W^\diamond\|^2 + \frac{L_q^2 \Delta w_{\min}^2 q_{\max}^2}{\mu_q} + 2\Delta w_{\min}^2 \mu_q \|W_n - W^\diamond\|^2 + 2\Delta w_{\min}^2 \mu_q q_{\max}^2 \\
&\leq 3\Delta w_{\min}^2 \mu_q \|W_n - W^\diamond\|^2 + \frac{L_q^2 \Delta w_{\min}^2 q_{\max}^2}{\mu_q} + 2\Delta w_{\min}^2 \mu_q q_{\max}^2 \\
&\leq \Delta w_{\min}\mu_q \|W_n - W^\diamond\|^2 + \frac{L_q^2 \Delta w_{\min}^2 q_{\max}^2}{\mu_q} + 2\Delta w_{\min}^2 \mu_q q_{\max}^2. \tag{41}
\end{aligned}$$

The first inequality follows by expanding the term and applying Cauchy–Schwarz. The second and fourth inequalities use Young's inequality together with the bound $|F(W_n)| \leq q_{\max}$. The last step holds under the step-size condition $3\Delta w_{\min}^2 \mu_q \leq \Delta w_{\min} \mu_q$. Combining (39) and (40) and substituting (41), we get:

$$
\begin{aligned}
\|W_{n+2} - W^\diamond\|^2 &\leq (1 - 2\Delta w_{\min}\mu_q)^2 \, \|W_n - W^\diamond\|^2 + 4\Delta w_{\min}^2 q_{\max}^2 + \frac{L_q^2 \Delta w_{\min}^2 q_{\max}^2}{\mu_q} \\
&\quad + \Delta w_{\min}\mu_q \, \|W_n - W^\diamond\|^2 + 2\Delta w_{\min}^2 \mu_q q_{\max}^2 \\
&\leq (1 - \Delta w_{\min}\mu_q)^2 \, \|W_n - W^\diamond\|^2 + \Delta w_{\min}^2 q_{\max}^2 \left( 4 + \frac{L_q^2}{\mu_q} + 2\mu_q \right).
\end{aligned}
\tag{42}
$$

Iterating the recursion for $n = 0, \ldots, 2N-1$ gives:

$$
\begin{aligned}
\|W_{2N} - W^\diamond\|^2 &\leq (1 - 2\Delta w_{\min}\mu_q)^{2N} \, \|W_0 - W^\diamond\|^2 + \sum_{n=0}^{2N-1} (1 - 2\Delta w_{\min}\mu_q)^n \Delta w_{\min}^2 q_{\max}^2 \left( 4 + \frac{L_q^2}{\mu_q} + 2\mu_q \right) \\
&\leq (1 - 2\Delta w_{\min}\mu_q)^{2N} \, \|W_0 - W^\diamond\|^2 + \frac{\Delta w_{\min} q_{\max}^2 \left( 4 + \frac{L_q^2}{\mu_q} + 2\mu_q \right)}{\mu_q}.
\end{aligned}
\tag{43}
$$

To ensure the total error remains within the tolerance $\delta$, we bound each term in the upper bound by $\delta/2$. This yields the following two sufficient conditions:

$$
\frac{\Delta w_{\min} q_{\max}^2 \left( 4 + \frac{L_q^2}{\mu_q} + 2\mu_q \right)}{\mu_q} \leq \frac{\delta}{2} \quad \text{and} \quad (1 - 2\mu_q \Delta w_{\min})^{2N} \|W_0 - W^\diamond\|^2 \leq \frac{\delta}{2}.
\tag{44}
$$

The first condition implies a lower bound on the achievable tolerance, requiring $\delta \geq \frac{2q_{\max}^2 \left( 4 + L_q^2/\mu_q + 2\mu_q \right) \Delta w_{\min}}{\mu_q}$. For the second condition, rearranging terms and taking the logarithm yields

$$
\log \left( \frac{2\|W_0 - W^\diamond\|^2}{\delta} \right) \leq 2N \log \left( \frac{1}{1 - 2\mu_q \Delta w_{\min}} \right).
\tag{45}
$$

Applying the inequality $\log(1/x) \geq 1 - x$ (valid for $x \in (0,1]$) with $x = 1 - 2\mu_q \Delta w_{\min}$, we observe that the right-hand side is lower-bounded by $2N(2\mu_q \Delta w_{\min}) = 4N\mu_q \Delta w_{\min}$. Therefore, to satisfy the inequality, it suffices to set

$$
N \geq \frac{1}{4\mu_q \Delta w_{\min}} \log \left( \frac{2\|W_0 - W^\diamond\|^2}{\delta} \right),
\tag{46}
$$

which completes the proof. $\qquad \square$

# D. Proof of Auxiliary Lemmas

## D.1. Proof of Lemma 3.5

*Proof.* According to the update of $Q_k$ in (12), we know

$$
Q_{k+1} - W^\diamond = (1 - \eta)(Q_k - W^\diamond) + \eta(P_{k+1} - W^\diamond).
$$

Taking square norm of both sides yield

$$
\begin{aligned}
\|Q_{k+1} - W^\diamond\|^2 &= (1-\eta)^2 \|Q_k - W^\diamond\|^2 + \eta^2 \|P_{k+1} - W^\diamond\|^2 + 2\eta(1-\eta)\langle P_{k+1} - W^\diamond, Q_k - W^\diamond \rangle \\
&= (1-\eta)^2 \|Q_k - W^\diamond\|^2 + \eta^2 \|P_{k+1} - W^\diamond\|^2 \\
&\quad + \eta(1-\eta)\|P_{k+1} - W^\diamond\|^2 + \eta(1-\eta)\|Q_k - W^\diamond\|^2 - \eta(1-\eta)\|P_{k+1} - Q_k\|^2 \\
&= (1-\eta)\|Q_k - W^\diamond\|^2 + \eta \|P_{k+1} - W^\diamond\|^2 - \eta(1-\eta)\|P_{k+1} - Q_k\|^2
\end{aligned}
\tag{47}
$$

where the second equation is because $\langle A, B \rangle = \frac{1}{2}\|A\|^2 + \frac{1}{2}\|B\|^2 - \frac{1}{2}\|A - B\|^2$. On the other hand, letting $\cos(P_{k+1} - W^\diamond, P_{k+1} - Q_k) := \cos\theta$, we have

$$\|Q_k - W^\diamond\|^2 = \|P_{k+1} - W^\diamond\|^2 + \|P_{k+1} - Q_k\|^2 - 2\|P_{k+1} - W^\diamond\|\|P_{k+1} - Q_k\|\cos\theta. \tag{48}$$

Plugging (48) into (47), we get

$$\|Q_{k+1} - W^\diamond\|^2 = \|P_{k+1} - W^\diamond\|^2 + (1-\eta)^2\|P_{k+1} - Q_k\|^2 - 2(1-\eta)\|P_{k+1} - W^\diamond\|\|P_{k+1} - Q_k\|\cos\theta$$

Since $\cos\theta > 0$ implies $\|P_{k+1} - Q_k\| \neq 0$, then choosing $1 > \eta > \max\left\{1 - \frac{2\|P_{k+1} - W^\diamond\|\cos\theta}{\|P_{k+1} - Q_k\|}, 0\right\}$ yields

$$\|Q_{k+1} - W^\diamond\|^2 < \|P_{k+1} - W^\diamond\|^2.$$

$\square$

### D.2. Proof of Lemma 3.10

*Proof.* From the digital signal processing perspective, the moving average update (12) defines a stable first-order infinite impulse response (IIS) low-pass filter from $P_k$ to $Q_k$. Taking the $z$-transform of (12) yields

$$Q(z) = (1-\eta)z^{-1}Q(z) + \eta P(z),$$

which gives the filter's transfer function

$$H(z) \triangleq \frac{Q(z)}{P(z)} = \frac{\eta}{1 - (1-\eta)z^{-1}}. \tag{49}$$

The transfer function has a single pole at $z = 1 - \eta$ with the following magnitude of the frequency response

$$|H(e^{j\omega})|^2 = \frac{\eta^2}{1 + (1-\eta)^2 - 2(1-\eta)\cos\omega}. \tag{50}$$

$\square$

## E. Proof of Theorem 3.7: Convergence of Algorithm 2

This section provides the convergence analysis details of the Algorithm 2 under the strongly convexity condition.

**Theorem 3.7** (Convergence of Algorithm 2). *Suppose Assumptions 3.1-3.4, 3.6 hold and the response functions $F_p, G_p, F_w, G_w$ satisfy Definition 2.1. If $C_\star \geq \frac{4\sqrt{2}\sigma}{\mu}\left(\frac{q_{\max}}{q_{\min}}\right)^{\frac{3}{2}}$, let $\alpha = \Theta\left(\frac{1}{\sqrt{K}}\right)$, $\beta = \Theta(\alpha\gamma\mu)$, $\eta = \Theta(\alpha\mu)$, $\gamma = \Theta(1)$, it holds that*

$$E_K \leq \mathcal{O}\left(\frac{\kappa_1 \kappa_2^5}{\sqrt{K}}\right) + \Theta(\Delta w_{\min}). \tag{15}$$

*where $\kappa_1 := L/\mu$ and $\kappa_2 := q_{\max}/q_{\min}$ are the condition number of the function and device.*

### E.1. Main proof

*Theorem 3.7.* Define the following notations $M_w(W_k) := Q_+(W_k) \odot Q_-(W_k) \in \mathbb{R}^D$, $M_p(P_k) := Q_+(P_k) \odot Q_-(P_k) \in \mathbb{R}^D$. The proof of the RIDER convergence relies on the following two lemmas, which provide the sufficient descent of $W_k$ and $\bar{W}_k$, respectively.

**Lemma E.1** (Descent lemma of lower-level problem). *Under Assumptions 3.1-3.2, it holds that*

$$\mathbb{E}_{\xi_k, b_k}[f(\bar{W}_{k+1})] \leq f(\bar{W}_k) - \frac{\alpha\gamma}{8q_{\max}}\|\nabla f(\bar{W}_k)\|^2_{M_p(P_k)} + \frac{\alpha\gamma\sigma^2}{2}\left\|\frac{G_p(P_k)}{\sqrt{F_p(P_k)}}\right\|^2_\infty + \Theta\left(\frac{\gamma\Delta w_{\min}}{q_{\min}}\right)$$

$$+ \frac{2q_{\max}\gamma}{\alpha}\mathbb{E}_{\xi_k}\left[\|Q_{k+1} - Q_k\|^2_{M_p(P_k)^\dagger}\right] + \frac{3\gamma^2 L}{2}\mathbb{E}_{\xi_k}\left[\|Q_{k+1} - Q_k\|^2\right] \tag{51}$$

$$- \left(\frac{\gamma}{2\alpha q_{\max}} - 3\gamma^2 L\right)\mathbb{E}_{\xi_k,b_k}\left[\left\|P_{k+1} - P_k + \alpha(\nabla f(\bar{W}_k;\xi_k) - \nabla f(\bar{W}_k))\odot F_p(P_k) - b_k\right\|^2\right]$$

$$+ 3\alpha^2\gamma^2 Lq_{\max}^2\sigma^2 + \frac{q_{\max}}{\alpha\gamma}\mathbb{E}_{\xi_k}\left[\|W_{k+1} - W_k\|^2_{M_p(P_k)^\dagger}\right] + \frac{3L}{2}\mathbb{E}_{\xi_k}[\|W_{k+1} - W_k\|^2].$$

*Keeping $W_k$ fixed, we also have*

$$\mathbb{E}_{\xi_k,b_k}[f(W_k + \gamma P_{k+1} - \gamma Q_k)] \tag{52}$$

$$\leq f(\bar{W}_k) - \frac{\alpha\gamma}{2q_{\max}}\|\nabla f(\bar{W}_k)\|^2_{M_p(P_k)} + \frac{\alpha\gamma\sigma^2}{2}\left\|\frac{G_p(P_k)}{\sqrt{F_p(P_k)}}\right\|^2_\infty + \Theta\left(\frac{\gamma\Delta w_{\min}}{q_{\min}}\right)$$

$$- \left(\frac{\gamma}{2\alpha q_{\max}} - \gamma^2 L\right)\mathbb{E}_{\xi_k,b_k}\left[\left\|P_{k+1} - P_k + \alpha(\nabla f(\bar{W}_k;\xi_k) - \nabla f(\bar{W}_k))\odot F_p(P_k) - b_k\right\|^2\right] + \alpha^2\gamma^2 Lq_{\max}^2\sigma^2.$$

$$\leq f(\bar{W}_k) - \frac{\alpha\gamma}{2q_{\max}}\|\nabla f(\bar{W}_k)\|^2_{M_p(P_k)} + \frac{\alpha\gamma\sigma^2}{2}\left\|\frac{G_p(P_k)}{\sqrt{F_p(P_k)}}\right\|^2_\infty + \alpha^2\gamma^2 Lq_{\max}^2\sigma^2 + \Theta\left(\frac{\gamma\Delta w_{\min}}{q_{\min}}\right)$$

Recall that we use the square norm of $P^*(W,Q) - Q = (W^* - W)/\gamma$ to measure the convergence of the upper-level problem.

**Lemma E.2** (Descent lemma of upper-level problem). *Under Assumption 2.1, it holds that*

$$\mathbb{E}_{b_k'}\left[\|P^*(W_{k+1}, Q_{k+1}) - Q_{k+1}\|^2\right] \leq \|P^*(W_k, Q_k) - Q_k\|^2 - \frac{\beta}{2\gamma q_{\max}}\|P^*(W_k, Q_k) - Q_k\|^2_{M_w(W_k)} + \Theta\left(\frac{\Delta w_{\min}}{\gamma q_{\max}}\right)$$

$$- \frac{\beta}{2\gamma q_{\max}}\left\|(P^*(W_k, Q_k) - Q_k)\odot F_w(W_k) - |P^*(W_k, Q_k) - Q_k|\odot G_w(W_k)\right\|^2$$

$$+ \frac{2\beta q_{\max}^3}{\gamma}\|P_{k+1} - P^*(W_k, Q_k)\|^2_{M_w(W_k)^\dagger} + \frac{2\beta^2}{\gamma^2}\|P_{k+1} - P^*(W_k, Q_k)\|^2. \tag{53}$$

**Lemma E.3** (Descent lemma of accumulated asymmetric sequence). *Under Assumption 2.1 and 3.6, it holds that*

$$\mathbb{E}_{\xi_k,b_k}[\varphi(P_{k+1})] \leq \varphi(P_k) - \frac{\alpha C_\star}{2}\|G_p(P_k)\|^2 + \alpha^2 Lq_{\max}^2\sigma^2 + \frac{\alpha q_{\max}^2}{2C_\star}\|\nabla f(\bar{W}_k)\|^2 + \Theta\left(\frac{\Delta w_{\min}}{\gamma q_{\max}}\right)$$

$$+ L\mathbb{E}_{\xi_k,b_k}\left[\left\|P_{k+1} - P_k + \alpha(\nabla f(\bar{W}_k;\xi_k) - \nabla f(\bar{W}_k))\odot F_p(P_k) - b_k\right\|^2\right]. \tag{54}$$

The proof of Lemma E.1, E.2 and E.3 are deferred to Appendices E.2, E.3 and E.4 respectively. For the response functions that satisfy Definition 2.1, we have

$$\min\{[M_w(W_k)]_d : d \in [D]\} \geq q_{\min}^2 > 0, \quad \text{and} \quad \min\{[M_p(P_k)]_d : d \in [D]\} \geq q_{\min}^2 > 0. \tag{55}$$

Now we deal with the weighted norm in two lemmas by Lemma A.1

$$\frac{\beta}{2\gamma q_{\max}}\|P^*(W_k, Q_k)\|^2_{M_p(P_k)} \geq \frac{\beta q_{\min}^2}{2\gamma q_{\max}}\|P^*(W_k, Q_k)\|^2 \tag{56}$$

$$\frac{\alpha}{4q_{\max}}\|\nabla f(\bar{W}_k)\|^2_{M_p(P_k)} \geq \frac{\alpha q_{\min}^2}{4q_{\max}}\|\nabla f(\bar{W}_k)\|^2 \tag{57}$$

$$\frac{q_{\max}}{\alpha}\|W_{k+1} - W_k\|^2_{M_p(P_k)^\dagger} \leq \frac{q_{\max}}{\alpha q_{\min}^2}\|W_{k+1} - W_k\|^2 \tag{58}$$

$$\frac{2\beta q_{\max}^3}{\gamma}\|P_{k+1} - P^*(W_k, Q_k)\|^2_{M_w(W_k)^\dagger} \leq \frac{2\beta q_{\max}^3}{\gamma q_{\min}^2}\|P_{k+1} - P^*(W_k, Q_k)\|^2 \tag{59}$$

$$\frac{2q_{\max}\gamma^2}{\alpha}\|Q_{k+1}-Q_k\|^2_{M_p(P_k)^\dagger}\leq\frac{2q_{\max}\gamma^2}{\alpha q_{\min}^2}\|Q_{k+1}-Q_k\|^2\,. \tag{60}$$

By inequality (58), the last two terms in the right-hand side (RHS) of (51) is bounded by

$$\frac{q_{\max}}{\alpha\gamma}\mathbb{E}_{b'_k}\left[\|W_{k+1}-W_k\|^2_{M_p(P_k)^\dagger}\right]+\frac{3L}{2}\mathbb{E}_{b'_k}\left[\|W_{k+1}-W_k\|^2\right] \tag{61}$$

$$\leq\frac{q_{\max}}{\alpha q_{\min}^2\gamma}\mathbb{E}_{b'_k}\left[\|W_{k+1}-W_k\|^2\right]+\frac{3L}{2}\mathbb{E}_{b'_k}\left[\|W_{k+1}-W_k\|^2\right]$$

$$\overset{(a)}{\leq}\frac{2q_{\max}}{\alpha q_{\min}^2\gamma}\mathbb{E}_{b'_k}\left[\|W_{k+1}-W_k\|^2\right]\leq\frac{2\beta^2 q_{\max}}{\alpha q_{\min}^2\gamma}\|(P_{k+1}-Q_k)\odot F_w(W_k)-|P_{k+1}-Q_k|\odot G_w(W_k)\|^2+\Theta\left(\frac{\beta q_{\max}\Delta w_{\min}}{\alpha q_{\min}^2\gamma}\right)$$

$$\leq\frac{4\beta^2 q_{\max}}{\alpha q_{\min}^2\gamma}\|(P^*(W_k,Q_k)-Q_k)\odot F_w(W_k)-|P^*(W_k,Q_k)-Q_k|\odot G_w(W_k)\|^2+\Theta\left(\frac{\beta q_{\max}\Delta w_{\min}}{\alpha q_{\min}^2\gamma}\right)$$

$$\overset{(b)}{\leq}+\frac{4\beta^2 q_{\max}}{\alpha q_{\min}^2\gamma}\|(P_{k+1}-Q_k)\odot F_w(W_k)-|P_{k+1}-Q_k|\odot G_w(W_k)$$

$$-((P^*(W_k,Q_k)-Q_k)\odot F_w(W_k)-|P^*(W_k,Q_k)-Q_k|\odot G_w(W_k))\|^2+\Theta\left(\frac{\gamma\Delta w_{\min}}{q_{\max}^2}\right)$$

$$\overset{(c)}{\leq}\frac{4\beta^2 q_{\max}}{\alpha q_{\min}^2\gamma}\|(P^*(W_k,Q_k)-Q_k)\odot F_w(W_k)-|P^*(W_k,Q_k)-Q_k|\odot G_w(W_k)\|^2+\frac{4\beta^2 q_{\max}^3}{\alpha q_{\min}^2\gamma}\|P_{k+1}-P^*(W_k,Q_k)\|^2$$

$$+\Theta\left(\frac{\gamma\Delta w_{\min}}{q_{\min}}\right)$$

where $(a)$ holds if the learning rate $\alpha$ is sufficiently small such that $\alpha\leq\frac{2q_{\max}}{3q_{\min}\gamma L}$; $(b)$ is because and $\beta\leq\frac{\gamma}{2q_{\max}},\alpha\leq\frac{q_{\max}}{q_{\min}^2\gamma L}$; and $(c)$ comes from the fact that the analog update is Lipschitz continuous (see Lemma A.2).

By inequality (60), the two terms related to $\|Q_{k+1}-Q_k\|^2$ in the RHS of (51) is bounded by

$$\frac{2q_{\max}\gamma}{\alpha}\mathbb{E}_{\xi_k}\left[\|Q_{k+1}-Q_k\|^2_{M_p(P_k)^\dagger}\right]+L\gamma^2\mathbb{E}_{\xi_k}\left[\|Q_{k+1}-Q_k\|^2\right]\leq\frac{3q_{\max}\gamma}{\alpha q_{\min}^2}\mathbb{E}_{\xi_k}\left[\|Q_{k+1}-Q_k\|^2\right]\,. \tag{62}$$

where the last inequality holds when choosing stepsize $\alpha\leq\frac{q_{\max}}{q_{\min}^2\gamma L}$.

With all the inequalities and lemmas above, we are ready to prove the main conclusion in Theorem 3.7 now. Define a Lyapunov function by

$$V_k:=f(\bar{W}_k)-f^*+C_1\|P^*(W_k,Q_k)-Q_k\|^2+C_2(\varphi(P_k)-\varphi^*). \tag{63}$$

By Lemmas E.1 and E.2, we show that $V_k$ has sufficient descent in expectation

$$\mathbb{E}_{\xi_k,b_k,b'_k}[V_{k+1}]=\mathbb{E}_{\xi_k,b_k,b'_k}\left[f(\bar{W}_{k+1})-f^*+C_1\|P^*(W_{k+1},Q_{k+1})-Q_{k+1}\|^2+C_2(\varphi(P_{k+1})-\varphi^*)\right] \tag{64}$$

$$\leq f(\bar{W}_k)-f^*-\frac{\alpha\gamma}{8q_{\max}}\|\nabla f(\bar{W}_k)\|^2_{M_p(P_k)}+\frac{\alpha\sigma^2\gamma}{2}\left\|\frac{G_p(P_k)}{\sqrt{F_p(P_k)}}\right\|^2_\infty+3\alpha^2 L q_{\max}^2\sigma^2+\Theta\left(\frac{\gamma\Delta w_{\min}}{q_{\min}}\right)$$

$$+\frac{6q_{\max}\gamma^2\eta^2}{\alpha q_{\min}^2}\mathbb{E}_{\xi_k,b_k}\left[\|P_{k+1}-P^*(W_k,Q_k)\|^2+\|P^*(W_k,Q_k)-Q_k\|^2\right]+\frac{4\beta^2 q_{\max}^3}{\alpha q_{\min}^2}\mathbb{E}_{\xi_k,b_k}[\|P_{k+1}-P^*(W_k,Q_k)\|^2]$$

$$-\left(\frac{1}{2\alpha q_{\max}}-3\gamma^2 L\right)\mathbb{E}_{\xi_k,b_k}\left[\left\|P_{k+1}-P_k+\alpha(\nabla f(\bar{W}_k;\xi_k)-\nabla f(\bar{W}_k))\odot F_p(P_k)-b_k\right\|^2\right]$$

$$+\frac{4\beta^2 q_{\max}}{\alpha q_{\min}^2}\|(P^*(W_k,Q_k)-Q_k)\odot F_w(W_k)-|P^*(W_k,Q_k)-Q_k|\odot G_w(W_k)\|^2$$

$$+C_1\Bigg(\|P^*(W_k,Q_k)-Q_k\|^2-\frac{\beta}{2\gamma q_{\max}}\|P^*(W_k,Q_k)-Q_k\|^2_{M_w(W_k)}+\frac{3\beta q_{\max}^3}{\gamma q_{\min}^2}\mathbb{E}_{\xi_k}[\|P_{k+1}-P^*(W_k,Q_k)\|^2]$$

$$- \frac{\beta}{2\gamma q_{\max}} \|(P^*(W_k, Q_k) - Q_k) \odot F_w(W_k) - |P^*(W_k, Q_k) - Q_k| \odot G_w(W_k)\|^2 + \Theta\left(\frac{\Delta w_{\min}}{\gamma q_{\max}}\right)\ \Big)$$

$$+ C_2\Big( \varphi(P_k) - \varphi^* - \frac{\alpha C_\star}{2} \|G_p(P_k)\|^2 + \alpha^2 L q_{\max}^2 \sigma^2 + \frac{\alpha q_{\max}^2}{2 C_\star} \|\nabla f(\bar{W}_k)\|^2$$

$$+ L\mathbb{E}_{\xi_k, b_k}\left[\left\|P_{k+1} - P_k + \alpha(\nabla f(\bar{W}_k; \xi_k) - \nabla f(\bar{W}_k)) \odot F_p(P_k) - b_k\right\|^2\right] + \Theta\left(\frac{\Delta w_{\min}}{\gamma q_{\max}}\right)\ \Big)$$

$$\leq V_k - \left(\frac{\alpha q_{\min}^2}{8 q_{\max}} - \frac{C_2 \alpha q_{\max}^2}{2 C_\star}\right) \|\nabla f(\bar{W}_k)\|^2 - \left(\frac{\alpha C_2 C_\star}{2} - \frac{\alpha \sigma^2}{2 q_{\min}}\right) \|G_p(P_k)\|^2 + \Theta\left(\frac{(C_1 + C_2)\Delta w_{\min}}{\gamma q_{\max}} + \frac{\gamma \Delta w_{\min}}{q_{\min}}\right)$$

$$+ (3 + C_2)\alpha^2 L q_{\max}^2 \sigma^2 - \left(\frac{\beta q_{\min}^2}{2\gamma q_{\max}} C_1 - \frac{6 q_{\max} \gamma^2 \eta^2}{\alpha q_{\min}^2}\right) \|P^*(W_k, Q_k) - Q_k\|^2$$

$$- \left(\frac{\beta}{2\gamma q_{\max}} C_1 - \frac{4\beta^2 q_{\max}}{\alpha q_{\min}^2}\right) \|(P^*(W_k, Q_k) - Q_k) \odot F_w(W_k) - |P^*(W_k, Q_k) - Q_k| \odot G_w(W_k)\|^2$$

$$+ \left(\frac{3\beta q_{\max}^3}{\gamma q_{\min}^2} C_1 + \frac{4\beta^2 q_{\max}^3}{\alpha q_{\min}^2} + \frac{6 q_{\max} \gamma^2 \eta^2}{\alpha q_{\min}^2}\right) \mathbb{E}_{\xi_k}[\|P_{k+1} - P^*(W_k, Q_k)\|^2]$$

$$- \left(\frac{1}{2\alpha q_{\max}} - 3\gamma^2 L - C_2 L\right) \mathbb{E}_{\xi_k, b_k}\left[\left\|P_{k+1} - P_k + \alpha(\nabla f(\bar{W}_k; \xi_k) - \nabla f(\bar{W}_k)) \odot F_p(P_k) - b_k\right\|^2\right].$$

The first inequality holds for:

$$\mathbb{E}_{\xi_k}\left[\|Q_{k+1} - Q_k\|^2\right] = \mathbb{E}_{\xi_k}\left[\| - \eta(P_{k+1} - Q_k)\|^2\right]$$
$$\leq 2\eta^2 \mathbb{E}_{\xi_k}\left[\|P_{k+1} - P^*(W_k, Q_k)\|^2 + \|P^*(W_k, Q_k) - Q_k\|^2\right] \tag{65}$$

and the learning rate $\beta$ is sufficiently small such that $\frac{2\beta^2}{\gamma^2} \leq \frac{\beta q_{\max}^3}{\gamma q_{\min}^2}$. Let $C_1 = \frac{10\beta\gamma q_{\max}^2}{\alpha q_{\min}^2}$, which leads to the positive coefficient in front of $\|P^*(W_k, Q_k) \odot F_w(W_k) - |P^*(W_k, Q_k)| \odot G_w(W_k)\|^2$. Let $\alpha$ large enough, which leads to the positive coefficient in front of $\mathbb{E}_{\xi_k, b_k}\left[\left\|P_{k+1} - P_k + \alpha(\nabla f(\bar{W}_k; \xi_k) - \nabla f(\bar{W}_k)) \odot F_p(P_k) - b_k\right\|^2\right]$.

$$\mathbb{E}_{\xi_k, b_k}[V_{k+1}] \leq V_k - \left(\frac{\alpha q_{\min}^2}{8 q_{\max}} - \frac{C_2 \alpha q_{\max}^2}{2 C_\star}\right) \|\nabla f(\bar{W}_k)\|^2 - \left(\frac{\alpha C_2 C_\star}{2} - \frac{\alpha \sigma^2}{2 q_{\min}}\right) \|G_p(P_k)\|^2 \tag{66}$$
$$+ \left(\frac{30\beta^2 q_{\max}^5}{\alpha q_{\min}^4} + \frac{4\beta^2 q_{\max}^3}{\alpha q_{\min}^2} + \frac{6 q_{\max} \gamma^2 \eta^2}{\alpha q_{\min}^2}\right) \mathbb{E}_{\xi_k, b_k}[\|P_{k+1} - P^*(W_k, Q_k)\|^2]$$
$$- \left(\frac{\beta q_{\min}^2}{2\gamma q_{\max}} C_1 - \frac{6 q_{\max} \gamma^2 \eta^2}{\alpha q_{\min}^2}\right) \|P^*(W_k, Q_k) - Q_k\|^2 + (3 + C_2)\alpha^2 L q_{\max}^2 \sigma^2 + \Theta(\Delta w_{\min}).$$

Now we bound the term $\left\|\nabla f(\bar{W}_k)\right\|^2$ in (66) by $\mu$-PL condition (which is implied by Assumption 3.3)

$$\frac{\alpha q_{\min}^2}{8 q_{\max}} \left\|\nabla f(\bar{W}_k)\right\|^2 \geq \frac{\alpha \mu q_{\min}^2}{4 q_{\max}} (f(\bar{W}_k) - f^*). \tag{67}$$

Notice that the $\|P_{k+1} - P^*(W_k, Q_k)\|^2$ appears in the RHS of (66) above, we also need the following lemma to bound it.

**Lemma E.4** (Quadratic growth, Theorem 2, (Karimi et al., 2016)). *Under Assumption 3.3, defining the optimal solution as* $W^* := \arg\min_W f(W)$, *it holds that*

$$\frac{2}{\mu}(f(W) - f^*) \geq \|W - W^*\|^2. \tag{68}$$

Replacing $W$ in (68) with $W_k + \gamma(P_{k+1} - Q_k)$, we bound the term $\|P_{k+1} - P^*(W_k, Q_k)\|^2$ in (53) by

$$\frac{2}{\mu}(f(W_k + \gamma(P_{k+1} - Q_k)) - f^*) \geq \|W_k + \gamma P_{k+1} - \gamma Q_k - W^*\|^2 = \gamma^2 \|P_{k+1} - P^*(W_k, Q_k)\|^2. \tag{69}$$

By inequality (52) in Lemma E.1 and (69), we bound the $\|P_{k+1} - P^*(W_k, Q_k)\|^2$ by

$$\left( \frac{30\beta^2 q_{\max}^5}{\alpha q_{\min}^4} + \frac{4\beta^2 q_{\max}^3}{\alpha q_{\min}^2} + \frac{6q_{\max}\gamma^2\eta^2}{\alpha q_{\min}^2} \right) \mathbb{E}_{\xi_k, b_k}[\|P_{k+1} - P^*(W_k, Q_k)\|^2] \tag{70}$$

$$\overset{(a)}{\leq} \left( \frac{34\beta^2 q_{\max}^5}{\alpha q_{\min}^4} + \frac{6q_{\max}\gamma^2\eta^2}{\alpha q_{\min}^2} \right) \mathbb{E}_{\xi_k, b_k}[\|P_{k+1} - P^*(W_k, Q_k)\|^2]$$

$$\overset{(b)}{\leq} \left( \frac{68\beta^2 q_{\max}^5}{\alpha\gamma^2\mu q_{\min}^4} + \frac{12q_{\max}\eta^2}{\alpha q_{\min}^2\mu} \right) \mathbb{E}_{\xi_k, b_k}[f(W_k + \gamma(P_{k+1} - Q_k)) - f^*]$$

$$\overset{(c)}{\leq} \left( \frac{68\beta^2 q_{\max}^5}{\alpha\gamma^2\mu q_{\min}^4} + \frac{12q_{\max}\eta^2}{\alpha q_{\min}^2\mu} \right) \left( f(\bar{W}_k) - f^* + \frac{\alpha\sigma^2}{2} \left\| \frac{G_p(P_k)}{\sqrt{F_p(P_k)}} \right\|_\infty^2 + \alpha^2 L q_{\max}^2\sigma^2 + \Theta\left( \frac{\gamma\Delta w_{\min}}{q_{\min}} \right) \right)$$

$$- \left( \frac{68\beta^2 q_{\max}^5}{\alpha\gamma^2\mu q_{\min}^4} + \frac{12q_{\max}\eta^2}{\alpha q_{\min}^2\mu} \right) \frac{\alpha}{2q_{\max}} \|\nabla f(\bar{W}_k)\|^2$$

$$\overset{(d)}{\leq} \left( \frac{68\beta^2 q_{\max}^5}{\alpha\gamma^2\mu q_{\min}^4} + \frac{12q_{\max}\eta^2}{\alpha q_{\min}^2\mu} \right) (f(\bar{W}_k) - f^*) + \mathcal{O}\left( \beta^2\sigma^2 \left\| \frac{G_p(P_k)}{\sqrt{F_p(P_k)}} \right\|_\infty^2 + \alpha\beta^2 q_{\max}^2\sigma^2 \right) + \Theta\left( \frac{\gamma q_{\max}\Delta w_{\min}}{q_{\min}^2} \right)$$

$$\overset{(e)}{\leq} \left( \frac{68\beta^2 q_{\max}^5}{\alpha\gamma^2\mu q_{\min}^4} + \frac{12q_{\max}\eta^2}{\alpha q_{\min}^2\mu} \right) (f(\bar{W}_k) - f^*) + \alpha\sigma^2 \left\| \frac{G_p(P_k)}{\sqrt{F_p(P_k)}} \right\|_\infty^2 + \alpha^2 L q_{\max}^2\sigma^2 + \Theta\left( \frac{\gamma q_{\max}\Delta w_{\min}}{q_{\min}^2} \right)$$

where $(a)$ uses $\frac{q_{\max}}{q_{\min}} \geq 1$; (b) comes from (69); (c) comes from (52) in Lemma E.1; (d) holds by setting $\eta \leq \frac{\beta q_{\max}^2}{\gamma q_{\min}}$ and $\beta \leq \frac{\gamma}{q_{\max}}$; (e) holds given $\alpha$ and $\beta$ is sufficiently small.

Substituting (67) and (70) back into (66) yields

$$\mathbb{E}_{\xi_k, b_k}[V_{k+1}] \tag{71}$$

$$\leq V_k - \left( \frac{\alpha C_2 C_\star}{2} - \frac{3\alpha\sigma^2}{2q_{\min}} \right) \|G_p(P_k)\|^2 - \left( \frac{\alpha\mu q_{\min}^2}{4q_{\max}} - \frac{68\beta^2 q_{\max}^5}{\alpha\gamma^2\mu q_{\min}^4} - \frac{12q_{\max}\eta^2}{\alpha q_{\min}^2\mu} - \frac{C_2\alpha q_{\max}^2}{2C_\star} \right) (f(\bar{W}_k) - f^*)$$

$$- \left( \frac{\beta q_{\min}^2}{2\gamma q_{\max}} C_1 - \frac{6q_{\max}\gamma^2\eta^2}{\alpha q_{\min}^2} \right) \|P^*(W_k, Q_k) - Q_k\|^2 + (4 + C_2)\alpha^2 L q_{\max}^2\sigma^2 + \Theta(\Delta w_{\min})$$

$$= V_k - \left( \frac{\alpha C_2 C_\star}{2} - \frac{3\alpha\sigma^2}{2q_{\min}} \right) \|G_p(P_k)\|^2 - \left( \frac{\alpha\mu q_{\min}^2}{8q_{\max}} - \frac{12q_{\max}\eta^2}{\alpha q_{\min}^2\mu} - \frac{C_2\alpha q_{\max}^2}{2C_\star} \right) (f(\bar{W}_k) - f^*)$$

$$- \left( \frac{\alpha\mu q_{\min}^5}{8\sqrt{34}q_{\max}^4} - \frac{12\sqrt{34}q_{\max}^2\eta^2}{5\alpha\mu q_{\min}^3} \right) C_1\|P^*(W_k, Q_k) - Q_k\|^2 + (4 + C_2)\alpha^2 L q_{\max}^2\sigma^2 + \Theta(\Delta w_{\min})$$

$$\leq V_k - \left( \frac{\alpha\mu q_{\min}^5}{8\sqrt{34}q_{\max}^4} - \frac{12\sqrt{34}q_{\max}^2\eta^2}{5\alpha\mu q_{\min}^3} \right) \left( (f(\bar{W}_k) - f^*) + C_1\|P^*(W_k, Q_k) - Q_k\|^2 + \|G_p(P_k)\|^2 \right)$$

$$+ (4 + C_2)\alpha^2 L q_{\max}^2\sigma^2 + \Theta(\Delta w_{\min})$$

$$\leq V_k - \frac{\alpha\mu q_{\min}^5}{16\sqrt{34}q_{\max}^4} \left( (f(\bar{W}_k) - f^*) + C_1\|P^*(W_k, Q_k) - Q_k\|^2 + \|G_p(P_k)\|^2 \right) + 5\alpha^2 L q_{\max}^2\sigma^2 + \Theta(\Delta w_{\min})$$

where the second step chooses the learning rate by $\beta = \frac{\alpha\gamma\mu q_{\min}^3}{4\sqrt{34}q_{\max}^3}$. The third step holds since $\frac{q_{\max}^2}{q_{\min}^2} \geq 1$, and choose $\frac{4\sigma^2}{C_\star q_{\min}} \leq C_2 \leq \frac{\mu q_{\min}^2 C_\star}{8q_{\max}^3}$, such that $\frac{\alpha C_2 C_\star}{2} - \frac{3\alpha\sigma^2}{2q_{\min}} \geq \frac{\alpha\mu q_{\min}^5}{8\sqrt{34}q_{\max}^4} - \frac{12\sqrt{34}q_{\max}^2\eta^2}{5\alpha\mu q_{\min}^3}$ and $\frac{C_2\alpha q_{\max}^2}{2C_\star} \leq \frac{\alpha\mu q_{\min}^2}{16q_{\max}}$. The last step chooses $\eta \leq \frac{\alpha\mu q_{\min}^4}{52q_{\max}^3}$ such that $\frac{12\sqrt{34}q_{\max}^2\eta^2}{5\alpha\mu q_{\min}^3} \leq \frac{\alpha\mu q_{\min}^5}{32\sqrt{34}q_{\max}^4}$.

Rearranging (71), taking expectations with respect to $\{\xi_k\}_{k=0}^K$, and averaging over $k = 0, \ldots, K$, and choosing the stepsize $\alpha = \Theta(K^{-1/2})$, we obtain

$$\frac{1}{K} \sum_{k=0}^K \mathbb{E}\left( \left( f(\bar{W}_k) - f^* \right) + C_1 \big\| P^*(W_k, Q_k) - Q_k \big\|^2 + \big\| G_p(P_k) \big\|^2 \right)$$

$$\leq \frac{16\sqrt{34}\, q_{\max}^4}{\mu\, q_{\min}^5} \left( \frac{V_0 - V_{[K]}}{\alpha K} + 5\alpha L q_{\max}^2 \sigma^2 \right) + \Theta(\Delta w_{\min}) \leq \mathcal{O}\left( \frac{\kappa_2^5}{\sqrt{K}} \right) + \Theta(\Delta w_{\min}). \tag{72}$$

Since $\|P^*(W_k, Q_k) - Q_k\| = \|W_k - W^*\|/\gamma$ and $f(\bar{W}_k) - f^* \geq \frac{\mu}{2}\|\bar{W}_k - W^*\|^2 = \frac{\mu}{2}\|W_k - W^* + \gamma(P_k - Q_k)\|^2$, we know that

$$\|P_k - Q_k\|^2 \leq 2\gamma^{-2}\|\bar{W}_k - W^*\|^2 + 2\gamma^{-2}\|W_k - W^*\|^2 \leq 2\gamma^{-2}\|\bar{W}_k - W^*\|^2 + 2\|P^*(W_k, Q_k) - Q_k\|^2$$
$$\leq \frac{4}{\mu}\gamma^{-2}(f(\bar{W}_k) - f^*) + 2\|P^*(W_k, Q_k) - Q_k\|^2$$

and thus by choosing $C_1 = \mathcal{O}(\gamma^2)$, we have

$$\frac{1}{K}\sum_{k=0}^{K} \mathbb{E}\left( \left\|W_k - W^*\right\|^2 + C_1\|P_k - Q_k\|^2 + \left\|G_p(P_k)\right\|^2 \right)$$
$$\leq \frac{16\sqrt{34}\, q_{\max}^4}{\mu\, q_{\min}^5} \left( \frac{V_0 - V_{[K]}}{\alpha K} + 5\alpha L q_{\max}^2 \sigma^2 \right) + \Theta(\Delta w_{\min}) \leq \mathcal{O}\left( \frac{\kappa_2^5}{\kappa_1\sqrt{K}} \right) + \Theta(\Delta w_{\min}). \tag{73}$$

The proof of Theorem 3.7 is completed. $\square$

### E.2. Proof of Lemma E.1: Descent of sequence $\bar{W}_k$

**Lemma E.1** (Descent lemma of lower-level problem). *Under Assumptions 3.1-3.2, it holds that*

$$\mathbb{E}_{\xi_k, b_k}[f(\bar{W}_{k+1})] \leq f(\bar{W}_k) - \frac{\alpha\gamma}{8q_{\max}}\|\nabla f(\bar{W}_k)\|_{M_p(P_k)}^2 + \frac{\alpha\gamma\sigma^2}{2}\left\| \frac{G_p(P_k)}{\sqrt{F_p(P_k)}} \right\|_\infty^2 + \Theta\left( \frac{\gamma\Delta w_{\min}}{q_{\min}} \right)$$
$$+ \frac{2q_{\max}\gamma}{\alpha}\mathbb{E}_{\xi_k}\left[ \|Q_{k+1} - Q_k\|_{M_p(P_k)^\dagger}^2 \right] + \frac{3\gamma^2 L}{2}\mathbb{E}_{\xi_k}\left[ \|Q_{k+1} - Q_k\|^2 \right] \tag{51}$$
$$- \left( \frac{\gamma}{2\alpha q_{\max}} - 3\gamma^2 L \right) \mathbb{E}_{\xi_k, b_k}\left[ \left\|P_{k+1} - P_k + \alpha(\nabla f(\bar{W}_k; \xi_k) - \nabla f(\bar{W}_k)) \odot F_p(P_k) - b_k\right\|^2 \right]$$
$$+ 3\alpha^2\gamma^2 L q_{\max}^2 \sigma^2 + \frac{q_{\max}}{\alpha\gamma}\mathbb{E}_{\xi_k}\left[ \|W_{k+1} - W_k\|_{M_p(P_k)^\dagger}^2 \right] + \frac{3L}{2}\mathbb{E}_{\xi_k}[\|W_{k+1} - W_k\|^2].$$

*Keeping $W_k$ fixed, we also have*

$$\mathbb{E}_{\xi_k, b_k}[f(W_k + \gamma P_{k+1} - \gamma Q_k)] \tag{52}$$
$$\leq f(\bar{W}_k) - \frac{\alpha\gamma}{2q_{\max}}\|\nabla f(\bar{W}_k)\|_{M_p(P_k)}^2 + \frac{\alpha\gamma\sigma^2}{2}\left\| \frac{G_p(P_k)}{\sqrt{F_p(P_k)}} \right\|_\infty^2 + \Theta\left( \frac{\gamma\Delta w_{\min}}{q_{\min}} \right)$$
$$- \left( \frac{\gamma}{2\alpha q_{\max}} - \gamma^2 L \right) \mathbb{E}_{\xi_k, b_k}\left[ \left\|P_{k+1} - P_k + \alpha(\nabla f(\bar{W}_k; \xi_k) - \nabla f(\bar{W}_k)) \odot F_p(P_k) - b_k\right\|^2 \right] + \alpha^2\gamma^2 L q_{\max}^2 \sigma^2.$$
$$\leq f(\bar{W}_k) - \frac{\alpha\gamma}{2q_{\max}}\|\nabla f(\bar{W}_k)\|_{M_p(P_k)}^2 + \frac{\alpha\gamma\sigma^2}{2}\left\| \frac{G_p(P_k)}{\sqrt{F_p(P_k)}} \right\|_\infty^2 + \alpha^2\gamma^2 L q_{\max}^2 \sigma^2 + \Theta\left( \frac{\gamma\Delta w_{\min}}{q_{\min}} \right)$$

*Lemma E.1.* The $L$-smooth assumption (Assumption 3.1) implies that

$$\mathbb{E}_{\xi_k, b_k}[f(\bar{W}_{k+1})] \leq f(\bar{W}_k) + \mathbb{E}_{\xi_k, b_k}[\langle \nabla f(\bar{W}_k), \bar{W}_{k+1} - \bar{W}_k \rangle] + \frac{L}{2}\mathbb{E}_{\xi_k, b_k}[\|\bar{W}_{k+1} - \bar{W}_k\|^2] \tag{74}$$
$$= f(\bar{W}_k) + \gamma \underbrace{\mathbb{E}_{\xi_k, b_k}[\langle \nabla f(\bar{W}_k), P_{k+1} - P_k \rangle]}_{(a)} + \underbrace{\mathbb{E}_{\xi_k}[\langle \nabla f(\bar{W}_k), W_{k+1} - W_k \rangle]}_{(b)} + \underbrace{\frac{L}{2}\mathbb{E}_{\xi_k, b_k}[\|\bar{W}_{k+1} - \bar{W}_k\|^2]}_{(c)}.$$
$$- \gamma \underbrace{\mathbb{E}_{\xi_k}[\langle \nabla f(\bar{W}_k), Q_{k+1} - Q_k \rangle]}_{(d)}$$

Next, we will handle each term in the RHS of (74) separately.

**Bound of the second term (a).** To bound term (a) in the RHS of (74), we leverage the assumption that noise has expectation 0 (Assumption 3.2)

$$\mathbb{E}_{\xi_k,b_k}[\langle \nabla f(\bar{W}_k), P_{k+1} - P_k \rangle] \tag{75}$$

$$= \alpha \mathbb{E}_{\xi_k,b_k} \left[ \left\langle \nabla f(\bar{W}_k) \odot \sqrt{F_p(P_k)}, \frac{P_{k+1} - P_k}{\alpha \sqrt{F_p(P_k)}} + (\nabla f(\bar{W}_k; \xi_k) - \nabla f(\bar{W}_k)) \odot \sqrt{F_p(P_k)} - \frac{b_k}{\alpha \sqrt{F_p(P_k)}} \right\rangle \right]$$

$$= -\frac{\alpha}{2} \|\nabla f(\bar{W}_k) \odot \sqrt{F_p(P_k)}\|^2$$

$$- \frac{1}{2\alpha} \mathbb{E}_{\xi_k,b_k} \left[ \left\| \frac{P_{k+1} - P_k}{\sqrt{F_p(P_k)}} + \alpha(\nabla f(\bar{W}_k; \xi_k) - \nabla f(\bar{W}_k)) \odot \sqrt{F_p(P_k)} - \frac{b_k}{\sqrt{F_p(P_k)}} \right\|^2 \right]$$

$$+ \frac{1}{2\alpha} \mathbb{E}_{\xi_k,b_k} \left[ \left\| \frac{P_{k+1} - P_k}{\sqrt{F_p(P_k)}} + \alpha \nabla f(\bar{W}_k; \xi_k) \odot \sqrt{F_p(P_k)} - \frac{b_k}{\sqrt{F_p(P_k)}} \right\|^2 \right].$$

The second term in the RHS of (75) can be bounded by

$$\frac{1}{2\alpha} \mathbb{E}_{\xi_k,b_k} \left[ \left\| \frac{P_{k+1} - P_k}{\sqrt{F_p(P_k)}} + \alpha(\nabla f(\bar{W}_k; \xi_k) - \nabla f(\bar{W}_k)) \odot \sqrt{F_p(P_k)} - \frac{b_k}{\sqrt{F_p(P_k)}} \right\|^2 \right] \tag{76}$$

$$= \frac{1}{2\alpha} \mathbb{E}_{\xi_k,b_k} \left[ \left\| \frac{P_{k+1} - P_k + \alpha(\nabla f(\bar{W}_k; \xi_k) - \nabla f(\bar{W}_k)) \odot F_p(P_k) - b_k}{\sqrt{F_p(P_k)}} \right\|^2 \right]$$

$$\geq \frac{1}{2\alpha q_{\max}} \mathbb{E}_{\xi_k,b_k} \left[ \|P_{k+1} - P_k + \alpha(\nabla f(\bar{W}_k; \xi_k) - \nabla f(\bar{W}_k)) \odot F_p(P_k) - b_k\|^2 \right].$$

The third term in the RHS of (75) can be bounded by the bounded variance (Assumption 3.2)

$$\frac{1}{2\alpha} \mathbb{E}_{\xi_k,b_k} \left[ \left\| \frac{P_{k+1} - P_k}{\sqrt{F_p(P_k)}} + \alpha \nabla f(\bar{W}_k; \xi_k) \odot \sqrt{F_p(P_k)} - \frac{b_k}{\sqrt{F_p(P_k)}} \right\|^2 \right] \tag{77}$$

$$\leq \frac{\alpha}{2} \mathbb{E}_{\xi_k,b_k} \left[ \left\| |\nabla f(\bar{W}_k; \xi_k)| \odot \frac{G_p(P_k)}{\sqrt{F_p(P_k)}} - \frac{b_k}{\alpha \sqrt{F_p(P_k)}} \right\|^2 \right] \tag{78}$$

$$\overset{(a)}{=} \frac{\alpha}{2} \mathbb{E}_{\xi_k,b_k} \left[ \left\| \nabla f(\bar{W}_k; \xi_k) \odot \frac{G_p(P_k)}{\sqrt{F_p(P_k)}} - \frac{b_k}{\alpha \sqrt{F_p(P_k)}} \right\|^2 \right] \tag{79}$$

$$\overset{(b)}{\leq} \frac{\alpha}{2} \left\| \nabla f(\bar{W}_k) \odot \frac{G_p(P_k)}{\sqrt{F_p(P_k)}} \right\|^2 + \frac{\alpha \sigma^2}{2} \left\| \frac{G_p(P_k)}{\sqrt{F_p(P_k)}} \right\|_\infty^2 + \Theta\left(\frac{\Delta w_{\min}}{q_{\min}}\right).$$

where $(a)$ is because $\mathbb{E}_{b_k} \left\langle |\nabla f(\bar{W}_k; \xi_k)| \odot \frac{G_p(P_k)}{\sqrt{F_p(P_k)}}, \frac{b_k}{\alpha \sqrt{F_p(P_k)}} \right\rangle = 0$ according to Assumption 3.4 and $(b)$ comes from $\mathbb{E}[\|A\|^2] = \|\mathbb{E}[A]\|^2 + \mathbb{E}[\|A - \mathbb{E}[A]\|^2]$ and Assumption 3.2.

Notice that the first term in the RHS of (75) and the second term in the RHS of (77) can be bounded together

$$-\frac{\alpha}{2} \|\nabla f(\bar{W}_k) \odot \sqrt{F_p(P_k)}\|^2 + \frac{\alpha}{2} \left\| \nabla f(\bar{W}_k) \odot \frac{G_p(P_k)}{\sqrt{F_p(P_k)}} \right\|^2 \tag{80}$$

$$= -\frac{\alpha}{2} \sum_{d \in [D]} \left( [\nabla f(\bar{W}_k)]_d^2 \left( [F_p(P_k)]_d - \frac{[G_p(P_k)]_d^2}{[F_p(P_k)]_d} \right) \right)$$

$$= -\frac{\alpha}{2}\sum_{d\in[D]}\left([\nabla f(\bar{W}_k)]_d^2\left(\frac{[F_p(P_k)]_d^2 - [G_p(P_k)]_d^2}{[F_p(P_k)]_d}\right)\right)$$

$$\leq -\frac{\alpha}{2q_{\max}}\sum_{d\in[D]}\left([\nabla f(\bar{W}_k)]_d^2\left([F_p(P_k)]_d^2 - [G_p(P_k)]_d^2\right)\right)$$

$$= -\frac{\alpha}{2q_{\max}}\|\nabla f(\bar{W}_k)\|_{M_p(P_k)}^2 \leq 0.$$

Plugging (76) to (80) into (75), we bound the term (a) by

$$\gamma\mathbb{E}_{\xi_k,b_k}[\langle\nabla f(\bar{W}_k), P_{k+1}-P_k\rangle] \leq -\frac{\alpha\gamma}{2q_{\max}}\|\nabla f(\bar{W}_k)\|_{M_p(P_k)}^2 + \frac{\alpha\gamma\sigma^2}{2}\left\|\frac{G_p(P_k)}{\sqrt{F_p(P_k)}}\right\|_\infty^2 + \Theta\left(\frac{\gamma\Delta w_{\min}}{q_{\min}}\right) \tag{81}$$

$$-\frac{\gamma}{2\alpha q_{\max}}\mathbb{E}_{\xi_k}\left[\left\|P_{k+1}-P_k + \alpha(\nabla f(\bar{W}_k;\xi_k)-\nabla f(\bar{W}_k))\odot F_p(P_k)-b_k\right\|^2\right].$$

**Bound of the third term (b).** By Young's inequality, we have

$$\mathbb{E}_{\xi_k}[\langle\nabla f(\bar{W}_k), W_{k+1}-W_k\rangle] \leq \frac{\alpha\gamma}{4q_{\max}}\|\nabla f(\bar{W}_k)\|_{M_p(P_k)}^2 + \frac{q_{\max}}{\alpha\gamma}\mathbb{E}_{\xi_k}[\|W_{k+1}-W_k\|_{M_p(P_k)^\dagger}^2]. \tag{82}$$

**Bound of the fourth term (c).** Repeatedly applying inequality $\|U+V\|^2 \leq 2\|U\|^2 + 2\|V\|^2$ for any $U,V\in\mathbb{R}^D$, we have

$$\frac{L}{2}\mathbb{E}_{\xi_k,b_k}[\|\bar{W}_{k+1}-\bar{W}_k\|^2] \tag{83}$$

$$\leq \frac{3L}{2}\mathbb{E}_{\xi_k}[\|W_{k+1}-W_k\|^2] + \frac{3\gamma^2 L}{2}\mathbb{E}_{\xi_k,b_k}[\|P_{k+1}-P_k\|^2] + \frac{3\gamma^2 L}{2}\mathbb{E}_{\xi_k}[\|Q_{k+1}-Q_k\|^2]$$

$$\leq \frac{3L}{2}\mathbb{E}_{\xi_k}[\|W_{k+1}-W_k\|^2] + 3\gamma^2 L\mathbb{E}_{\xi_k,b_k}\left[\left\|P_{k+1}-P_k + \alpha(\nabla f(\bar{W}_k;\xi_k)-\nabla f(\bar{W}_k))\odot F_p(P_k)\right\|^2\right]$$

$$+ 3\alpha^2 L\gamma^2\mathbb{E}_{\xi_k}\left[\left\|(\nabla f(\bar{W}_k;\xi_k)-\nabla f(\bar{W}_k))\odot F_p(P_k)\right\|^2\right] + \frac{3\gamma^2 L}{2}\mathbb{E}_{\xi_k}[\|Q_{k+1}-Q_k\|^2]$$

$$\leq \frac{3L}{2}\mathbb{E}_{\xi_k}[\|W_{k+1}-W_k\|^2] + 3\gamma^2 L\mathbb{E}_{\xi_k,b_k}\left[\left\|P_{k+1}-P_k + \alpha(\nabla f(\bar{W}_k;\xi_k)-\nabla f(\bar{W}_k))\odot F_p(P_k)-b_k\right\|^2\right] + 3\alpha^2\gamma^2 L q_{\max}^2\sigma^2$$

$$+ \frac{3\gamma^2 L}{2}\mathbb{E}_{\xi_k}[\|Q_{k+1}-Q_k\|^2] + \Theta\left(3\gamma^2 L\alpha\Delta w_{\min}\right)$$

where the last inequality comes from Assumption 3.4 and the bounded variance assumption (Assumption 3.2)

$$3\alpha^2\gamma^2 L\mathbb{E}_{\xi_k}\left[\left\|(\nabla f(\bar{W}_k;\xi_k)-\nabla f(\bar{W}_k))\odot F_p(P_k)\right\|^2\right] \tag{84}$$

$$\leq 3\alpha^2\gamma^2 L q_{\max}^2\mathbb{E}_{\xi_k}\left[\left\|\nabla f(\bar{W}_k;\xi_k)-\nabla f(\bar{W}_k)\right\|^2\right] \leq 3\alpha^2\gamma^2 L q_{\max}^2\sigma^2.$$

**Bound of the fifth term (d).** By Young's inequality, we have

$$-\gamma\mathbb{E}_{\xi_k,b_k}[\langle\nabla f(\bar{W}_k), Q_{k+1}-Q_k\rangle] \leq \frac{\alpha\gamma}{8q_{\max}}\|\nabla f(\bar{W}_k)\|_{M_p(P_k)}^2 + \frac{2q_{\max}\gamma}{\alpha}\mathbb{E}_{\xi_k}[\|Q_{k+1}-Q_k\|_{M_p(P_k)^\dagger}^2]. \tag{85}$$

**Combination of the upper bound** (a)**,** (b)**,** (c) **and** (d)**.** Plugging (81), (82), (83) into (74) and letting $\alpha \leq \frac{1}{2\gamma L q_{\max}}$,

$$\mathbb{E}_{\xi_k,b_k}[f(\bar{W}_{k+1})] \leq f(\bar{W}_k) - \frac{\alpha\gamma}{8q_{\max}}\|\nabla f(\bar{W}_k)\|_{M_p(P_k)}^2 + \frac{\alpha\gamma\sigma^2}{2}\left\|\frac{G_p(P_k)}{\sqrt{F_p(P_k)}}\right\|_\infty^2 + \Theta\left(\frac{\gamma\Delta w_{\min}}{q_{\min}}\right)$$

$$+ \frac{2q_{\max}\gamma}{\alpha}\mathbb{E}_{\xi_k}\left[\|Q_{k+1}-Q_k\|_{M_p(P_k)^\dagger}^2\right] + \frac{3\gamma^2 L}{2}\mathbb{E}_{\xi_k}\left[\|Q_{k+1}-Q_k\|^2\right] \tag{86}$$

$$-\left(\frac{\gamma}{2\alpha q_{\max}} - 3\gamma^2 L\right)\mathbb{E}_{\xi_k,b_k}\left[\left\|P_{k+1}-P_k + \alpha(\nabla f(\bar{W}_k;\xi_k)-\nabla f(\bar{W}_k))\odot F_p(P_k)-b_k\right\|^2\right]$$

$$+ 3\alpha^2\gamma^2 L q_{\max}^2 \sigma^2 + \frac{q_{\max}}{\alpha\gamma} \mathbb{E}_{\xi_k} \left[ \|W_{k+1} - W_k\|_{M_p(P_k)^\dagger}^2 \right] + \frac{3L}{2} \mathbb{E}_{\xi_k}[\|W_{k+1} - W_k\|^2].$$

Now the proof of (51) is completed. Leveraging the $L$-smooth assumption (Assumption 3.1), we have

$$\mathbb{E}_{\xi_k, b_k}[f(W_k + \gamma(P_{k+1} - Q_k))] = f(\bar{W}_k) + \gamma\mathbb{E}_{\xi_k, b_k}[\langle \nabla f(\bar{W}_k), P_{k+1} - P_k \rangle] + \frac{L\gamma^2}{2}\mathbb{E}_{\xi_k, b_k}[\|P_{k+1} - P_k\|^2]. \quad (87)$$

Plugging (81) and (83) into (87), we have

$$\mathbb{E}_{\xi_k, b_k}[f(W_k + \gamma P_{k+1} - \gamma Q_k)] \quad (88)$$

$$\leq f(\bar{W}_k) - \frac{\alpha\gamma}{2q_{\max}}\|\nabla f(\bar{W}_k)\|_{M_p(P_k)}^2 + \frac{\alpha\gamma\sigma^2}{2}\left\|\frac{G_p(P_k)}{\sqrt{F_p(P_k)}}\right\|_\infty^2 + \Theta\left(\frac{\gamma\Delta w_{\min}}{q_{\min}}\right)$$

$$- \left(\frac{\gamma}{2\alpha q_{\max}} - \gamma^2 L\right) \mathbb{E}_{\xi_k, b_k}\left[\|P_{k+1} - P_k + \alpha(\nabla f(\bar{W}_k; \xi_k) - \nabla f(\bar{W}_k)) \odot F_p(P_k) - b_k\|^2\right] + \alpha^2\gamma^2 L q_{\max}^2 \sigma^2.$$

$$\leq f(\bar{W}_k) - \frac{\alpha\gamma}{2q_{\max}}\|\nabla f(\bar{W}_k)\|_{M_p(P_k)}^2 + \frac{\alpha\gamma\sigma^2}{2}\left\|\frac{G_p(P_k)}{\sqrt{F_p(P_k)}}\right\|_\infty^2 + \alpha^2\gamma^2 L q_{\max}^2 \sigma^2 + \Theta\left(\frac{\gamma\Delta w_{\min}}{q_{\min}}\right)$$

where the second inequality holds by $\alpha \leq \frac{1}{2\gamma L q_{\max}}$. Now the proof of (52) is completed. $\square$

### E.3. Proof of Lemma E.2: Descent of sequence $W_k$

**Lemma E.2** (Descent lemma of upper-level problem). *Under Assumption 2.1, it holds that*

$$\mathbb{E}_{b_k'}\left[\|P^*(W_{k+1}, Q_{k+1}) - Q_{k+1}\|^2\right] \leq \|P^*(W_k, Q_k) - Q_k\|^2 - \frac{\beta}{2\gamma q_{\max}}\|P^*(W_k, Q_k) - Q_k\|_{M_w(W_k)}^2 + \Theta\left(\frac{\Delta w_{\min}}{\gamma q_{\max}}\right)$$

$$- \frac{\beta}{2\gamma q_{\max}}\left\|(P^*(W_k, Q_k) - Q_k) \odot F_w(W_k) - |P^*(W_k, Q_k) - Q_k| \odot G_w(W_k)\right\|^2$$

$$+ \frac{2\beta q_{\max}^3}{\gamma}\|P_{k+1} - P^*(W_k, Q_k)\|_{M_w(W_k)^\dagger}^2 + \frac{2\beta^2}{\gamma^2}\|P_{k+1} - P^*(W_k, Q_k)\|^2. \quad (53)$$

*Lemma E.2.* Recall the definition $P^*(W, Q) - Q = (W^* - W)/\gamma$, it holds that

$$\|P^*(W_{k+1}, Q_{k+1}) - Q_{k+1}\|^2 = \frac{1}{\gamma^2}\|W_{k+1} - \text{Proj}_{\mathcal{W}^*}(W_{k+1})\|^2 \leq \frac{1}{\gamma^2}\|W_{k+1} - \text{Proj}_{\mathcal{W}^*}(W_k)\|^2 \quad (89)$$

$$= \frac{1}{\gamma^2}\|W_k - \text{Proj}_{\mathcal{W}^*}(W_k)\|^2 + \frac{2}{\gamma^2}\langle W_k - \text{Proj}_{\mathcal{W}^*}(W_k), W_{k+1} - W_k \rangle + \frac{1}{\gamma^2}\|W_{k+1} - W_k\|^2$$

$$= \|P^*(W_k, Q_k) - Q_k\|^2 - \frac{2}{\gamma}\langle P^*(W_k, Q_k) - Q_k, W_{k+1} - W_k \rangle + \frac{1}{\gamma^2}\|W_{k+1} - W_k\|^2$$

where the first inequality comes from the fact that $\text{Proj}_{\mathcal{W}^*}(W_{k+1})$ is the closest point to $W_{k+1}$ in $\mathcal{W}^*$. We bound the second term in the RHS of (89) by

$$- \mathbb{E}_{b_k'}\left[\frac{2}{\gamma}\langle P^*(W_k, Q_k) - Q_k, W_{k+1} - W_k \rangle\right] \quad (90)$$

$$= -\frac{2}{\gamma}\langle P^*(W_k, Q_k) - Q_k, \beta(P_{k+1} - Q_k) \odot F_w(W_k) - \beta|P_{k+1} - Q_k| \odot G_w(W_k)\rangle$$

$$= -\frac{2\beta}{\gamma}\langle P^*(W_k, Q_k) - Q_k, (P^*(W_k, Q_k) - Q_k) \odot F_w(W_k) - |P^*(W_k, Q_k) - Q_k| \odot G_w(W_k)\rangle$$

$$- \frac{2\beta}{\gamma}\langle P^*(W_k, Q_k) - Q_k, (P_{k+1} - Q_k) \odot F_w(W_k) - |P_{k+1} - Q_k| \odot G_w(W_k)$$

$$- ((P^*(W_k, Q_k) - Q_k) \odot F_w(W_k) - |P^*(W_k, Q_k) - Q_k| \odot G_w(W_k))\rangle.$$

where the first equality holds because $\mathbb{E}[b'_k] = 0$.

The first term in the RHS of (90) is bounded by

$$-\frac{2\beta}{\gamma}\langle P^*(W_k, Q_k) - Q_k, (P^*(W_k, Q_k) - Q_k) \odot F_w(W_k) - |P^*(W_k, Q_k) - Q_k| \odot G_w(W_k)\rangle \tag{91}$$

$$= -\frac{2\beta}{\gamma}\left\langle (P^*(W_k, Q_k) - Q_k) \odot \sqrt{F_w(W_k)}, \frac{(P^*(W_k, Q_k) - Q_k) \odot F_w(W_k) - |P^*(W_k, Q_k) - Q_k| \odot G_w(W_k)}{\sqrt{F_w(W_k)}}\right\rangle$$

$$\overset{(a)}{=} -\frac{\beta}{\gamma}\|(P^*(W_k, Q_k) - Q_k) \odot \sqrt{F_w(W_k)}\|^2 + \frac{\beta}{\gamma}\left\||P^*(W_k, Q_k) - Q_k| \odot \frac{G_w(W_k)}{\sqrt{F_w(W_k)}}\right\|^2$$

$$-\frac{\beta}{\gamma}\left\|(P^*(W_k, Q_k) - Q_k) \odot \sqrt{F_w(W_k)} + |P^*(W_k, Q_k) - Q_k| \odot \frac{G_w(W_k)}{\sqrt{F_w(W_k)}}\right\|^2$$

$$\overset{(b)}{\leq} -\frac{\beta}{\gamma q_{\max}}\|P^*(W_k, Q_k) - Q_k\|^2_{M_w(W_k)} - \frac{\beta}{\gamma}\left\|(P^*(W_k, Q_k) - Q_k) \odot \sqrt{F_w(W_k)} + |P^*(W_k, Q_k) - Q_k| \odot \frac{G_w(W_k)}{\sqrt{F_w(W_k)}}\right\|^2$$

$$\leq -\frac{\beta}{\gamma q_{\max}}\|P^*(W_k, Q_k) - Q_k\|^2_{M_w(W_k)} - \frac{\beta}{\gamma q_{\max}}\|(P^*(W_k, Q_k) - Q_k) \odot F_w(W_k) - |P^*(W_k, Q_k) - Q_k| \odot G_w(W_k)\|^2$$

where $(a)$ leverages $2\langle U, V\rangle = \|U\|^2 - \|V\|^2 - \|U - V\|^2$ for any $U, V \in \mathbb{R}^D$, $(b)$ is achieved by

$$-\frac{\beta}{2}\|(P^*(W_k, Q_k) - Q_k) \odot \sqrt{F_w(W_k)}\|^2 + \frac{\beta}{2}\left\||P^*(W_k, Q_k) - Q_k| \odot \frac{G_w(W_k)}{\sqrt{F_w(W_k)}}\right\|^2 \tag{92}$$

$$= -\frac{\beta}{2}\sum_{d\in[D]}\left([P^*(W_k, Q_k) - Q_k]_d^2\left([F_w(W_k)]_d - \frac{[G_w(W_k)]_d^2}{[F_w(W_k)]_d}\right)\right)$$

$$= -\frac{\beta}{2}\sum_{d\in[D]}\left([P^*(W_k, Q_k) - Q_k]_d^2\left(\frac{[F_w(W_k)]_d^2 - [G_w(W_k)]_d^2}{[F_w(W_k)]_d}\right)\right)$$

$$\leq -\frac{\beta}{2q_{\max}}\sum_{d\in[D]}\left([P^*(W_k, Q_k) - Q_k]_d^2\left([F_w(W_k)]_d^2 - [G_w(W_k)]_d^2\right)\right) = -\frac{\beta}{2q_{\max}}\|P^*(W_k, Q_k) - Q_k\|^2_{M_w(W_k)}.$$

The second term in the RHS of (90) follows the Lipschitz continuity of analog update (see Lemma A.2)

$$-\frac{2\beta}{\gamma}\langle P^*(W_k, Q_k) - Q_k, (P_{k+1} - Q_k) \odot F_w(W_k) - |P_{k+1} - Q_k| \odot G_w(W_k) \tag{93}$$

$$-((P^*(W_k, Q_k) - Q_k) \odot F_w(W_k) - |P^*(W_k, Q_k) - Q_k| \odot G_w(W_k))\rangle$$

$$= \frac{2\beta}{\gamma}\langle P^*(W_k, Q_k) - Q_k, -(P_{k+1} - Q_k) \odot F_w(W_k) + |P_{k+1} - Q_k| \odot G_w(W_k)$$

$$+((P^*(W_k, Q_k) - Q_k) \odot F_w(W_k) - |P^*(W_k, Q_k) - Q_k| \odot G_w(W_k))\rangle$$

$$\leq \frac{\beta}{2\gamma q_{\max}}\|P^*(W_k, Q_k) - Q_k\|^2_{M_w(W_k)} + \frac{2\beta q_{\max}}{\gamma} \times \Big\|(P_{k+1} - Q_k) \odot F_w(W_k)$$

$$-|P_{k+1} - Q_k| \odot G_w(W_k) - ((P^*(W_k, Q_k) - Q_k) \odot F_w(W_k) - |P^*(W_k, Q_k) - Q_k| \odot G_w(W_k))\Big\|^2_{M_w(W_k)^\dagger}.$$

$$\leq \frac{\beta}{2\gamma q_{\max}}\|P^*(W_k, Q_k) - Q_k\|^2_{M_w(W_k)} + \frac{2\beta q_{\max}^3}{\gamma}\|P_{k+1} - P^*(W_k, Q_k)\|^2_{M_w(W_k)^\dagger}.$$

Substituting (91) and (93) into (90), we bound the second term in the RHS of (89) by

$$-\frac{2}{\gamma}\mathbb{E}_{b'_k}[\langle P^*(W_k, Q_k) - Q_k, W_{k+1} - W_k\rangle] \tag{94}$$

$$\leq -\frac{\beta}{2\gamma q_{\max}}\|P^*(W_k, Q_k) - Q_k\|^2_{M_w(W_k)} - \frac{\beta}{\gamma q_{\max}}\|(P^*(W_k, Q_k) - Q_k) \odot F_w(W_k) - |P^*(W_k, Q_k) - Q_k| \odot G_w(W_k)\|^2$$

$$+ \frac{2\beta q_{\max}^3}{\gamma} \|P_{k+1} - P^*(W_k, Q_k)\|_{M_w(W_k)^\dagger}^2 .$$

The third term in the RHS of (89) follows the Lipschitz continuity of the analog update (Lemma A.2)

$$\frac{1}{\gamma^2} \mathbb{E}_{b_k'} \left[ \|W_{k+1} - W_k\|^2 \right] = \frac{\beta^2}{\gamma^2} \|(P_{k+1} - Q_k) \odot F_w(W_k) - |P_{k+1} - Q_k| \odot G_w(W_k)\|^2 + \Theta\left( \frac{\beta \Delta w_{\min}}{\gamma^2} \right)$$

$$\leq \frac{2\beta^2}{\gamma^2} \|(P^*(W_k, Q_k) - Q_k) \odot F_w(W_k) - |P^*(W_k, Q_k) - Q_k| \odot G_w(W_k)\|^2$$

$$+ \frac{2\beta^2}{\gamma^2} \big\| (P_{k+1} - Q_k) \odot F_w(W_k) - |P_{k+1} - Q_k| \odot G_w(W_k)$$

$$- \big( (P^*(W_k, Q_k) - Q_k) \odot F_w(W_k) - |P^*(W_k, Q_k) - Q_k| \odot G_w(W_k) \big) \big\|^2 + \Theta\left( \frac{\beta \Delta w_{\min}}{\gamma^2} \right)$$

$$\leq \frac{2\beta^2}{\gamma^2} \|(P^*(W_k, Q_k) - Q_k) \odot F_w(W_k) - |P^*(W_k, Q_k) - Q_k| \odot G_w(W_k)\|^2 + \frac{2\beta^2}{\gamma^2} \|P_{k+1} - P^*(W_k, Q_k)\|^2$$

$$+ \Theta\left( \frac{\beta \Delta w_{\min}}{\gamma^2} \right) \tag{95}$$

Plugging (94) and (95) into (89) yields

$$\mathbb{E}_{b_k'} \left[ \|P^*(W_{k+1}, Q_{k+1}) - Q_{k+1}\|^2 \right] \leq \|P^*(W_k, Q_k) - Q_k\|^2 - \frac{\beta}{2\gamma q_{\max}} \|P^*(W_k, Q_k) - Q_k\|_{M_w(W_k)}^2 + \Theta\left( \frac{\beta \Delta w_{\min}}{\gamma^2} \right)$$

$$- \left( \frac{\beta}{\gamma q_{\max}} - \frac{2\beta^2}{\gamma^2} \right) \|(P^*(W_k, Q_k) - Q_k) \odot F_w(W_k) - |P^*(W_k, Q_k) - Q_k| \odot G_w(W_k)\|^2$$

$$+ \frac{2\beta q_{\max}^3}{\gamma} \|P_{k+1} - P^*(W_k, Q_k)\|_{M_w(W_k)^\dagger}^2 + \frac{2\beta^2}{\gamma^2} \|P_{k+1} - P^*(W_k, Q_k)\|^2. \tag{96}$$

Notice the learning rate $\beta$ is chosen as $\beta \leq \frac{\gamma}{2q_{\max}}$, we have

$$\mathbb{E}_{b_k'} \left[ \|P^*(W_{k+1}, Q_{k+1}) - Q_{k+1}\|^2 \right] \leq \|P^*(W_k, Q_k) - Q_k\|^2 - \frac{\beta}{2\gamma q_{\max}} \|P^*(W_k, Q_k) - Q_k\|_{M_w(W_k)}^2 + \Theta\left( \frac{\Delta w_{\min}}{\gamma q_{\max}} \right)$$

$$- \frac{\beta}{2\gamma q_{\max}} \|(P^*(W_k, Q_k) - Q_k) \odot F_w(W_k) - |P^*(W_k, Q_k) - Q_k| \odot G_w(W_k)\|^2$$

$$+ \frac{2\beta q_{\max}^3}{\gamma} \|P_{k+1} - P^*(W_k, Q_k)\|_{M_w(W_k)^\dagger}^2 + \frac{2\beta^2}{\gamma^2} \|P_{k+1} - P^*(W_k, Q_k)\|^2 \tag{97}$$

which completes the proof. $\qquad\square$

### E.4. Proof of Lemma E.3: Descent of accumulated asymmetric sequence $\varphi(P_k)$

In this section, we formally define the accumulated asymmetric function $\varphi(P) : \mathbb{R}^D \to \mathbb{R}^D$ element-wise defined by

$$[\varphi(P)]_d := \int_{\tau_i^{\min}}^{[P]_d} [G_p(P)]_d \, \mathrm{d}[P]_d. \tag{98}$$

**Lemma E.3** (Descent lemma of accumulated asymmetric sequence). *Under Assumption 2.1 and 3.6, it holds that*

$$\mathbb{E}_{\xi_k, b_k}[\varphi(P_{k+1})] \leq \varphi(P_k) - \frac{\alpha C_\star}{2} \|G_p(P_k)\|^2 + \alpha^2 L q_{\max}^2 \sigma^2 + \frac{\alpha q_{\max}^2}{2C_\star} \|\nabla f(\bar{W}_k)\|^2 + \Theta\left( \frac{\Delta w_{\min}}{\gamma q_{\max}} \right)$$

$$+ L\mathbb{E}_{\xi_k, b_k} \left[ \left\| P_{k+1} - P_k + \alpha(\nabla f(\bar{W}_k; \xi_k) - \nabla f(\bar{W}_k)) \odot F_p(P_k) - b_k \right\|^2 \right]. \tag{54}$$

*Lemma E.3.* The Lipschitz continuity of the generic response function ensures that the accumulated asymmetric function $\varphi(P)$ is $L$-smooth.

$$\mathbb{E}_{\xi_k, b_k}[\varphi(P_{k+1})] \leq \varphi(P_k) + \underbrace{\mathbb{E}_{\xi_k, b_k}[\langle G_p(P_k), P_{k+1} - P_k \rangle]}_{(a)} + \underbrace{\frac{L}{2}\mathbb{E}_{\xi_k, b_k}[\|P_{k+1} - P_k\|^2]}_{(b)} \tag{99}$$

Next, we will handle each term in the RHS of (99) separately.

**Bound of the second term (a).**

$$\mathbb{E}_{\xi_k, b_k}\left[\langle G_p(P_k), P_{k+1} - P_k \rangle\right]$$

$$=\mathbb{E}_{\xi_k}\left[\langle G_p(P_k), -\alpha\nabla f(\bar{W}_k, \xi_k) \odot F_p(P_k) - \alpha|\nabla f(\bar{W}_k, \xi_k)| \odot G_p(P_k)\rangle\right]$$

$$=\langle G_p(P_k), -\alpha\nabla f(\bar{W}_k) \odot F_p(P_k) - \alpha\mathbb{E}_{\xi_k}[|\nabla f(\bar{W}_k, \xi_k)|] \odot G_p(P_k)\rangle$$

$$=\left\langle \sqrt{\alpha\mathbb{E}_{\xi_k}[|\nabla f(\bar{W}_k, \xi_k)|]} \odot G_p(P_k), -\frac{\sqrt{\alpha}}{\sqrt{\mathbb{E}_{\xi_k}[|\nabla f(\bar{W}_k, \xi_k)|]}}\nabla f(\bar{W}_k) \odot F_p(P_k) - \sqrt{\alpha\mathbb{E}_{\xi_k}[|\nabla f(\bar{W}_k, \xi_k)|]} \odot G_p(P_k) \right\rangle$$

$$\overset{(a)}{=} -\frac{1}{2}\left\|\sqrt{\alpha\mathbb{E}_{\xi_k}[|\nabla f(\bar{W}_k, \xi_k)|]} \odot G_p(P_k)\right\|^2 + \frac{1}{2}\left\|\frac{\sqrt{\alpha}}{\sqrt{\mathbb{E}_{\xi_k}[|\nabla f(\bar{W}_k, \xi_k)|]}}\nabla f(\bar{W}_k) \odot F_p(P_k)\right\|^2$$

$$\qquad -\frac{1}{2}\left\|\frac{\sqrt{\alpha}}{\sqrt{\mathbb{E}_{\xi_k}[|\nabla f(\bar{W}_k, \xi_k)|]}}\nabla f(\bar{W}_k) \odot F_p(P_k) + \sqrt{\alpha\mathbb{E}_{\xi_k}[|\nabla f(\bar{W}_k, \xi_k)|]} \odot G_p(P_k)\right\|^2$$

$$\leq -\frac{\alpha}{2}\sum_{d\in[D]}\mathbb{E}_{\xi_k}\left[[|\nabla f(\bar{W}_k, \xi_k)|]_d\right][G_p(P_k)]_d^2 + \frac{\alpha}{2}\sum_{d\in[D]}\frac{[|\nabla f(\bar{W}_k)|]_d^2}{\mathbb{E}_{\xi_k}\left[[|\nabla f(\bar{W}_k, \xi_k)|]_d\right]}[F_p(P_k)]_d^2$$

$$\overset{(b)}{\leq} -\frac{\alpha C_\star}{2}\|G_p(P_k)\|^2 + \frac{\alpha}{2C_\star}\|\nabla f(\bar{W}_k) \odot F_p(P_k)\|^2$$

$$\leq -\frac{\alpha C_\star}{2}\|G_p(P_k)\|^2 + \frac{\alpha q_{\max}^2}{2C_\star}\|\nabla f(\bar{W}_k)\|^2. \tag{100}$$

The equality (a) holds by $\langle A, -B\rangle = -\frac{\|A\|^2}{2} - \frac{\|B\|^2}{2} + \frac{\|A-B\|^2}{2}$, the equality (b) holds by Assumption 3.6.

**Bound the third term (b).** Follow (83), we have:

$$\frac{L}{2}\mathbb{E}_{\xi_k, b_k}[\|P_{k+1} - P_k\|^2]$$

$$\leq L\mathbb{E}_{\xi_k, b_k}\left[\left\|P_{k+1} - P_k + \alpha(\nabla f(\bar{W}_k; \xi_k) - \nabla f(\bar{W}_k)) \odot F_p(P_k) - b_k\right\|^2\right] + \alpha^2 L q_{\max}^2 \sigma^2 + \Theta\left(\alpha L \Delta w_{\min}\right). \tag{101}$$

Substituting (100) and (101) into (99) and note that $\alpha \leq \frac{1}{2\gamma L q_{\max}}$, we have:

$$\mathbb{E}_{\xi_k, b_k}[\varphi(P_{k+1})] \leq \varphi(P_k) - \frac{\alpha C_\star}{2}\|G_p(P_k)\|^2 + \alpha^2 L q_{\max}^2 \sigma^2 + \frac{\alpha q_{\max}^2}{2C_\star}\|\nabla f(\bar{W}_k)\|^2 + \Theta\left(\frac{\Delta w_{\min}}{\gamma q_{\max}}\right)$$

$$\qquad + L\mathbb{E}_{\xi_k, b_k}\left[\left\|P_{k+1} - P_k + \alpha(\nabla f(\bar{W}_k; \xi_k) - \nabla f(\bar{W}_k)) \odot F_p(P_k) - b_k\right\|^2\right]. \tag{102}$$

$\square$

# F. Additional Experiments and Experimental Setups

### F.1. Device implementation details

In our experiments, we use two ReRAM array device presets from IBM AIHWKit simulator (Rasch et al., 2021): the HfO$_2$-based RRAM preset and the ReRamArrayOM preset in (Gong et al., 2022b). Both presets inherit from the `SoftBoundsReferenceDevice` model with potentiation and depression response functions as

$$q_+(w) = \alpha_+\left(1 - \frac{w}{\tau_{\max}}\right), \qquad q_-(w) = \alpha_-\left(1 + \frac{w}{\tau_{\min}}\right), \tag{103}$$

*Table 3.* Summary of the device parameters for the two HfO$_2$-based ReRAM models.

| Device model | $(\tau_{\min}, \tau_{\max})$ | $\Delta w_{\min}$ | d2d variation $\sigma_\pm$ | c2c variation $\sigma_{\text{c2c}}$ |
|---|---|---|---|---|
| HfO$_2$-based ReRAM model (Gong et al., 2022b) | $(-1.0, 1.0)$ | 0.4622 | 0.7125 | 0.2174 |
| ReRamArrayOMPresetDevice (Gong et al., 2022b) | $(-1.0, 1.0)$ | 0.0949 | 0.7829 | 0.4158 |

where $\alpha_+, \alpha_- > 0$ determine the effective update magnitudes for potentiation and depression, respectively. Following the SoftBounds-reference parameterization, we decompose these slopes as

$$\alpha_+ = \gamma + \rho, \qquad \alpha_- = \gamma - \rho, \tag{104}$$

where $\gamma$ controls the common slope magnitude and $\rho$ controls the up/down asymmetry. In our model, these quantities are sampled independently for each cross-point as

$$\gamma_{ij} = \exp\left(\sigma_{\text{d2d}}\xi_{ij}^{(\gamma)}\right), \qquad \rho_{ij} = \sigma_\pm \xi_{ij}^{(\rho)}, \tag{105}$$

where $\xi_{ij}^{(\gamma)}, \xi_{ij}^{(\rho)} \sim \mathcal{N}(0,1)$. Thus, $\sigma_{\text{d2d}}$ controls device-to-device variation in the slope magnitude, while $\sigma_\pm$ controls device-to-device variation in the potentiation/depression asymmetry.

At the array level, the response functions are applied elementwise. For a weight matrix $W$, we define

$$\left[Q_+(W)\right]_{ij} = q_{+,ij}(W_{ij}) = \alpha_{+,ij}\left(1 - \frac{W_{ij}}{\tau_{\max}}\right), \tag{106}$$

$$\left[Q_-(W)\right]_{ij} = q_{-,ij}(W_{ij}) = \alpha_{-,ij}\left(1 + \frac{W_{ij}}{\tau_{\min}}\right) \tag{107}$$

where each cross-point $(i, j)$ has its own sampled device parameters $(\gamma_{ij}, \rho_{ij})$.

Besides, IBM AIHWKit simulator (Rasch et al., 2021) also provides the cycle-to-cycle variation, which is modeled separately as a fresh random perturbation at each update pulse. For example, the scalar update response at cycle $k$ can be written as

$$\Delta w_{ij,k}^+ = \Delta w_{\min} q_{+,ij}(w_{ij,k})\left(1 + \sigma_{\text{c2c}}\zeta_{ij,k}^+\right), \tag{108}$$

$$\Delta w_{ij,k}^- = -\Delta w_{\min} q_{-,ij}(w_{ij,k})\left(1 + \sigma_{\text{c2c}}\zeta_{ij,k}^-\right), \tag{109}$$

where $\zeta_{ij,k}^+, \zeta_{ij,k}^- \sim \mathcal{N}(0,1)$ are sampled independently across devices and update cycles.

Under this model, there exists a unique SP at which the average positive and negative update sizes become equal; this weight is the SP $w^\diamond$. The ground truth value for SP of the device can be computed as:

$$w^\diamond = \frac{\alpha_+ - \alpha_-}{\frac{\alpha_+}{\tau_{\max}} - \frac{\alpha_-}{\tau_{\min}}} = \frac{2\rho}{\frac{\gamma+\rho}{\tau_{\max}} - \frac{\gamma-\rho}{\tau_{\min}}} \tag{110}$$

We evaluate two HfO$_2$-based ReRAM device models, whose non-idealities include device-to-device (d2d) and cycle-to-cycle (c2c) variation, update asymmetry, and limited conductance states, as summarized in Table 3.

### F.2. Details of implementation for Figure 1 and Figure 2

In practice, however, the SP of an analog device is usually determined experimentally by applying alternating positive and negative update pulses until the weight converges. We denote by $r(N)$ the SP estimated in this way after $N$ alternating pulses. To study how well this procedure recovers the true SP, we run simulations on the same $512 \times 512$ array with different pulse budgets $N \in \{500, 1000, 2000, 4000, 8000\}$, using a SoftBounds-based RPU preset with 2000 states. Figure 1a shows the empirical offsets of the SP mean and standard deviation over the $512 \times 512$ array, defined as the ground-truth statistics minus the corresponding estimated statistics. Here, the estimated SP statistics are computed by first estimating the SP for each array element, and then taking the mean and standard deviation across the $512 \times 512$ per-element SP estimates.

*Table 4.* Hyperparameters for TT-v2, AGAD, and E-RIDER used in the analog training runs for Table 1.

| Hyperparameter | TT-v2 | AGAD | E-RIDER |
|---|---|---|---|
| Units in mini-batch | \ | True | True |
| Transfer frequency (`transfer_every`) | 1.0 | 1.0 | 1.0 |
| Transfer columns (`transfer_columns`) | True | True | True |
| Self-transfer (`no_self_transfer`) | True | True | True |
| Reads per transfer (`n_reads_per_transfer`) | 1 | 1 | 1 |
| Input chopping (`in_chop_prob`) | \ | 0.1 | 0.05 |
| Input chopper random (`in_chop_random`) | \ | False | False |
| Output chopping (`out_chop_prob`) | \ | 0.0 | 0.0 |
| Fast learning rate (`fast_lr`) | 0.005 | 0.01 | 0.5 |
| Scale fast LR (`scale_fast_lr`) | \ | False | False |
| Transfer learning rate (`transfer_lr`) | 0.005 | 0.2 | 0.05 |
| Scale transfer LR (`scale_transfer_lr`) | True | True | False |
| Auto granularity (`auto_granularity`) | \ | 1000 | 1000 |
| Auto scale (`auto_scale`) | \ | False | False |
| Auto momentum (`auto_momentum`) | \ | 0.99 | 0.99 |
| Momentum (`momentum`) | 0.1 | 0.0 | 0.0 |
| Forget buffer (`forget_buffer`) | True | True | True |
| Tail weighting (`tail_weightening`) | \ | 5.0 | 5.0 |
| Threshold scale (`thres_scale`) | 0.8 | \ | \ |
| Gamma (`gamma`) | \ | \ | 0.1 |

As $N$ increases, the distribution of $r(N)$ gradually moves towards and eventually almost coincides with the ground-truth distribution of $w^\diamond$. For smaller pulse counts, however, both the mean and the standard deviation of $r$ can deviate significantly from those of $w^\diamond$. We model this mismatch element-wise using the following stochastic offset model:

$$r = w^\diamond + \mu_r + \sigma_r \xi, \qquad \xi \sim \mathcal{N}(0, 1), \tag{111}$$

where $\mu_r$ captures the systematic offsets between the means of the two distributions and $\sigma_r$ accounts for the variance offsets introduced by using a finite number of pulses. The combined term $o_r = \mu_r + \sigma_r \xi$ represents the residual offset on the reference device after SP subtraction. Our experiments in Figures 1a demonstrate that when the number of alternating pulses is not sufficiently large, this offset $o_r$ introduces a non-negligible error in the estimated SP and the subsequent training. This effect is clearly visible in Figure 2, where we train a LeNet-5(MNIST) with TT-v1 using SPs estimated from different numbers of pulses: smaller $N$ leads to significant degradation and even failure to converge.

Furthermore, the pulse cost required for accurate SP estimation increases as the device update granularity $\Delta w_{\min}$ improves. In Figure 1b, we sweep $\Delta w_{\min}$ from $5 \times 10^{-3}$ down to $1.6 \times 10^{-6}$ and, for each setting, estimate the SP using alternating pulses with a discrete pulse-budget schedule $N \in \{200, 500, 10^3, 2 \times 10^3, \ldots, 8.192 \times 10^6\}$. We then report the smallest $N$ such that the relative error of the estimated SP mean is within $1\%$ of the ground-truth mean. The results show that higher-precision devices require substantially more pulses to reach the same accuracy target. Consequently, determining the SP by alternating pulses exhibits an inherent trade-off between pulse overhead and accuracy.

## F.3. Hyperparameter settings for Tables 1 and 2

We report the hyperparameter settings used in our MNIST experiments with fully analog LeNet-5 and FCN models and the IO/analog readout noise configuration used in both MNIST and CIFAR 100 tasks. We use a fully analog LeNet-5–style CNN with two $5 \times 5$ convolution layers with 16 and 32 channels and two analog fully connected layers of sizes 512 and 128. The network uses $\tanh$ activations in the hidden layers. For the fully connected network, we set the input dimension of 784, two hidden layers of sizes 256 and 128, and a 10-class output layer. The model uses sigmoid activations in the hidden layers. For LeNet-5, we use a batch size of 8, while for the FCN we use a batch size of 10.

We focus our hyperparameter search on the learning-rate-related parameters, auto-granularity (threshold scale), and the input chopping probability. Consequently, the main distinctions between TT-v2, AGAD and E-RIDER are reflected in these tuned parameters. In addition, E-RIDER includes an extra residual scaling parameter $\gamma$, which controls the contribution of the residual term during gradient computation and is absent in AGAD. For the global learning rate on the LeNet model, we

*Table 5.* Hyperparameters for baseline algorithms and E-RIDER used in the CIFAR-100 fine-tuning experiments.

| Category | TT-v2 | AGAD | E-RIDER |
|---|---|---|---|
| Global learning rate (`lr`) | 0.15 | 0.15 | 0.2 |
| Fast learning rate (`fast_lr`) | 0.01 | 0.01 | 0.1 |
| Transfer learning rate (`transfer_lr`) | 0.5 | 0.5 | 0.01 |
| Scale transfer LR (`scale_transfer_lr`) | True | True | True |
| Momentum (`momentum`) | 0.1 | 0.0 | 0.0 |
| Threshold scale (`thres_scale`) | 1.0 | \ | \ |
| Input chopping (`in_chop_prob`) | \ | 0.1 | 0.05 |
| Output chopping (`out_chop_prob`) | \ | 0.0 | 0.0 |
| Auto granularity (`auto_granularity`) | \ | 1000 | 1000 |
| Gamma (`gamma`) | \ | \ | 0.1 |

*Table 6.* Hyperparameters for TT-v2, AGAD, and E-RIDER used in the analog training runs for Table 2.

| Category | TT-v2 | AGAD | E-RIDER |
|---|---|---|---|
| Input chopping (`in_chop_prob`) | \ | 0.1 | 0.05 |
| Input chopper random (`in_chop_random`) | \ | False | False |
| Output chopping (`out_chop_prob`) | \ | 0.0 | 0.0 |
| Fast learning rate (`fast_lr`) | 0.03 | 0.05 | 0.5 |
| Scale fast LR (`scale_fast_lr`) | \ | False | False |
| Transfer learning rate (`transfer_lr`) | 0.01 | 0.3 | 0.05 |
| Scale transfer LR (`scale_transfer_lr`) | True | True | False |
| Threshold scale (`thres_scale`) | 0.8 | \ | \ |
| Gamma (`gamma`) | \ | \ | 0.1 |

set TT-v2 to 0.005, and AGAD and E-RIDER to 0.05. Other parameters are kept identical across methods unless stated otherwise. It is worth noting that we use a very small learning rate for TT-v2, because with a small number of states and a large reference-point offset, training becomes unstable and can diverge after several epochs when a larger learning rate is used. Despite the smaller learning rate, the accuracy we report for TT-v2 is measured after training has converged. The complete set of hyperparameter is reported in Table 4 for Lenet-5 and Table 6 for FCN.

Unless otherwise stated, TT-v2, AGAD and E-RIDER share the same IO and readout noise settings across both architectures: the input and output quantization resolutions are set to 7-bit and 9-bit, respectively; the output read noise is set to 0.06; and the input/output bounds are set to 1 and 12. This configuration is intended to reflect realistic analog hardware non-idealities; the corresponding parameters are summarized in Table 7.

### F.4. Hyperparameter settings for Figure 4

For the ablation study comparing the total pulse budget of E-RIDER against the baseline zero-shifting algorithm, both methods use the same SoftBounds-based preset in the RPU configuration, and we set the desired update pulse length to be 5. For E-RIDER, we set the global learning rate to 0.1; the device-side learning rates are `fast_lr`=0.1 and `transfer_lr`=1, with `auto_granularity`=1000 and `in_chop_prob`=0.2. For the baseline zero-shifting setup, we use the TT-v2 algorithm after SP estimation is completed, train with a global learning rate of 0.05, and set `fast_lr`=0.01 and `transfer_lr`=1. We report the hyperparameter settings used in our CIFAR-100 experiments with ResNet-18. We replace the final fully connected layer and the last residual block with analog counterparts. For E-RIDER and AGAD, we report results after 80 epochs; for TT-v2, which consistently diverges during training, we report the best accuracy achieved before divergence. We use a batch size of 128, and decay the learning rate by a factor of 0.1 every 35 epochs. The complete set of hyperparameter is reported in Table 5.

### F.5. Larger-scale ImageNet-1K fine-tuning experiment

To test the performance of E-RIDER on larger-scale dataset, we additionally evaluated E-RIDER on a reduced-scale ImageNet-1K setting. We construct a class-balanced training subset by sampling 200 images per class from the ImageNet-1K training split, and retain the full 50k-image validation set for evaluation. The architecture is a standard VGG-11-BN

*Table 7.* IO and analog readout noise settings for the forward, backward, and transfer-forward passes.

| Parameter | Forward | Backward | Transfer-forward |
|---|---|---|---|
| MV type | ONE_PASS | ONE_PASS | ONE_PASS |
| Noise management | ABS_MAX | ABS_MAX | NONE |
| Bound management | ITERATIVE | ITERATIVE | NONE |
| Input bound (`inp_bound`) | 1.0 | 1.0 | 1.0 |
| Input resolution (`inp_res`) | 0.0079365 | 0.0079365 | 0.0079365 |
| Input noise (`inp_noise`) | 0.0 | 0.0 | 0.0 |
| Output bound (`out_bound`) | 12.0 | 12.0 | 12.0 |
| Output resolution (`out_res`) | 0.0019608 | 0.0019608 | 0.0019608 |
| Output noise (`out_noise`) | 0.06 | 0.06 | 0.06 |
| Input stochastic rounding | False | False | False |
| Output stochastic rounding | False | False | False |

*Table 8.* Top-1 test accuracy for two dynamic SP tracking methods E-RIDER and AGAD on the reduced-scale ImageNet-1K setting under different reference mean/std. Best results are highlighted in bold.

| Method | Mean＼Std | 0.05 | 0.4 | 0.7 | 1.0 |
|---|---|---|---|---|---|
| AGAD | 0.05 | 63.82 | 63.40 | 63.60 | 63.67 |
| E-RIDER | | **66.41** | **66.48** | **66.62** | **66.65** |
| AGAD | 0.20 | 63.61 | 63.77 | 64.06 | 64.05 |
| E-RIDER | | **66.56** | **66.64** | **66.66** | **66.32** |
| AGAD | 0.30 | 63.74 | 64.05 | 63.75 | 64.09 |
| E-RIDER | | **66.51** | **66.35** | **66.21** | **66.37** |
| AGAD | 0.40 | 64.02 | 63.66 | 63.64 | 64.07 |
| E-RIDER | | **66.35** | **66.34** | **66.19** | **66.13** |

model with 8 convolutional layers and 3 fully connected layers, containing about 132M parameters. We replace the final two fully connected layers, fc2 and fc3, with analog layers of dimensions $4096 \times 4096$ and $4096 \times 1000$, respectively.

For the non-ideal setting, we use the same non-perfect IO configuration as in Table 7. The hardware device we use is the ReRamArrayOMPresetDevice (Gong et al., 2022b). Its device-to-device variation, cycle-to-cycle variation, asymmetry, and limited conductance states are summarized in Table 3. The update asymmetry is modeled through soft bounds with separate up and down update responses provided in IBM AIHWKit simulator (Rasch et al., 2021).

The pretrained VGG-11-BN model achieves $70.18\%$ top-1 test accuracy under digital evaluation. Direct deployment with non-ideal analog layers reduces test accuracy to $63.17\%$. We then fine-tune the analog model and compare E-RIDER against the dynamic SP-tracking baseline AGAD. Table 8 reports the resulting top-1 test accuracy under different offsets of the initial reference mean and standard deviation.

E-RIDER consistently improves over AGAD by $2 - 3\%$ across all tested reference offsets. Both methods remain robust to imperfect initial SP estimation, but E-RIDER maintains a clear improvement. These results indicate that the proposed E-RIDER scales beyond the MNIST and CIFAR benchmarks used in the main paper.

### F.6. Ablation studies of key hyperparameters

We provide additional ablation studies for three key E-RIDER hyperparameters: chopping probability $p$, moving averaging stepsize $\eta$ and residual learning perturbation $\gamma$. First, Figure 5 shows the effect of the chopper probability $p$ on MNIST-FCN. When $p = 0$, E-RIDER reduces to RIDER and the test accuracy drops sharply. For all tested $p > 0$, however, the final accuracy remains stable. We therefore use $p = 0.1$ as the default setting.

The moving-average stepsize $\eta$ controls the update speed of the SP-tracking estimate. As shown in Table 9, performance is stable once $\eta > 0$. The best final accuracy occurs near our default $\eta = 0.5$.

The residual learning perturbation $\gamma$ controls the contribution of the residual term in the gradient computation. Table 10 shows stable convergence for $\gamma \in [0.1, 0.4]$, with the best performance at $\gamma = 0.1$, which is set as the default value for E-RIDER. Starting at $\gamma = 0.5$, the final accuracy collapses.

*Table 9.* Ablation of the moving-average stepsize $\eta$ in E-RIDER.

| $\eta$ | 0.0 | 0.2 | 0.4 | 0.6 | 0.8 | 1.0 |
|---|---|---|---|---|---|---|
| Final acc. (%) | 89.43 | 91.46 | **93.57** | 92.06 | 91.38 | 92.01 |
| Best acc. (%) | 89.43 | 93.42 | **93.93** | 92.06 | 92.33 | 93.36 |

*Table 10.* Ablation of the residual learning perturbation $\gamma$ in E-RIDER.

| $\gamma$ | 0.1 | 0.2 | 0.3 | 0.4 | 0.5 | 0.6 | 0.7 |
|---|---|---|---|---|---|---|---|
| Final acc. (%) | **93.72** | 93.01 | 92.81 | 89.09 | 11.35 | 11.61 | 11.35 |
| Best acc. (%) | **93.72** | 93.01 | 92.81 | 89.09 | 33.04 | 56.88 | 49.61 |

