# OpenReview forum: "Dynamic Symmetric Point Tracking: Tackling Non-ideal Reference in Analog In-memory Training"
_ICML.cc/2026/Conference — ICML 2026 regular_

### Official Review · Reviewer_ey7n · 2026-03-13

**Soundness:** 3
**Presentation:** 3
**Significance:** 3
**Originality:** 3
**Overall Recommendation:** 4
**Confidence:** 3

**Summary:**

This paper studies analog in-memory computing (AIMC) training under update asymmetry, where device dynamics induce drift toward a device-specific symmetric point (SP) that may not coincide with the optimum of the learning objective. The paper first analyzes the pulse complexity of standard zero-shifting (ZS) calibration for estimating the SP and argues theoretically that the pulse cost grows inversely with the device update granularity. It then proposes RIDER, a dynamic SP tracking method that jointly performs model training and SP estimation during training, instead of relying on a separate pre-calibration stage. The paper further introduces E-RIDER, an enhanced variant that combines chopping and moving-average filtering to accelerate SP tracking and reduce programming overhead.

**Compliance With Llm Reviewing Policy:**

Affirmed.

**Final Justification:**

Thank you for your response. I appreciate your clarification, and I would like to maintain my original score.

**Key Questions For Authors:**

1. How essential is strong convexity to the main convergence result?
If the guarantees rely critically on strong convexity, please discuss whether an analogous result can be expected under weaker conditions, such as PL-type conditions or in nonconvex settings.

2. How sensitive are RIDER and E-RIDER to $\eta$, $\gamma$, and the chopping probability $p$?

3. Can authors quantify the cost of periodic synchronization in E-RIDER?

**Limitations:**

Yes, the paper includes an impact statement and mentions energy and carbon benefits.

**Strengths And Weaknesses:**

**Strengths:**

- The paper addresses a well-motivated hardware-aware optimization problem in analog in-memory training. The analysis of ZS provides a useful scaling law for pulse complexity, and the RIDER result establishes convergence with an $O(1/\sqrt{K})$ rate plus a hardware-dependent error floor.

- The distinction between RIDER and E-RIDER is explained reasonably well, especially through the signal-processing interpretation of chopping and filtering.

- It connects optimization theory, device-level non-idealities, and algorithm design.

**Weaknesses:**

- The empirical evaluation, while useful, is still limited. The neural architectures and datasets are relatively modest, and the CIFAR-100 experiment uses only partially analog training rather than a fully analog large-scale setting.

- While the theoretical analysis is valuable, the mathematical model assumes relatively stable device parameters and overlooks unavoidable real-world physical constraints.

---

> ### Author Rebuttal · Authors · 2026-03-30
>
> We thank the reviewer for appreciating our theoretical contributions and algorithms. We address the remaining concerns below.
>
> **Q1. Performance on complex and larger-scale dataset.**
>
> Please see the response to **Reviewer e2m3, Q3** for new ImageNet-1K results on VGG-11-BN (132M params).
>
> **Q2. Stable device parameters assumptions.**
>
>
> Our simulations are based on AIHWKit simulator, which already include device and cycle variations to reflect practical instability. Under these nonideal settings, E-RIDER still outperforms the baselines, which aligns with the intuition that online SP tracking is more adaptive to SP variation than a two-stage approach. Theoretically, we focus on the fixed-SP setting because device properties are approximately stable over a training run under fixed programming conditions [R7, R10]. Extending to time-varying SP via an online optimization framework is an interesting future direction.
>
>
>
> **Q3. Strong convexity assumption and PL extension.**
>
> Extending to PL is possible but challenging, as we need to handle multiple lower-level solutions in (8); see e.g.,
>
> \begin{align}
>     \min_{W, P\in\mathcal{S}^{\star}(W,Q_k)} ~\\|P-Q_k\\|^2,  \text{ s.t. }    \mathcal{S}^{\star} (W,Q_k):=\arg\min_{P} ~f(W + \gamma (P-Q_k))
> \end{align}
>
> Under PL, the overall bilevel objective $F(W):=\min_{P\in\mathcal{S}^*(W,Q_k)} ~\\|P-Q_k\\|^2=\operatorname{dist}(W,\mathcal{W})^2$ can be both non-differentiable and nonconvex. In such case, simple stochastic subgradient descent approach may not converge without additional regularity conditions [R12]. Moreover, in Theorem 3.7, we establish global convergence for the bilevel problem (8), not merely stationary convergence. This is harder under PL, since global convergence for general bilevel problems is already challenging, and the sum of a PL function and a least-squares term is not PL in general [R11]. We leave the PL extension to future work.
>
> **Q4. Ablation studies of key hyperparameters.**
>
> We provided the ablation studies for key hyperparameters here.
>
> - **Chopper probablity $p$.** Figure 5 (Appendix F.4, MNIST-FCN setting) shows that for the case $p\neq 0$, **the test accuracy of E-RIDER remains relatively stable**, whether $p$ is close to $1$ (e.g., $0.8$), close to $0$ (e.g., $0.1$), or at an intermediate level (e.g., $0.4$). We observe this trend across the datasets we tested, and thus use $p=0.1$ as default.
> - **Moving average stepsize $\eta$.**
>
> | $\eta$ | 0.0 | 0.2 | 0.4 | 0.6 | 0.8 | 1.0 |
> |-|-|-|-|-|-|-|
> | final acc | 0.8943 | 0.9146 | 0.9357 | 0.9206 | 0.9138 | 0.9201 |
> | best acc | 0.8943 | 0.9342 | 0.9393 | 0.9206 | 0.9233 | 0.9336 |
>
> Performance is stable for $\eta > 0$. Default $\eta=0.5$; the best final accuracy is near our default setting.
>
>
> - **Residual learning perturbation $\gamma$.**
>
> | $\gamma$ | 0.1 | 0.2 | 0.3 | 0.4 | 0.5 | 0.6 | 0.7 |
> |-|-|-|-|-|-|-|-|
> | final acc | 0.9372 | 0.9301 | 0.9281 | 0.8909 | 0.1135 | 0.1161 | 0.1135 |
> | best acc | 0.9372 | 0.9301 | 0.9281 | 0.8909 | 0.3304 | 0.5688 | 0.4961 |
>
>
> $\gamma$ from 0.1 to 0.4 yields stable convergence, with best performance at $\gamma=0.1$. Starting from $\gamma=0.5$, accuracy collapses to random-guess level, indicating a clear phase transition that informs practical hyperparameter selection.
>
>
> **Q5. Cost of periodic synchronization in E-RIDER.**
>
>
> Please refer to **Reviewer e2m3, Q1, 2nd point**: with default $p=0.1$, synchronization occurs every ~10 iterations at average cost $O(pD)$ per iteration, negligible vs. the $O(BD)$ analog MVM cost where $B \gg p$.
>
>
> > [R1] Wu, et al. Analog In-memory Training on General Non-ideal Resistive Elements: The Impact of Response Functions. NeurIPS, 2025.
>
> > [R2] Wu, et al. Towards exact gradient-based training on analog in-memory com- puting. NeurIPS, 2024.
>
> > [R3] Stecconi, et al. Analog resistive switching devices for training deep neural networks with the novel Tiki-Taka algorithm. Nano Letters, 2024.
>
> > [R4] Tang, et al. ECRAM as scalable synaptic cell for high-speed, low-power neuromorphic computing. IEDM, 2018.
>
> > [R5] Jang, et al. ReRAM-based synaptic device for neuromorphic computing. ISCAS, 2014.
>
> > [R6] Burr, et al. Recent Progress in Phase-Change Memory Technology. IEEE JETCAS, 2016.
>
> > [R7] Noh, et al. Retention-aware zero-shifting technique for Tiki-Taka algorithm-based analog deep learning accelerator. Science advances, 2024.
>
> > [R8] Kim, et al. Zero-shifting technique for deep neural network training on resistive cross-point arrays. arXiv preprint:1907.10228, 2019.
>
> > [R9] Rasch, et al. Fast and robust analog in-memory deep neural network training. Nature Comms., 2024.
>
> > [R10] Gong, et al. Deep learning acceleration in 14nm cmos compatible reram array: device, material and algorithm co-optimization. 2022.
>
> > [R11] Xiao, et al. Unlocking Global Optimality in Bilevel Optimization: A Pilot Study. ICLR, 2025.
>
> > [R12] Bolte, et al. Subgradient sampling for nonsmooth nonconvex minimization. SIOPT, 2023.

---

> > ### Author Rebuttal · Reviewer_ey7n · 2026-04-04
> >
> > I'd like to thank the authors for their clarifications. Most of my questions have been answered.

---

> > > ### Author Response · Authors · 2026-04-04
> > >
> > > We sincerely thank the reviewer for the careful reading and constructive feedback. We are glad that most of your concerns have been resolved, and we will include the ImageNet experiments and ablation studies in the main paper as suggested.
> > >
> > > We would be grateful if you would consider revising your score, and we would be happy to address any further questions you may have.

---

### Official Review · Reviewer_zY4u · 2026-03-13

**Soundness:** 3
**Presentation:** 3
**Significance:** 3
**Originality:** 3
**Overall Recommendation:** 3
**Confidence:** 3

**Summary:**

In this work, the author proposes theoretical characteristics of dynamic symmetric point tracking in analog in-memory computing. This method is used within the in-memory computing paradigm, which offers energy efficiency for vision and language models. The proposed approach is based on a point tracking algorithm that jointly performs tracking and training on the analog device.

**Compliance With Llm Reviewing Policy:**

Affirmed.

**Final Justification:**

While the rebuttal mostly replies to the questions -does not mean fully resolving-, I still have some concerns about the overall device-level evaluations and the applicability, plus the algorithm clarifications. I do not think either (b) or (c) selection in rebuttal feedback by me will answer my questions. Please consider my decision between (a) - (b). I am keeping my scores as is. I am not inclined toward one more round of discussion. My old score was 3; if the consensus is the acceptance, I am fine for this.

**Key Questions For Authors:**

1- How does the proposed method generalize to larger-scale and more complex datasets?

2- Can the authors better explain the hardware assumptions, especially device asymmetry, pulse efficiency, and the device-specific limitations that motivate the method?

3- What is the exact role of the multiple algorithms, especially Algorithm 3, and is it -only- an improved version of the earlier methods or a different setting for specific devices-applications?

4- How general is the proposed method across different device types, or is it tied to specific analog-in-memory hardware and certain assumptions?

**Limitations:**

Impact was added, but not the limitations.

**Strengths And Weaknesses:**

The proposed method appears novel from a theoretical perspective. The results on the datasets support the claims made in the experiments. However, the main weakness of this work is its performance on complex and larger-scale data.

The introduction part, while technically sound, would have been presented in a more advertising style for the machine learning conference, especially the tasks and applications we can run on an analog-in-memory-based learning system (theoretically, it will be limited).

The device asymmetry concept needs a better definition and some literature support. Even some more hardware-heavy discussion for this specific component would suffice. "However, in practice, the SP is device-specific, so it requires per-device estimation and calibration to zero before training" is not enough to show ML community the regarding detailed device problems; what specifics from the memory-centric devices create this bottleneck?

In general, the hardware terms need a greater explanation, reference, and discussion. (e.g., pulse efficiency)

Figure 2 is not explained clearly.

"Q2) Can we develop a dynamic SP tracking algorithm during training to reduce the overall pulse complexity?" -> Is it not possible with the current algorithms to solve this problem? To what extent is this related to the novelty of the paper?

In the definition 2.1 explanation, the expected hardware-based discussion is very briefly presented over the different devices. Seemingly, the proposed modelling in the paper is generic, but still, how much device-specific this model is should be discussed.

While keeping two different algorithms with different perspectives, and also an enhanced version, the reviewer is not sure whether this simplifies the discussion by broadening the scientific scope or just complicates the process.

Is Algorithm 3 just an improved version of the previous, or is it a different setting based on the specific applications (e.g., signal processing) and devices?

The proposed method is evaluated on the MNIST and CIFAR datasets and demonstrates clear performance improvements over baseline methods, supporting the main claims of the work. However, a detailed ablation study is not provided to identify the effective components of the proposed method. The manuscript can also benefit from different datasets, or at least variants of MNIST, given the heavy workload in the Appendix; in this studious work, only tests on MNIST - CIFAR is limited. It would make more sense to also apply the method to larger and more complex datasets, such as ImageNet, COCO, and Google Open Images. In general, the experimental part is relatively weak. There is also no hardware-device discussion in the results.

---

> ### Author Rebuttal · Authors · 2026-03-30
>
> We thank the reviewer for appreciating our theoretical novelty. We hope our response to your comments below can resolve your remaining concerns.
>
> **Q1. Performance on larger-scale dataset.**
>
> Please see the response to **Reviewer e2m3, Q3** for new ImageNet-1K fine-tuning results on VGG-11-BN. E-RIDER consistently improves over AGAD by ~2-3%.
>
> **Q2. Hardware assumptions.**
>
> We will expand hardware background in the final version using the additional page.
> - **Device asymmetry.** Device asymmetry is defined as $G(\cdot)\not\equiv 0$ (cf. $G(\cdot)$ in (6b)), or equivalently, by unequal potentiation and depression responses $q_+(w)\not\equiv q_-(w)$. A nonzero $G(\cdot)$ introduces a systematic bias in (2), causing weight updates to drift toward the device-specific SP rather than the training optimum $W^∗$; see (4). In practice, this arises from asymmetric physical mechanisms: e.g., conductive filament formation/rupture in HfO$_2$-based ReRAM device [R5], or crystallization/amorphization in PCM [R6]. Device models and SP derivations are in *Appendix F.1, Eqs. (103)-(104)*. Notably, asymmetric devices offer practical advantages such as faster response and lower pulse voltage [R3], making them attractive for deployment.
> - **Definition 2.1.** On analog device, the weight updates as either $w^\prime=w+q_+(w)\Delta w_{\min}$ or $w^\prime=w-q_-(w) \Delta w_{\min}$, so the response function describes how current weight $w$ responds to a small weight change $\Delta w_{\min}$; see [R1, Figure 1]. Training-friendly response functions capture the practical regime: the lower bound $q_{\min}$ excludes dead-update regions; the upper bound $q_{\max}$ rules out unbounded gains. Differentiability is motivated by continuous conductance mechanisms (ion migration, phase boundary movement, electrochemical reactions) in PCM, ReRAM, and ECRAM [R1,R2,R3].
> - **Device-specific limitations and pulse efficiency**. We study two key device limitations: asymmetry and response granularity $\Delta w_{\min}$ (quantization precision).  Most existing methods assume a known SP and therefore follow an inherently two-stage approach: using zero-shifting method to estimate the SP before training. This work reveals a **device dilemma** in Theorem 2.2: smaller $\Delta w_{\min}$ improves accuracy but increases the pulse complexity in zero-shifting algorithms. This explains why two-stage approaches are not pulse-efficient, motivating our dynamic one-stage framework. *Pulse complexity* is the minimum number of programming pulses (the number of applying (2)) required to achieve a target training error $\epsilon$. Prior works focus on accuracy or SP-error robustness but do not formally quantify pulse complexity. While some prior methods [R9] also perform dynamic SP tracking, our work is the first to characterize overall pulse efficiency and to show, both theoretically (Corollary 3.9) and empirically (Figure 4), improvements over two-stage approaches [R7–R8].
>
> **Q3. Different roles of Algorithm 2 and 3.**
>
> Algorithm 2 (RIDER) is recovered from Algorithm 3 (E-RIDER) by setting $p=0$. We keep both algorithms because they provide different theoretical insights and we want to introduce the techniques we added progressively. Algorithm 2 captures the core mechanism: the moving-average $Q_k$ alone guarantees convergence. Algorithm 3 builds on this by adding the chopper-and-filtering design (Lemma 3.10), which further improves separation between gradient signal and SP-attraction, as confirmed by ablation results in the response to **Reviewer ey7n, Q4**.
>
> **Q4. Generality across different device types.**
>
> Our method applies to any device with response functions satisfying Definition 2.1. Within AIMC, this covers PCM, ReRAM [R3], CBRAM, ECRAM [R4], MRAM, FTJ, etc; see [R1] for additional references.
>
> **Q5&Q7. Presentation and hardware-device discussion.**
>
> We will make the introduction more ML-facing by clarifying that AIMC devices are suited for on-device training under energy/memory constraints. We will also make the experimental hardware setup (HfO$_2$-based RRAM preset [R5, R10]; details in Appendix F) more explicit; see also our response to **Reviewer e2m3, Q2**.
>
> **Q6. Explanation of Figure 2.**
>
> Figure 2 shows the two-stage approach: Stage I uses zero-shifting with $N$ pulses to estimate the SP, then Stage II applies TTv2 for training using that estimate. The figure plots training loss vs. epochs under varying $N$. The figure shows that when $N$ is insufficient $<6$k, the inaccurate SP estimation significantly degrades subsequent training, motivating our joint approach that avoids this bottleneck.
>
> **Q8. Ablation studies.**
>
> Relative to our baselines, AGAD can be viewed as TTv2 augmented with chopper-and-filtering, while our method further incorporates residual learning. The experimental comparisons thus serve as component-wise ablations. We also provided the ablation studies for key hyperparameters in the response to **Reviewer ey7n, Q4**.

---

> > ### Author Rebuttal · Reviewer_zY4u · 2026-04-04
> >
> > My questions have been answered mostly; I am keeping my scores as is.

---

> > > ### Author Response · Authors · 2026-04-04
> > >
> > > We sincerely thank the reviewer for the careful reading and constructive feedback. We will include the ImageNet experiments and additional hardware explanation in the main paper as suggested.
> > >
> > > Since you indicated that your concerns have been fully resolved, we would be very grateful if you could kindly reconsider the current score. If there are any remaining questions or additional concerns that may affect your evaluation, we would greatly appreciate the opportunity to address them.

---

### Official Review · Reviewer_e2m3 · 2026-03-13

**Soundness:** 3
**Presentation:** 3
**Significance:** 3
**Originality:** 3
**Overall Recommendation:** 5
**Confidence:** 3

**Summary:**

The paper addresses the problem of inaccurate model training on special energy-saving computer hardware due to its inherent flaws, and proposes new methods that track and adjust for these flaws during training to make the training more efficient and accurate without extra costly preparations.

**Compliance With Llm Reviewing Policy:**

Affirmed.

**Final Justification:**

The rebuttal satisfactorily addressed my main concern about scalability, and the additional ImageNet-1K experiment provides convincing evidence that E-RIDER remains effective in larger and more realistic analog fine-tuning settings.

**Key Questions For Authors:**

How does E-RIDER perform when trained and tested on large-scale, high-dimensional datasets (e.g., ImageNet, COCO) that have more complex data distributions, compared to its performance on small-to-medium datasets like MNIST and CIFAR-100?

**Strengths And Weaknesses:**

Strengths

1. This paper pioneers a comprehensive theoretical analysis of the ZS algorithm’s pulse complexity, clarifying its inverse relationship with device granularity and filling critical research gaps.

2. RIDER enables efficient joint model training and dynamic SP tracking with low pulse cost.

3. Enhances with E-RIDER, integrating chopping/filtering for faster SP estimation and periodic synchronization to cut weight programming costs, ideal for edge devices.

4. Demonstrates strong robustness and superior accuracy across MNIST/CIFAR-100 datasets and LeNet-5/ResNet-18 models under varying SP offset scenarios.

5. Offers a practical low-power solution for AIMC training by mitigating device update asymmetry, avoiding costly pre-calibration and residual errors.

Weaknesses

1. The enhanced E-RIDER relies on chopping and filtering techniques as well as periodic synchronization mechanisms, which adds certain complexity to the algorithm’s practical implementation and requires careful tuning of hyperparameters like chopper probability.

2.  The method’s performance gain heavily depends on the effectiveness of dynamic SP tracking, and in scenarios with extreme device non-idealities, its stability and error mitigation capability still need further verification.

---

> ### Author Rebuttal · Authors · 2026-03-30
>
> We thank the reviewer for appreciating our theoretical contribution and the efficient algorithm design. We address the remaining concerns below.
>
> **Q1. Hyperparameter tuning and implementation for E-RIDER.**
>
> - **Chopping and filtering technique**. The key hyperparameters are chopper probability $p$ and the stepsize $\eta$ for the moving averaging filter. We have provided the sensitivity analysis over both in the response to **Reviewer ey7n, Q4**. Both of them are insensitive and easy-to-tune with default value.
>
> - **Periodical synchronization**. In practice, we program the digital $Q_k$ to the analog array $\tilde Q_k$ column by column whenever the chopper variable flips sign, and we also account for writing noise in the implementation. With the default chopper probability $p=0.1$, synchronization occurs once every ~10 iterations. Each synchronization incurs a programming cost of ${\cal O}(D)$, where $D$ denotes the dimension of the weight $W$. The average synchronization cost per iteration is thus ${\cal O}(pD)$, , which is not a bottleneck since the analog MVM cost scales as ${\cal O}(BD)$, where $B \gg p$ is the batch size.
>
> **Q2. Stability and error mitigation capability of dynamic SP tracking.**
>
> In practice, SP skew is usually small. In our simulation on MNIST and CIFAR 100, for the hardware device, we adopt HfO$_2$-based ReRAM model [R10], whose non-idealities include device-to-device and cycle-to-cycle variation, asymmetry, and limited conductance states, as summarized in the table below:
> |  ($\tau_{\min}, \tau_{\max}$)  | $\Delta w_{\min}$ | d2d variation | c2c variation |
> |---:|---:|---:|---:|
> | (-1.0, 1.0) | 0.4622 | 0.7125 | 0.2174 |
>
> Asymmetry is modeled through the soft-bounds formulation with separate up/down update response functions (detailed model in Appendix F.1, Eq. 106). In particular, the mismatch between $q_{+}(w)$ and $q_{-}(w)$ is influenced by the upper and lower weight boundaries, $\tau_{\max}$ and $\tau_{\min}$. For this preset, the measured ground-truth SP is about $0.014$, and [R3] reports SP skew below $5%$ in an RRAM device. Given these statistics, reference mean $0.4$ and standard deviation $1.0$ in Tables 1 and 2 can be considered as a much more challenging setting. The results show that our method remains stable and effective even under this extreme scenario.
>
>
> **Q3. Experiments on larger-scale dataset (ImageNet-1K)**
>
> During the rebuttal, we conducted experiments on a reduced-scale ImageNet-1K setting. We construct a class-balanced training subset by sampling 200 images per class from ImageNet-1K training split. For evaluation, we retain the full 50k-image validation set.
>
> **Architecture:** Standard VGG-11-BN (8 conv + 3 FC layers, ~132M params). We replace fc2 and fc3 with analog layers of dimensions (4096, 4096) and (4096, 1000).  For the non-ideal setting, we use the non-perfect I/O configuration similar to that in Table 6 of Appendix F.2. For the hardware device, we adopt ReRamArrayOMPresetDevice [R10], whose non-idealities include device-to-device (d2d) and cycle-to-cycle (c2c) variation, asymmetry, and limited conductance states, as summarized in the table below:
> |  ($\tau_{\min}, \tau_{\max}$)  | $\Delta w_{\min}$ | d2d variation  | c2c variation |
> |---:|---:|---:|---:|
> | (-1.0, 1.0) | 0.0949 | 0.7829  | 0.4158 |
>
> Asymmetry is modeled via soft-bounds with separate up/down update responses.
>
>
> The pretrained VGG-11-BN achieves 0.7018 top-1 accuracy under digital evaluation. Deploying with non-ideal analog layers drops accuracy to 0.6317. We fine-tune and compare E-RIDER against the AGAD baseline. Results (top-1 accuracy):
>
> **AGAD**
>
> | mean \ std | 0.05   | 0.4    | 0.7    | 1.0    |
> | ---------- | ------ | ------ | ------ | ------ |
> | 0.05       | 0.6382 | 0.6340 | 0.6360 | 0.6367 |
> | 0.20       | 0.6361 | 0.6377 | 0.6406 | 0.6405 |
> | 0.30       | 0.6374 | 0.6405 | 0.6375 | 0.6409 |
> | 0.40       | 0.6402 | 0.6366 | 0.6364 | 0.6407 |
>
> **E-RIDER**
>
> | mean \ std | 0.05   | 0.4    | 0.7    | 1.0    |
> | ---------- | ------ | ------ | ------ | ------ |
> | 0.05       | 0.6641 | 0.6648 | 0.6662 | 0.6665 |
> | 0.20       | 0.6656 | 0.6664 | 0.6666 | 0.6632 |
> | 0.30       | 0.6651 | 0.6635 | 0.6621 | 0.6637 |
> | 0.40       | 0.6635 | 0.6634 | 0.6619 | 0.6613 |
>
>
> E-RIDER consistently outperforms AGAD by ~2-3%. Both methods remain robust to imperfect initial SP estimation (large reference mean/std offset), with E-RIDER maintaining clear superiority across all settings. These results on a 132M-parameter model with 1000 classes demonstrate the scalability of our approach to realistic analog fine-tuning scenarios beyond the MNIST/CIFAR benchmarks in the main paper.

---

> > ### Author Rebuttal · Reviewer_e2m3 · 2026-04-04
> >
> > Thank you for your response. I maintain my score.

---

> > > ### Author Response · Authors · 2026-04-05
> > >
> > > We sincerely thank the reviewer for the constructive feedback and suggestions. We will include the ImageNet experiments and ablation studies in the main paper.

---

### Decision · Program_Chairs · 2026-04-30

**Decision:**

Accept (regular)

**Comment:**

This paper provides a comprehensive theoretical analysis of the pulse complexity of the ZS algorithm, clarifying its inverse relationship with device granularity and filling critical research gaps. It proposes new methods that render training more efficient and accurate without requiring additional costly preparations.